# Transformers Provably Learn Two-Mixture of Linear Classification via Gradient Flow

**Hongru Yang** *
The University of Texas at Austin
& Princeton University
`hy6385@utexas.edu`

**Zhangyang Wang**
The University of Texas at Austin
`atlaswang@utexas.edu`

**Jason D. Lee**
Princeton University
`jasonlee@princeton.edu`

**Yingbin Liang**
The Ohio State University
`liang.889@osu.edu`

## Abstract

Understanding how transformers learn and utilize hidden connections between words is crucial to understand the behavior of large language models. To study this mechanism, we consider the task of two-mixture of linear classification which features a hidden correspondence structure between words, and study the training dynamics of a symmetric two-headed transformer with ReLU neurons. Motivated by the stage-wise learning phenomenon observed in our experiments, we design and theoretically analyze a three-stage training algorithm, which can effectively characterize the actual gradient descent dynamics when we simultaneously train the neuron weights and the softmax attention. The first stage is a neuron learning stage, where the neurons align with the underlying signals. The second stage is an attention feature learning stage, where we analyze the feature learning process of how the attention learns to utilize the relationship between the tokens to solve certain hard samples. In the meantime, the attention features evolve from a nearly non-separable state (at the initialization) to a well-separated state. The third stage is a convergence stage, where the population loss is driven towards zero. The key technique in our analysis of softmax attention is to identify a critical sub-system inside a large dynamical system and bound the growth of the non-linear sub-system by a linear system. Along the way, we utilize a novel structure called *mean-field infinite-width transformer*. Finally, we discuss the setting with more than two mixtures. We empirically show the difficulty of generalizing our analysis of the gradient flow dynamics to the case even when the number of mixtures equals three, although the transformer can still successfully learn such distribution. On the other hand, we show by construction that there exists a transformer that can solve mixture of linear classification given any arbitrary number of mixtures.

## 1 Introduction

Since the invention of self-attention (Vaswani et al., 2017), transformers have become the dominating backbone architecture in many machine learning applications such as computer vision (Dosovitskiy et al., 2020; Liu et al., 2021), natural language processing (Devlin et al., 2018) and protein structure prediction (Jumper et al., 2021). Within the past two years, ChatGPT and GPT-4 (OpenAI, 2023; Bubeck et al., 2023) along with other large language models (LLMs) (Touvron et al., 2023; Manyika & Hsiao, 2023; Gemini, 2023; Anthropic, 2024) have demonstrated astonishing abilities in language understanding, math Olympiad (AlphaProof & AlphaGeometry, 2024) and coding, etc.

Despite the wide range of success of transformers, how those models can achieve such impressive performance still remains largely unknown. One mystery lies in how transformers are trained to utilize the connections between words to solve various tasks. For example, consider the prompt

---

*Work done while visiting Princeton University

"Bob is watching television in the living room. Where is Bob?". To answer the question, the model must utilize the (hidden) correspondence between "Where" and "living room".

There are many previous works studying the mechanism of self-attention under different settings through training dynamics such as (Li et al., 2024b; Huang et al., 2024; Li et al., 2022; Tarzanagh et al., 2023b;a; Vasudeva et al., 2024). However, many of those previous works were either focusing on attentions that are position-based or studying a setting where the words in the dictionary don't have explicit relationships between each other. The goal of this work is to study some simple yet informative setting such that we can get a clear understanding how the softmax attention learns and utilizes certain connections between words to solve tasks through training. To achieve this goal, we consider the task of mixtures of linear classification. In this task, we have a dictionary consisting of group identifiers and classification features, and there is a hidden **correspondence structure**: each group identifier is associated with one classification feature. Each input contains a group identifier and a corresponding signed classification feature. The rest of tokens are sampled from signed classification features from other groups. The label of the sequence is determined by the sign of the classification feature corresponding to the group identifier. Thus, a model needs to capture the correspondence structure between group identifiers and classification features to solve this task. To see how attention can solve this task, later in our training dynamics analysis, we are going to show that the softmax attention builds such correspondence structure by allocating more weights on the group identifier when it sees classification feature queries.

Although the intuition suggests that we can construct a transformer to solve this task, it is still unknown that how the transformer models can be *trained* to learn such a hidden correspondence structure. This inspires the following intriguing question:

> *How do transformers learn and utilize the hidden correspondence structure to solve mixture of linear classification via gradient descent?*

**Our contributions.** In this work, we study the training dynamics of a two-headed transformer given two mixtures. Our contributions are summarized as follows:

1. As a guidance for our theory, we first conduct experiments to observe the training dynamics of the transformer where we train all the weights simultaneously. Our experiment results show a clear stage-wise learning phenomenon where the neuron weights learn before the attention modules.

2. Motivated by our experimental observations, we design a three-stage layer-wise training algorithm and further analyze its gradient flow dynamics. We characterize the feature learning process where the attention features evolve from a nearly non-separable state to a well-separated state. In particular, our analysis captures how the self-attention associates the group signals and classification signals to solve this task. Our analysis closely reflects the actual behavior of the gradient descent dynamics when we train all weights simultaneously.

3. In order to analyze the change in softmax attention, we formulate a set of relevant variables whose evolution can be characterized by a large non-linear dynamical system of ordinary differential equations. To make the analysis of the dynamical system tractable, we reduce the dimension of the system by identifying some key variables and forming a smaller sub-system. We are able to relate the evolution of the smaller system with a linear dynamical system to track its dynamics. Our proof techniques can be of independent interest.

4. We explore the training dynamics of the transformers when the number of mixtures goes beyond two. We empirically show the difficulty of analyzing training dynamics even when the number of mixture equals three, although the two-headed transformer model can still learn this distribution. On the other hand, we give a general two-headed transformer construction that can solve mixture of linear classification given arbitrary number of mixtures.

## 1.1 RELATED WORK

**Mixture of linear regression/classification.** Mixture of linear regression/classification is a classical model in statistics and machine learning (De Veaux, 1989; Jordan & Jacobs, 1994) which was applied in areas such as object recognition (Quattoni et al., 2004) and machine translation (Liang et al., 2006). This problem can be solved by tensor-based methods (Anandkumar et al., 2014; 2012;

Hsu et al., 2012; Chaganty & Liang, 2013) and the expectation-maximization algorithm (Khalili & Chen, 2007; Yi et al., 2014; Balakrishnan et al., 2017; Wang et al., 2015). Previously, mixture of linear classification has been used in studying mixtures of experts (Chen et al., 2022).

**Training dynamics of transformers.** The training dynamics of transformers have been studied under various settings. One particular category is in-context learning. For example, in-context linear regression has been studied under linear attention (Zhang et al., 2024), softmax attention (Huang et al., 2023), non-linear embedding (Yang et al.), and multi-task with multiple heads (Chen et al., 2024). Other in-context learning settings include classification (Li et al., 2024b), causal structure (Nichani et al., 2024), Markov chain or n-gram (Edelman et al., 2024; Makkuva et al., 2024; Chen et al.), nearest neighbor (Li et al., 2024c) and chain-of-thought (Li et al., 2024a).

For non-in-context settings, it has been shown that trained transformer can learn spatial structures (Jelassi et al., 2022), topic models (Li et al., 2023b), and feature-position correlation (Huang et al., 2024). In addition, (Tian et al., 2023) showed that self-attention behaves like a discriminative scan algorithm. There is one line of works studying a setting where the attention weight can converge to a SVM solution (Tarzanagh et al., 2023b;a; Vasudeva et al., 2024). In addition, (Li et al., 2022) studied a classification task with label-relevant and label-irrelevant tokens. (Li et al., 2023a) provided analysis of training graph transformers for node classification tasks. (Wang et al.) showed transformers can learn a sparse token selection task which lead to an optimization-based separation between transformers and MLPs. There are also works trying to prove convergence of transformer training via NTK (Wu et al., 2023; Deora et al.) and mean-field (Kim & Suzuki, 2024; Gao et al.).

Our work also study a non-in-context learning setting. However, many of the previous works were either considering attentions that are position-based or studying a setting where the words don't have explicit relationships with each other. Our study complements the above work by considering a setting where the dictionary possesses a correspondence structure.

## 2 SETTINGS

**Notations.** For a vector $v \in \mathbb{R}^d$, we use $\text{diag}(v)$ to denote a diagonal matrix with $v$ being the diagonal entries. When we subtract the vector $v$ by a scalar $a$, we subtract each entry of $v$ by $a$, i.e., $v - a \in \mathbb{R}^d$ and $(v - a)_i = v_i - a$. For a set $\mathcal{S}$ with elements in $\mathbb{R}$ and $|\mathcal{S}| = n$, we can vectorize the set $\mathcal{S}$ in any arbitrary but fixed order and denote the vector as $\text{vec}(\mathcal{S}) \in \mathbb{R}^n$. We use $\widetilde{\Omega}, \widetilde{\Theta}, \widetilde{O}$ to hide polylogarithmic factors.

### 2.1 DATA DISTRIBUTION

**Definition 2.1** (Mixture of linear classification)**.** *Define the $K$-mixture of linear classification data distribution $\mathcal{D}$ as follows: Assume $d \geq 2K$. Let $\mathcal{C} = \{c_k\}_{k=1}^{K} \subset \mathbb{R}^d$ denote the group signals and $\mathcal{V} = \{v_k\}_{k=1}^{K} \subset \mathbb{R}^d$ denote the classification signals, where $\mathcal{C} \cup \mathcal{V}$ is orthonormal. Let $L$ be the number of tokens per sample. Each data entry $(X, y) \in \mathbb{R}^{d \times L} \times \{\pm 1\}$ is created as follows:*

1. *Sample $y \sim \text{Uniform}(\{\pm 1\})$.*

2. *Sample $k \sim \text{Uniform}([K])$ and $l_0 \sim \text{Uniform}([L])$. Set $X_{l_0} = c_k$.*

3. *Sample $l_1 \sim \text{Uniform}([L] \setminus \{l_0\})$ and set $X_{l_1} = yv_k$.*

4. *For each $l_2 \in [L] \setminus \{l_0, l_1\}$, sample $k' \overset{i.i.d.}{\sim} \text{Uniform}([K] \setminus \{k\})$ and set $X_{l_2} = \epsilon v_{k'}$, where $\epsilon \overset{i.i.d.}{\sim} \text{Uniform}(\{\pm 1\})$.*

Based on our data distribution, the model needs to utilize the correspondence between the group signals and classification signals to solve this task. We further elaborate our data model as follows. Take the sequence $X = [c_1, y_2 v_2, y_1 v_1, y_3 v_3]$ with $y_1, y_2, y_3 \in \{\pm 1\}$ as an example. Based on the form of $X$, the token $c_1$ indicates that this instance is in group 1. Then the label of this sequence depends on the sign associated with $v_1$ (that shares the same index with $c_1$), and hence is given by $y_1$.

**For the training dynamics part of this work, we consider $K = 2$ and $L = 3$,** and we discuss the setting with arbitrary number of mixtures in Section 6. Since the samples are invariant of permutation between tokens, we can ignore the order of the tokens and define $X_{k,s,s'} := (c_k, sv_k, s'v_{k'})$ for $k \neq k' \in [2], s, s' \in \{\pm 1\}$. We refer to the samples $X_{k,s,s}$ as **consistent samples** and samples $X_{k,s,-s}$ as **conflicting samples**.

## 2.2 TRANSFORMER ARCHITECTURE AND LOSS FUNCTION

Given inputs $X \in \mathbb{R}^{d \times L}$, we first define a single softmax attention head $H : \mathbb{R}^{d \times L} \to \mathbb{R}$ as

$$H(X) = \sum_{l=1}^{L} w^\top X \text{softmax} \left( X^\top W_K^\top W_Q x_l \right)$$

where $w \in \mathbb{R}^d$, $b \in \mathbb{R}$, $W_K, W_Q \in \mathbb{R}^{m \times d}$, and $x_l$ denotes the $l$-th token in $X$. Then, we define our one-layer symmetric two-headed transformer with ReLU neurons $f : \mathbb{R}^{d \times L} \to \mathbb{R}$ as

$$f(X) = \sigma \left( H_+(X) + b \right) - \sigma \left( H_-(X) + b \right) \tag{1}$$

where $b \in \mathbb{R}$ and $\sigma$ is the ReLU activation function. We denote the weights in $H_+$ as $w_+, W_{K+}, W_{Q+}$ and similarly for $H_-$. We introduce a shorthand notation $K_+ = W_{K+}X$ and $Q_+ = W_{Q+}X$. For $\mu, \nu \in X$, we define the output of attention, or softmax probability, as $p_{q\leftarrow\mu,k\leftarrow\nu}^{(+)}(X) := \text{softmax}\left( X^\top W_{K+}^\top W_{Q+}\mu \right)_{l(\nu)}$, where $l(\nu)$ denotes the index such that $X_{l(\nu)} = \nu$.

**Loss Function.** We train $w_\pm, W_{K\pm}, W_{Q\pm}$ to minimize the population loss $\mathbb{E}_{(X,y)}[\ell(yf(X))]$ where $\ell(x) = \log\left(1 + \exp\left(-x\right)\right)$ is the binary cross-entropy loss. The gradient of the cross-entropy loss is given as $\ell'(x) = -\frac{1}{1+e^x}$ and we denote $g(x) := 1/(1 + e^x)$.

## 3 GUIDING EXPERIMENTS AND ALGORITHM

### 3.1 EXPERIMENT RESULTS

To understand the dynamics of the training process, we train the two-headed transformer in Equation (1) on the data distribution in Definition 2.1. In our experiments, all the weights in the transformer are trained simultaneously via gradient descent with learning rate 0.1.

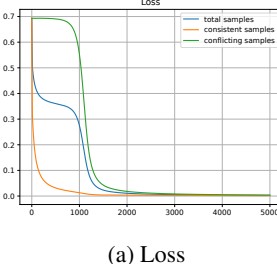
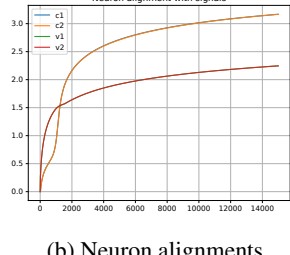
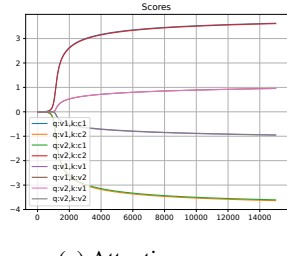

|  (a) Loss  |  (b) Neuron alignments  |  (c) Attention scores  |

Figure 1: Experiment results. The attention scores and neuron alignment are of the positive head.

We make the following observations from our experimental results in Figure 1, which will guide us to further develop our three-stage training algorithm and our analysis in Section 4:

- Learning for different types of samples follows different orders. From Figure 1a, the loss for the samples with consistent signs drops in the early stage of training, while the loss for the samples with conflicting signs only start to drop after sufficient amount of training.

- There is also a learning order of the weights. From Figure 1b, the neuron weights start to align with the underlying signals since the beginning of the training. On the other hand, by Figure 1c, only after sufficient amount of training, the attention scores start to exhibit noticeable change. We further conduct experiments by fixing the neuron weights as Gaussian and only training the attention. Our results show that the model is not trainable in this case.

- The time when the attention scores exhibit noticeable change coincides with the time when the loss for samples with conflicting signs drops. This suggests that attentions play a central role in learning those samples.

- The attention weights are positively associating the group signals and classification signals from the same group and negatively associating them from different groups. This can be seen from Figure 1c that the score $c_1^\top W_{K+}^{(t)\top} W_{Q+}^{(t)} v_1$ is increasing to a large positive magnitude and the score $c_1^\top W_{K+}^{(t)\top} W_{Q+}^{(t)} v_2$ is decreasing to a large negative magnitude.

In order to understand the mechanism of neurons and attentions separately, the second observation suggests to decompose the training process into stages: neurons are trained first, followed by training the attention. The process can be finished by training neurons again to drive the loss towards zero.

## 3.2 THREE-STAGE TRAINING ALGORITHM

Motivated by our experiment results, we introduce our three-stage layer-wise training algorithm in Algorithm 1. In our layer-wise training algorithm, in the first stage we train the neuron weights $w_+, w_-$. In the second stage, we train the attention weights $W_{K+}, W_{Q+}, W_{K-}, W_{Q-}$ on the samples with conflicting signs. In the third stage, we train the neuron weights again. Our algorithm design closely reflects our experimental observation in Section 3.

---

**Algorithm 1** Three-stage Training

---

Initialize $w_\pm^{(0)} = 0$, $W_{K\pm}^{(0)}, W_{Q\pm}^{(0)} \sim \mathcal{N}(0, \omega^2 \frac{1}{m})$ and $b$ to be a sufficiently small positive constant such as $1/2$.

**while** $t \leq T_1$ **do**               ▷ Stage 1: training the neuron weights with all samples

     $\frac{dw_\pm^{(t)}}{dt} = -\mathbb{E}_{(X,y)}[\nabla_{w_\pm} \ell(yf^{(t)}(X))]$.

     $[W_{K\pm}^{(t)}, W_{Q\pm}^{(t)}] = [W_{K\pm}^{(0)}, W_{Q\pm}^{(0)}]$.

**end while**

**while** $t \in [T_1, T_2]$ **do**       ▷ Stage 2: training the softmax attention with the conflicting samples

     $\frac{d[W_{K\pm}^{(t)}, W_{Q\pm}^{(t)}]}{dt} = -\mathbb{E}_{(X,y)}[\nabla_{[W_{K\pm}, W_{Q\pm}]} \ell(yf^{(t)}(X))|X = X_{k,s,-s}]$.

     $w_\pm^{(t)} = w_\pm^{(T_1)}$.

**end while**

**while** $t \in [T_2, T_3]$ **do**        ▷ Stage 3: training the neuron weights to minimize the population loss

     $\frac{dw_\pm^{(t)}}{dt} = -\mathbb{E}_{(X,y)}[\nabla_{w_\pm} \ell(yf^{(t)}(X))]$.

     $[W_{K\pm}^{(t)}, W_{Q\pm}^{(t)}] = [W_{K\pm}^{(T_2)}, W_{Q\pm}^{(T_2)}]$.

**end while**

---

## 4 MAIN RESULTS

Before we state the main results, we first introduce the parameter conditions used in our analysis.

**Condition 1.** *We use the following parameter conditions:*

- *The attention initialization scale satisfies $\omega < C < 1$ for some small constant $C$.*

- *The attention weight embedding dimension $m \geq \text{poly}(1/\omega)$.*

We now state the guarantee for Algorithm 1 under an idealized infinite-width transformer model which we will introduce later in Definition 5.1. Our experiment results show that this idealized model is indeed an accurate model that captures the real training dynamics.

**Theorem 4.1** (Main theorem)**.** *Under Condition 1 and the infinite-width transformer model in Definition 5.1, there exists $T_1 = \Theta(1)$, $T_2 - T_1 = O(\log(1/\omega))$, $T_3 - T_2 = O(1/\epsilon)$ such that after running Algorithm 1, we have $\mathbb{E}[\ell(yf^{(T_3)}(X))] \leq \epsilon$. The neuron weights of the transformer satisfies*

- *Positive alignment with the signed classification signals* $w_+^{(T_3)\top} v_k = \widetilde{\Theta}(1)$ *and* $w_-^{(T_3)\top}(-v_k) = \widetilde{\Theta}(1)$.

- *Positive alignment with the group signals* $w_\pm^{(T_3)\top} c_k = \widetilde{\Theta}(1)$.

- *Symmetry between heads and groups:* $w_+^{(T_3)\top} v_k = w_-^{(T_3)\top}(-v_k)$, $w_+^{(T_3)\top} v_1 = w_+^{(T_3)\top} v_2$, $w_+^{(T_3)\top} c_k = w_-^{(T_3)\top} c_k$, $w_+^{(T_3)\top} c_1 = w_+^{(T_3)\top} c_2$.

*In particular, the output of the softmax attention for sample* $X_{1,+,-}$ *satisfies the following properties:*

- *Concentration on the group signal key:*

$$p_{q\leftarrow v_1, k\leftarrow c_1}^{(+,T_3)}(X_{1,+,-}) - p_{q\leftarrow v_1, k\leftarrow c_1}^{(-,T_3)}(X_{1,+,-}) = \Theta(1)$$
$$p_{q\leftarrow -v_2, k\leftarrow c_1}^{(+,T_3)}(X_{1,+,-}) - p_{q\leftarrow -v_2, k\leftarrow c_1}^{(-,T_3)}(X_{1,+,-}) = \Theta(1)$$
$$p_{q\leftarrow c_1, k\leftarrow c_1}^{(+,T_3)}(X_{1,+,-}) - p_{q\leftarrow c_1, k\leftarrow c_1}^{(-,T_3)}(X_{1,+,-}) \geq 0$$

- *Near symmetry within a single attention head:*

$$\left| p_{q\leftarrow v_1, k\leftarrow v_1}^{(+,T_3)}(X_{1,+,-}) - p_{q\leftarrow v_1, k\leftarrow -v_2}^{(+,T_3)}(X_{1,+,-}) \right| \leq \widetilde{O}(\omega)$$
$$\left| p_{q\leftarrow -v_2, k\leftarrow v_1}^{(+,T_3)}(X_{1,+,-}) - p_{q\leftarrow -v_2, k\leftarrow -v_2}^{(+,T_3)}(X_{1,+,-}) \right| \leq \widetilde{O}(\omega)$$
$$\left| p_{q\leftarrow c_1, k\leftarrow v_1}^{(+,T_3)}(X_{1,+,-}) - p_{q\leftarrow c_1, k\leftarrow -v_2}^{(+,T_3)}(X_{1,+,-}) \right| \leq \widetilde{O}(\omega).$$

We now describe how the transformer can correctly classify the sample $X_{1,+,-}$. Based on Theorem 4.1, the output of the transformer $f^{(T_3)}(X_{1,+,-})$ can be analyzed as follows: by the attention's concentration on the group signal key, we have

$$w_+^{(T_3)\top} c_1 \left( p_{q\leftarrow v_1, k\leftarrow c_1}^{(+,T_3)}(X_{1,+,-}) + p_{q\leftarrow -v_2, k\leftarrow c_1}^{(+,T_3)}(X_{1,+,-}) + p_{q\leftarrow c_1, k\leftarrow c_1}^{(+,T_3)}(X_{1,+,-}) \right)$$
$$- w_-^{(T_3)\top} c_1 \left( p_{q\leftarrow v_1, k\leftarrow c_1}^{(-,T_3)}(X_{1,+,-}) + p_{q\leftarrow -v_2, k\leftarrow c_1}^{(-,T_3)}(X_{1,+,-}) + p_{q\leftarrow c_1, k\leftarrow c_1}^{(-,T_3)}(X_{1,+,-}) \right) \geq \Omega(1).$$

On the other hand, due to attention's near symmetry property, regardless of queries, we have $w_+^{(T_3)\top} v_1$ and $w_+^{(T_3)\top}(-v_2)$ approximately cancelling each other:

$$w_+^{(T_3)\top}(-v_2) \left( p_{q\leftarrow v_1, k\leftarrow -v_2}^{(+,T_3)}(X_{1,+,-}) + p_{q\leftarrow -v_2, k\leftarrow -v_2}^{(+,T_3)}(X_{1,+,-}) + p_{q\leftarrow c_1, k\leftarrow -v_2}^{(+,T_3)}(X_{1,+,-}) \right)$$
$$+ w_+^{(T_3)\top} v_1 \left( p_{q\leftarrow v_1, k\leftarrow v_1}^{(+,T_3)}(X_{1,+,-}) + p_{q\leftarrow -v_2, k\leftarrow v_1}^{(+,T_3)}(X_{1,+,-}) + p_{q\leftarrow c_1, k\leftarrow v_1}^{(+,T_3)}(X_{1,+,-}) \right) \approx 0.$$

Therefore, the positive head can output a value larger than the negative head and thus the sample can be correctly classified. Our results on the behavior of the softmax attention in Theorem 4.1 agrees with our experiment results in Figure 1c. By symmetry, the trained transformer will behave similarly on other samples with conflicting signs.

We now introduce another interpretation of Theorem 4.1 that relates to logistic linear regression. Notice that the transformer architecture in Equation (1) can be rewritten as

$$f(X) = [w_+^\top, w_-^\top, 1] \underbrace{\begin{bmatrix} \dot{\sigma}(H_+(X) + b) \sum_{l=1}^{L} X \mathrm{softmax}\left(X^\top W_{K+}^\top W_{Q+} x_l\right) \\ -\dot{\sigma}(H_-(X) + b) \sum_{l=1}^{L} X \mathrm{softmax}\left(X^\top W_{K-}^\top W_{Q-} x_l\right) \\ \dot{\sigma}(H_+(X) + b)b - \dot{\sigma}(H_-(X) + b)b \end{bmatrix}}_{\mathcal{F}(X)} \tag{2}$$

We define the vector $\mathcal{F}(X) \in \mathbb{R}^{2d+1}$ to be the **attention feature** of input $X$. Thus, if the transformer model can successfully learn the data distribution $\mathcal{D}$, we must have the set of attention features $\{\mathcal{F}(X)\}_{X \in \mathcal{D}}$ linearly separable. We show in Lemma 5.4 that the attention features under uniform attention is not linearly separable and we describe how the attention features evolve and become well-separated after training in Section 5.2.

Finally, for samples with consistent signs, although their attention features also change after stage 2, we are able to show that those changes do not affect the *separability* of their attention features.

## 5 PROOF OUTLINE

In order to analyze the training dynamics of the softmax attention, first of all, we need to keep track of the dynamics of the score variables $\mu^\top W_K^\top W_Q \nu$ for $\mu, \nu \in \mathcal{C} \cup \mathcal{V}$. However, upon writing down the gradient flow updates for $\mu^\top W_K^\top W_Q \nu$, it can be seen that their updates are governed by the variables in the form of $\mu^\top W_K^\top W_K \nu$ and $\mu^\top W_Q^\top W_Q \nu$ which we call **key and query self-score variables**. By keeping track of the gradient flow updates for those three types of variables all at once, we now have a complete dynamical system where all the terms appearing in the updates are tracked. The gradient flow dynamical system can be found in Appendix B.2.

Before we analyze the training dynamics, we observe one complication brought by random initialization: from our data distribution in Definition 2.1, the two mixtures are symmetric to each other and thus, ideally, the transformer's behavior should be the same for each class, e.g., $c_1^\top W_{K+}^{(t)\top} W_{Q+}^{(t)} v_1 = c_2^\top W_{K+}^{(t)\top} W_{Q+}^{(t)} v_2$. However, due to random initialization, $c_1^\top W_{K+}^{(0)\top} W_{Q+}^{(0)} v_1 \neq c_2^\top W_{K+}^{(0)\top} W_{Q+}^{(0)} v_2$. In order to solve the above complication, we consider the infinite-width transformer where the embedding dimension $m$ of the softmax attention goes to infinity. Within this model, we can derive nice symmetry properties between two groups and between two heads (Appendix C), that are convenient for our analysis.

**Definition 5.1** (Infinite-width transformer). *The infinite-width transformer is defined as the architecture in Equation* (1) *with* $m \to \infty$.

Our experiment results show that as long as the initialization scale $\omega$ is small enough, and the embedding width is large enough, the infinite-width transformer can indeed serve as an accurate model reflecting the true dynamics. For clarity, we present all of our results and analysis under the infinite-width transformer model. The finite-width transformer is a *discretized* version of the infinite-width transformer. Thus, the actual gradient flow dynamics can be seen as the infinite-width dynamics with perturbation introduced by finite-width discretization. We discuss how to take the discretization into account in Appendix G.

### 5.1 STAGE 1: NEURONS ALIGN WITH THE SIGNALS

In stage 1 of Algorithm 1, we train the neuron weights and keep the attention weights fixed at their random initialization. Since the neuron weights are zero-initialized, the outputs of the two attention heads are both zero at initialization. Recall that we fix the bias term $b$ in our transformer (Equation (1)) to be some small positive constant. Thus, there exists a period of time from initialization that the outputs of ReLU will be positive for *all* samples. We first characterize how the neuron weights align with the signals in such a case.

**Lemma 5.2** (Neuron alignment, stage 1.1). *Under Definition 5.1, for* $t \in [0, T_1]$, *if* $H_+^{(t)}(X), H_-^{(t)}(X) > -b$ *for all* $X \in \mathcal{D}$, *then for* $k \in \{1, 2\}$,

$$\frac{\mathrm{d}w_+^{(t)\top} v_k}{\mathrm{d}t} = \frac{1}{8}\left(\sum_{k' \in [2], s=s'} g^{(t)}(X_{k',s,s'})\right), \quad \frac{\mathrm{d}w_-^{(t)\top} v_k}{\mathrm{d}t} = -\frac{1}{8}\left(\sum_{k' \in [2], s=s'} g^{(t)}(X_{k',s,s'})\right).$$

*At the same time, for* $k \in \{1, 2\}$,

$$\frac{\mathrm{d}w_+^{(t)\top} c_k}{\mathrm{d}t} = 0, \quad \frac{\mathrm{d}w_-^{(t)\top} c_k}{\mathrm{d}t} = 0.$$

From the above lemma, we can see that when the output of ReLUs are positive for all samples, the neuron weights will align with the signed classification signals while not aligning with the group signals at all. A direct consequence is that the attention head output $H_+^{(t)}(X_{1,-,-})$ will keep decreasing until it becomes smaller than $-b$, as long as $b$ is not too large. Then the neuron weight $w_+$ is no longer able to receive gradient from sample $X_{1,-,-}$. We can show that when this happens, the neurons start to positively align with the group signals.

**Lemma 5.3** (Neuron alignment with group signals, stage 1.2). *Under Definition 5.1, for* $t \in [0, T_1]$, *if* $H_+^{(t)}(X_{k,-,-}) < -b$ *for* $k \in \{1, 2\}$, *then*

$$\frac{\mathrm{d}w_+^{(t)\top} c_k}{\mathrm{d}t} = \frac{1}{8}g^{(t)}(X_{k,+,+}).$$

At the same time, the neurons still keep aligning with the signed classification signals. Thus, if $T_1 = \Theta(1)$, then we have $w_+^{(t)\top} v_k = \Theta(1)$ and $w_+^{(t)\top} c_k = \Theta(1)$ after stage 1.

Although the neuron weights can learn the signals in stage 1, it is not hard to show that the transformer will output zero for samples with conflicting signs like $X_{1,+,-}$ and thus those samples are not correctly classified. In fact, we can show that under the averaged initial value model, our data distribution $\mathcal{D}$ is not learnable by merely training the neuron weights.

**Lemma 5.4** (Non-separability of uniform attention feature). *There exist no neuron weights $w_\pm$ such that $yf(X) > 0$ for all $X \in \{X_{k,s,-s}\}_{k \in [2], s \in \{\pm 1\}}$ when the softmax attention is uniform.*

Recall our definition of attention feature in Equation (2). Lemma 5.4 can be interpreted as that the uniform attention feature is not linearly separable. Due to randomness, the actual attention feature at initialization is not exactly uniform. However, by a simple continuity argument, Lemma 5.4 can be translated as the maximum margin of the initial attention feature is at most $\widetilde{O}(\omega^2/\sqrt{m})$. [1]

## 5.2 Stage 2: Attention learns from the samples with conflicting signs

Since by Lemma 5.4, the samples with conflicting signs are hard to learn by merely training the neuron weights, in stage 2, we directly train the softmax attention weights $W_{K\pm}, W_{Q\pm}$ on those hard samples. As we mentioned in the beginning of Section 5, in order to keep track of how the softmax attention changes during training, we need to analyze a dynamical system with all the score variables $\mu^\top W_{K+}^{(t)\top} W_{Q+}^{(t)} \nu$, key self-score variables $\mu^\top W_{K+}^{(t)\top} W_{K+}^{(t)} \nu$ and query self-score variables $\mu^\top W_{Q+}^{(t)\top} W_{Q+}^{(t)} \nu$. From the gradient flow update in Appendix B.2, observe that we can write the dynamical system in the following form:

$$
\frac{\mathrm{d}}{\mathrm{d}t}
\begin{bmatrix}
\mathrm{vec}(\{\mu^\top W_{K+}^{(t)\top} W_{Q+}^{(t)} \nu\}_{\mu,\nu \in \mathcal{C} \cup \mathcal{V}}) \\
\mathrm{vec}(\{\mu^\top W_{K+}^{(t)\top} W_{K+}^{(t)} \nu\}_{\mu,\nu \in \mathcal{C} \cup \mathcal{V}}) \\
\mathrm{vec}(\{\mu^\top W_{Q+}^{(t)\top} W_{Q+}^{(t)} \nu\}_{\mu,\nu \in \mathcal{C} \cup \mathcal{V}})
\end{bmatrix}
= A^{(t)}
\begin{bmatrix}
\mathrm{vec}(\{\mu^\top W_{K+}^{(t)\top} W_{Q+}^{(t)} \nu\}_{\mu,\nu \in \mathcal{C} \cup \mathcal{V}}) \\
\mathrm{vec}(\{\mu^\top W_{K+}^{(t)\top} W_{K+}^{(t)} \nu\}_{\mu,\nu \in \mathcal{C} \cup \mathcal{V}}) \\
\mathrm{vec}(\{\mu^\top W_{Q+}^{(t)\top} W_{Q+}^{(t)} \nu\}_{\mu,\nu \in \mathcal{C} \cup \mathcal{V}})
\end{bmatrix}
$$

where $A^{(t)}$ is a square matrix with dimension $|\{\mu^\top W_{K+}^{(t)\top} W_{Q+}^{(t)} \nu\}_{\mu,\nu}| + |\{\mu^\top W_{K+}^{(t)\top} W_{K+}^{(t)} \nu\}_{\mu,\nu}| + |\{\mu^\top W_{Q+}^{(t)\top} W_{Q+}^{(t)} \nu\}_{\mu,\nu}| = 16 + 10 + 10 = 36$. Thus, directly analyzing the dynamical system can easily become intractable. We need to come up with ways to utilize the structure of the problem to reduce the dimension we need to analyze.

One observation from Figure 1c is that the score variables $c_1^\top W_{K+}^{(t)\top} W_{Q+}^{(t)} v_1$ and $c_1^\top W_{K+}^{(t)\top} W_{Q+}^{(t)} v_2$ exhibit the maximum magnitude change compared with all other score variables. In fact, these two quantities play a key role in making $H_+(X_{1,+,-}) > H_-(X_{1,+,-})$. Thus, if we are able to prove that $c_1^\top W_{K+}^{(t)\top} W_{Q+}^{(t)} v_1$ will become sufficiently large at the end of stage 2, then we can make good progress on proving that the attention features become separable.

To make our presentation simpler, for a moment, assume we have the near symmetry properties:

$$
c_1^\top W_{K+}^{(t)\top} W_{Q+}^{(t)} v_1 \approx -c_1^\top W_{K+}^{(t)\top} W_{Q+}^{(t)} v_2
$$
$$
c_1^\top W_{K+}^{(t)\top} W_{K+}^{(t)} c_1 \approx -c_1^\top W_{K+}^{(t)\top} W_{K+}^{(t)} c_2
$$
$$
v_1^\top W_{Q+}^{(t)\top} W_{Q+}^{(t)} v_1 \approx -v_1^\top W_{Q+}^{(t)\top} W_{Q+}^{(t)} v_2
$$

which can indeed be made rigorous in Lemma E.4. Applying this near symmetry property to the gradient of $c_1^\top W_{K+}^{(t)\top} W_{Q+}^{(t)} v_1$ at $t = T_1$ (see Appendix E.5.1), we have

$$
\frac{\mathrm{d}c_1^\top W_{K+}^{(t)\top} W_{Q+}^{(t)} v_1}{\mathrm{d}t}\bigg|_{t=T_1} \approx \frac{1}{3}g^{(T_1)}4\left(v_1^\top W_{Q+}^{(T_1)\top} W_{Q+}^{(T_1)} v_1\right)\left(\frac{2}{3}w_+^{(T_1)\top} c_1\right)
$$
$$
+ \frac{1}{3}g^{(T_1)}4\left(c_1^\top W_{K+}^{(T_1)\top} W_{K+}^{(T_1)} c_1\right)\left(\frac{2}{3}w_+^{(T_1)\top} c_1\right).
$$

---

[1] Recall that the maximum margin of a set $\mathcal{S}$ is defined as $\gamma := \max_{\|w\|_2=1} \min_{x \in \mathcal{S}} w^\top x$.

Due to symmetry and cancellation at initialization, it only depends on 2 self-score variables: $c_1^\top W_{K+}^{(t)\top} W_{K+}^{(t)} c_1$ and $v_1^\top W_{Q+}^{(t)\top} W_{Q+}^{(t)} v_1$. At the same time, the gradient of $c_1^\top W_{K+}^{(t)\top} W_{K+}^{(t)} c_1$ and $v_1^\top W_{Q+}^{(t)\top} W_{Q+}^{(t)} v_1$ at $t = T_1$ also depends only on $c_1^\top W_{K+}^{(t)\top} W_{Q+}^{(t)} v_1$. Thus, if this simpler dependence can hold through the rest of stage 2, then we get a much smaller $3 \times 3$ system to analyze. Unfortunately, this is not true. Take the gradient of $c_1^\top W_{K+}^{(t)\top} W_{Q+}^{(t)} v_1$ as an example. Training the attention weights can break the symmetry that holds at $T_1$, and the gradient of $c_1^\top W_{K+}^{(t)\top} W_{Q+}^{(t)} v_1$ will start to depend on other self-score variables. The way to overcome this issue is to realize that as long as the maximum change in the output of the softmax is not too large, we can bound the effects from other self-score variables and the gradient of $c_1^\top W_{K+}^{(t)\top} W_{Q+}^{(t)} v_1$ is still mainly contributed by $c_1^\top W_{K+}^{(t)\top} W_{K+}^{(t)} c_1$ and $v_1^\top W_{Q+}^{(t)\top} W_{Q+}^{(t)} v_1$.

Now, we have successfully reduce the dimension of the system that we need to analyze to $3 \times 3$. However, there is still a problem we need to handle: the system has a matrix with *non-linear* dependence on those 3 variables. Generally, it is impossible to derive a close-form solution. Fortunately, we can show that (1) those 3 variables are increasing throughout the training and (2) when the change of the softmax attention (from initialization) is within a certain range, the growth of those 3 variables can be lower bounded by a linear dynamical system.

**Lemma 5.5** (Informal version of Lemma E.2). *For $t \geq T_1$, when all the score variables are bounded within some range, there exists a constant $b > 0$ such that*

$$\frac{\mathrm{d}}{\mathrm{d}t} \begin{bmatrix} c_1^\top W_{K+}^{(t)\top} W_{Q+}^{(t)} v_1 \\ c_1^\top W_{K+}^{(t)\top} W_{K+}^{(t)} c_1 \\ v_1^\top W_{Q+}^{(t)\top} W_{Q+}^{(t)} v_1 \end{bmatrix} \geq b \begin{bmatrix} 0 & 1 & 1 \\ 1 & 0 & 0 \\ 1 & 0 & 0 \end{bmatrix} \begin{bmatrix} c_1^\top W_{K+}^{(t)\top} W_{Q+}^{(t)} v_1 \\ c_1^\top W_{K+}^{(t)\top} W_{K+}^{(t)} c_1 \\ v_1^\top W_{Q+}^{(t)\top} W_{Q+}^{(t)} v_1 \end{bmatrix}.$$

With Lemma 5.5, we can prove that the score $c_1^\top W_{K+}^{(t)\top} W_{Q+}^{(t)} v_1$ will become sufficiently large at the end of stage 2.

On the other hand, when the query token is the group signal, we can prove

$$v_1^\top W_{K+}^{(t)\top} W_{Q+}^{(t)} c_1 < 0, \quad v_2^\top W_{K+}^{(t)\top} W_{Q+}^{(t)} c_1 > 0.$$

By symmetry, this implies $p_{q \leftarrow c_1, k \leftarrow c_1}^{(+,t)}(X_{1,+,-}) > p_{q \leftarrow c_1, k \leftarrow c_1}^{(-,t)}(X_{1,+,-})$. This finishes the proof of the trained attention's property on the concentration on the group signal token in Theorem 4.1.

Finally, when $v_1$ and $v_2$ are the keys, we can prove the near symmetry property between scores:

$$v_1^\top W_{K+}^{(t)\top} W_{Q+} v_1 \approx -v_2^\top W_{K+}^{(t)\top} W_{Q+} v_1,$$
$$v_1^\top W_{K+}^{(t)\top} W_{Q+} v_2 \approx -v_2^\top W_{K+}^{(t)\top} W_{Q+} v_2$$
$$v_1^\top W_{K+}^{(t)\top} W_{Q+} c_1 \approx -v_2^\top W_{K+}^{(t)\top} W_{Q+} c_1$$

which can be translated to the near symmetry property within a single head in Theorem 4.1.

## 5.3 STAGE 3: CONVERGENCE

In stage 3, we train the neuron weights again. At the end of stage 2, the attention features for samples with conflicting signs becomes well-separated. In addition, we can show that the attention features for the samples with consistent signs are still well-separated. Thus, the training in stage 3 behaves like logistic linear regression on linearly separable data and training the neuron weight can drive the population loss towards zero. This proves the part in Theorem 4.1 that the population loss can become sufficiently small after training.

## 6 GENERAL NUMBER OF MIXTURES

In this section, we study the case when the mixtures have more than two groups. Our experiment results can be found in Appendix A.4. We summarize our experimental findings below.

**Observation 1.** *The transformer architecture in Equation* (1) *can learn the data distribution for* $K \geq 3$. *However, the near symmetry property in the attention no longer holds. In addition, the training takes longer with more groups.*

On the other hand, we observe that for $K = 3$, the training dynamics is still stage-wise and there are score variables exhibiting major growth compared with others. Thus, the techniques developed in our work can be used to study the training dynamics for $K \geq 3$.

Although the training dynamics for $K \geq 3$ has some differences from the $K = 2$ case, we are able to provide a construction of transformers that can solve any arbitrary $K$ mixtures.

**Lemma 6.1.** *There exists a transformer architecture solving $K$-mixture of linear classification.*

*Proof.* The construction is given as follows. Let

$$
W_{K+}^\top W_{Q+} = 2C \cdot \sum_{k=1}^{K} v_k c_k^\top + C \cdot \sum_{k=1}^{K} c_k c_k^\top + C \cdot \sum_{k=1}^{K} v_k v_k^\top,
$$

$$
W_{K-}^\top W_{Q-} = -2C \cdot \sum_{k=1}^{K} v_k c_k^\top + C \cdot \sum_{k=1}^{K} c_k c_k^\top + C \cdot \sum_{k=1}^{K} v_k v_k^\top
$$

The transformer is given as

$$
f(X) = \sum_{l_1} \sum_{k=1}^{K} v_k^\top X \mathrm{softmax}(X^\top W_{K+}^\top W_{Q+} X_{l_1}) + \sum_{l_1} \sum_{k=1}^{K} -v_k^\top X \mathrm{softmax}(X^\top W_{K-}^\top W_{Q-} X_{l_1})
$$

Let $C \to \infty$. First of all, consider classification signal query $v_k$. Then,

$$
\sum_{k=1}^{K} v_k^\top X \mathrm{softmax}(X^\top W_{K+}^\top W_{Q+} v_k) + \sum_{k=1}^{K} -v_k^\top X \mathrm{softmax}(X^\top W_{K+}^\top W_{Q+} v_k)
$$
$$
= v_k^\top v_k - v_k^\top v_k = 0
$$

Similar analysis for query $-v_k$. On the other hand, for group signal query $c_k$, suppose the corresponding classification signal in $X$ is $v_k$, we have

$$
\sum_{k=1}^{K} v_k^\top X \mathrm{softmax}(X^\top W_{K+}^\top W_{Q+} c_k) + \sum_{k=1}^{K} -v_k^\top X \mathrm{softmax}(X^\top W_{K+}^\top W_{Q+} c_k)
$$
$$
= v_k^\top v_k - v_k^\top c_k = 1
$$

Similar analysis if the corresponding classification signal is $-v_k$. Thus, we have $yf(X) > 0$ for all $(X, y) \in \mathcal{D}$. □

## 7 DISCUSSION

In this work, we studied the training dynamics of a two-headed transformer solving two-mixture of linear classification. We characterized the feature learning process of how the softmax attention utilizes the relationship between the class signals and their corresponding classification signals in this task. One open problem is that the current proof techniques critically utilize the fact that both the transformer architecture and the data distribution are symmetric. Can we discover some hidden structures of the problem such that we can analyze the training dynamics for $K \geq 3$? More interestingly, we demonstrated that during training, the attention features evolved from a near non-separable state to a well-separated state, i.e, the maximum margin of the attention features increases during training. What properties does the data distribution need to have such that the maximum margin can increase in training? We leave those interesting questions as future works.

ACKNOWLEDGMENT

HY would like to thank Alex Damian for extensive discussion and suggestions. The work of ZW was in part supported by an NSF Scale-MoDL grant (award number: 2133861) and the CAREER Award (award number: 2145346). JDL acknowledges support of Open Philanthropy, NSF IIS 2107304, NSF CCF 2212262, NSF CAREER Award 2144994, and NSF CCF 2019844. The work of YL was supported in part by the U.S. National Science Foundation with the grants ECCS-2113860 and DMS-2134145.

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

CONTENTS

# A   ADDITIONAL EXPERIMENT RESULTS

## A.1   SOFTMAX ATTENTION

We provide the softmax probability of the positive head on different samples.

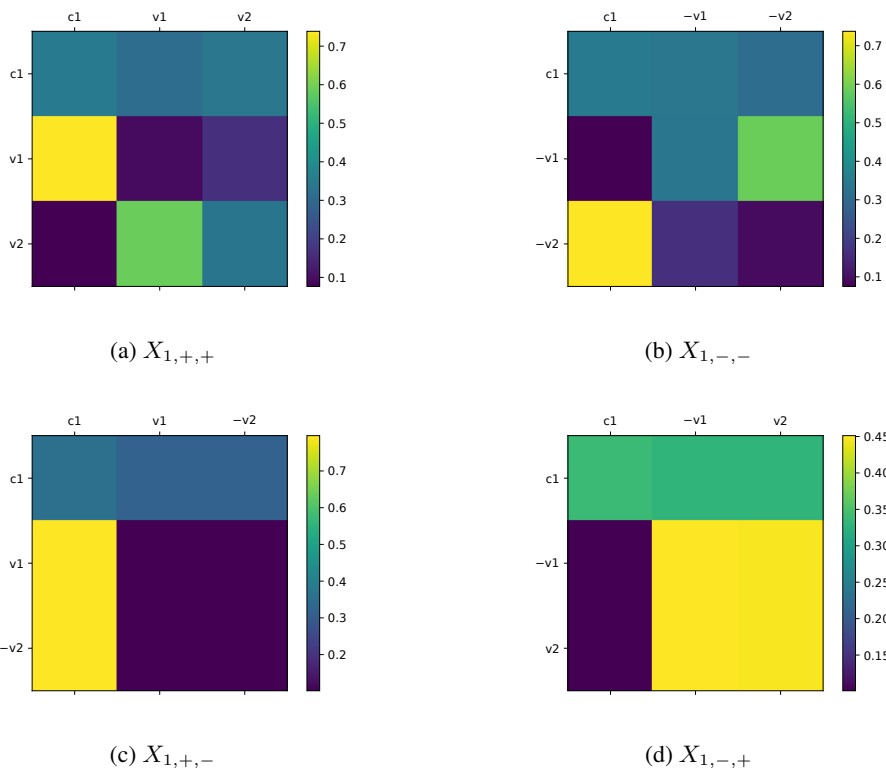

(a) $X_{1,+,+}$     (b) $X_{1,-,-}$

(c) $X_{1,+,-}$     (d) $X_{1,-,+}$

Figure 2: Softmax attention of the positive head on different samples. The vertical axis denotes the query tokens and the horizontal axis denotes the key tokens.

## A.2   ONLY TRAIN THE ATTENTION

Recall that in Section 3, we observed there is a learning order when all the weights are trained simultaneously: the attention weights learn after the neuron weights. To further study whether this learning order is necessary, we conduct another experiment where we initialize the neurons as Gaussian and only train the attention. Our result suggests that in this case, the transformer is no longer trainable.

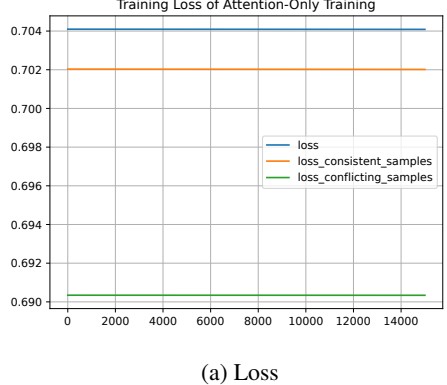

(a) Loss

Figure 3: The loss for training only the attention while keeping the neurons fixed at initialization.

## A.3 FULL ATTENTION SCORES OF TWO MIXTURES

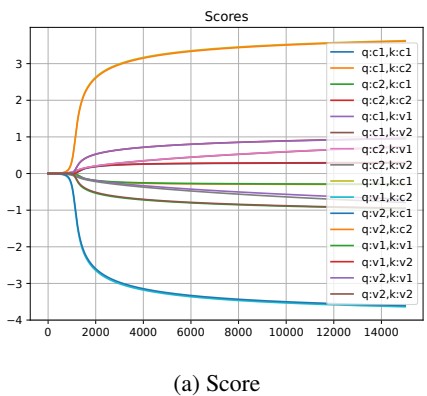

(a) Score

Figure 4: Full attention scores for $K = 2$.

## A.4 EXPERIMENTS ON MULTI-MIXTURE MODELS

In this section, we provide experiment results on synthetic data for multi-mixture models. Specifically, Figure 5, Figure 6 and Figure 7 respectively indicate that the training dynamics of 3-mixture, 4-mixture and 8-mixture models are all showing some similarities as that for 2-mixture model, demonstrating the broad applicability of our developed theoretical results.

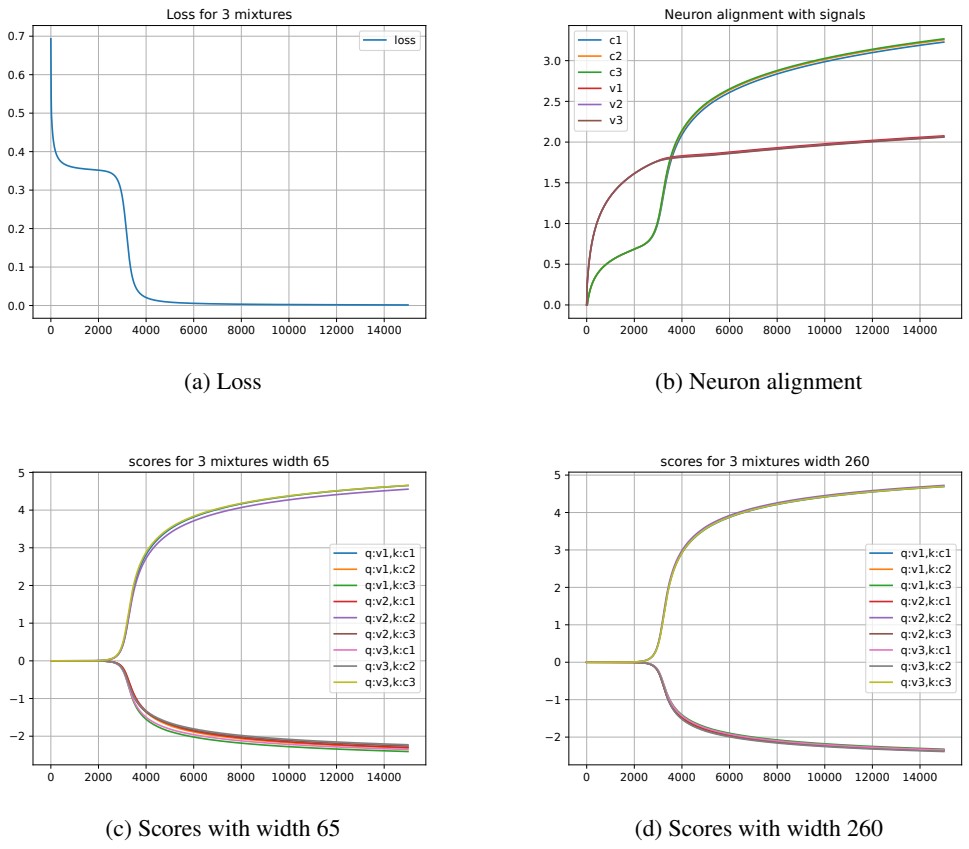

(a) Loss

(b) Neuron alignment

(c) Scores with width 65

(d) Scores with width 260

Figure 5: The experiment results for the case with $K = 3$ mixtures.

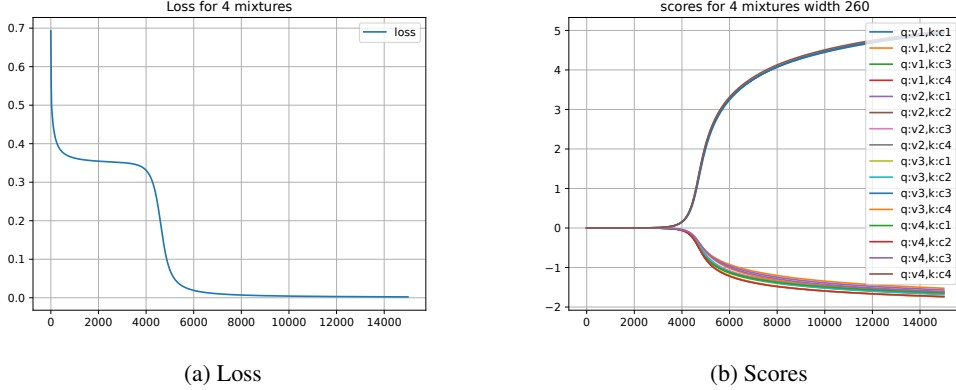

Figure 6: The experiment results for the case with $K = 4$ mixtures.

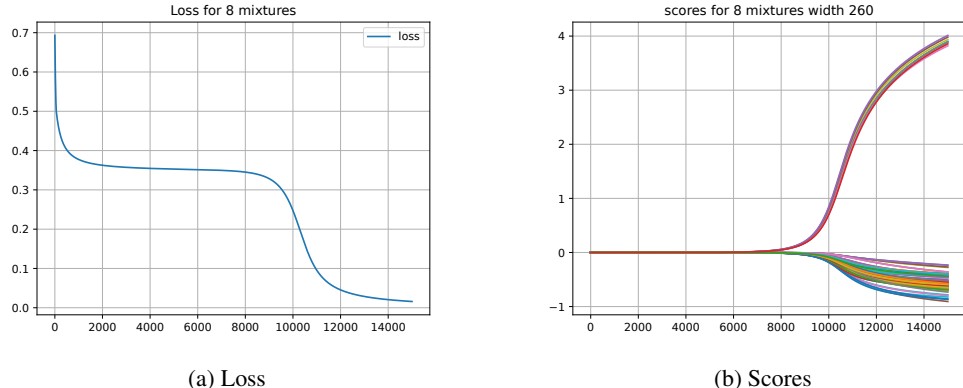

Figure 7: The experiment results for the case with $K = 8$ mixtures.

## A.5 REAL WORLD EXPERIMENT

In this section, we present our real world experiment results. In our experiments, we use MNIST dataset and extract the images with label 1 and label 2 to play the role of classification signals. We create two random vectors to play the role of group signals. We plot the training loss curves for both simultaneously training all the weights and separately training the neuron weights and attention weights in Figure 8. In separate weight training, the three stages are set to 100 steps each.

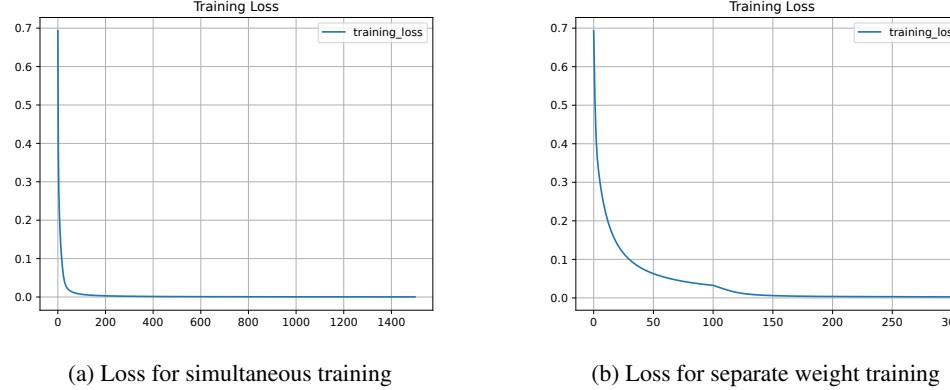

(a) Loss for simultaneous training    (b) Loss for separate weight training

Figure 8: The experiment results for simultaneous weight training and separate weight training.

# B   GRADIENT FLOW

## B.1   GRADIENTS OF EACH WEIGHT MATRICES

$$\frac{dw_+^{(t)}}{dt} = \mathbb{E}\left[g^{(t)}(X)y\dot{\sigma}_+^{(t)}\sum_{l=1}^{L}Xp_l^{(+,t)}\right]$$

$$\frac{dw_-^{(t)}}{dt} = -\mathbb{E}\left[g^{(t)}(X)y\dot{\sigma}_-^{(t)}\sum_{l=1}^{L}Xp_l^{(+,t)}\right]$$

$$\frac{dW_{K+}^{(t)}}{dt} = \mathbb{E}\left[g^{(t)}(X)y\dot{\sigma}_+^{(t)}\sum_{l=1}^{L}\sum_{h=1}^{L}w_+^\top X_h\sum_{h'=1}^{L}\frac{\partial p_{l,h}^{(+,t)}}{\partial s_{l,h'}^{(+)}}q_{+l}^{(t)}x_{h'}^\top\right]$$

$$= \mathbb{E}\left[g^{(t)}(X)y\dot{\sigma}_+^{(t)}\sum_{l=1}^{L}\sum_{h=1}^{L}w_+^\top X_h\sum_{h'=1}^{L}p_{l,h}^{(+,t)}(\mathbb{I}(h=h')-p_{l,h'}^{(+,t)})q_{+l}^{(t)}x_{h'}^\top\right]$$

$$= \mathbb{E}\left[g^{(t)}(X)y\dot{\sigma}_+^{(t)}\sum_{l=1}^{L}\left(-w_+^\top Xp_l^{(+,t)}q_{+l}^{(t)}(Xp_l^{(+,t)})^\top + q_{+l}^{(t)}w_+^\top X\mathrm{diag}(p_l^{(+,t)})X^\top\right)\right]$$

$$= \mathbb{E}\left[g^{(t)}(X)y\dot{\sigma}_+^{(t)}\sum_{l=1}^{L}q_{+l}^{(t)}\left(-w_+^\top Xp_l^{(+,t)}p_l^{(+,t)\top} + w_+^\top X\mathrm{diag}(p_l^{(+,t)})\right)X^\top\right]$$

$$= \mathbb{E}\left[g^{(t)}(X)y\dot{\sigma}_+^{(t)}\sum_{l=1}^{L}q_{+l}^{(t)}p_l^{(+,t)\top}\mathrm{diag}\left(w_+^\top X - w_+^\top Xp_l^{(+,t)}\right)X^\top\right]$$

$$\frac{dW_{Q+}^{(t)}}{dt} = \mathbb{E}\left[g^{(t)}(X)y\dot{\sigma}_+^{(t)}\sum_{l=1}^{L}w_+^\top\sum_{h=1}^{L}X_h\sum_{h'=1}^{L}p_{l,h}^{(+,t)}(\mathbb{I}(h=h')-p_{l,h'}^{(+,t)})k_{+h'}^{(t)}x_l^{(i)\top}\right]$$

$$= \mathbb{E}\left[g^{(t)}(X)y\dot{\sigma}_+^{(t)}\sum_{l=1}^{L}\left(-w_+^\top Xp_l^{(+,t)}(K_+^{(t)}p_l^{(+,t)}) + K_+^{(t)}\mathrm{diag}(p_l^{(+,t)})X^\top w_+\right)x_l^\top\right]$$

$$= \mathbb{E}\left[g^{(t)}(X)y\dot{\sigma}_+^{(t)}\sum_{l=1}^{L}K_+^{(t)}\left(-w_+^\top Xp_l^{(+,t)}p_l^{(+,t)} + \mathrm{diag}(p_l^{(+,t)})X^\top w_+\right)x_l^\top\right]$$

$$= \mathbb{E}\left[g^{(t)}(X)y\dot{\sigma}_+^{(t)}\sum_{l=1}^{L}K_+^{(t)}\mathrm{diag}\left(X^\top w_+ - w_+^\top Xp_l^{(+,t)}\right)p_l^{(+,t)}x_l^\top\right]$$

## B.2 GRADIENT FLOW DYNAMICAL SYSTEM

Let $\mu, \nu \in \mathcal{V} \cup \mathcal{C}$. We have

$$\frac{\mathrm{d}w_+^{(t)\top}\mu}{\mathrm{d}t} = \mathbb{E}\left[g^{(t)}(X)y\dot{\sigma}_+^{(t)}\sum_{l=1}^{L}\left(Xp_l^{(+,t)}\right)^{\top}\right]\mu$$

$$\frac{\mathrm{d}\nu^{\top}W_{K+}^{(t)\top}W_{Q+}^{(t)}\mu}{\mathrm{d}t} = \mathbb{E}\left[g^{(t)}(X)y\dot{\sigma}_+^{(t)}\sum_{l=1}^{L}\mu^{\top}W_{Q+}^{(t)\top}q_{+l}^{(t)}p_l^{(+,t)\top}\mathrm{diag}\left(w_+^{(t)\top}X - w_+^{(t)\top}Xp_l^{(+,t)}\right)X^{\top}\nu\right]$$

$$+ \mathbb{E}\left[g^{(t)}(X)y\dot{\sigma}_+^{(t)}\sum_{l=1}^{L}\nu^{\top}W_{K+}^{(t)\top}K_+^{(t)}\mathrm{diag}\left(X^{\top}w_+ - w_+^{\top}Xp_l^{(+,t)}\right)p_l^{(+,t)}x_l^{\top}\mu\right]$$

$$\frac{\mathrm{d}\nu^{\top}W_{K+}^{(t)\top}W_{K+}^{(t)}\mu}{\mathrm{d}t} = \mathbb{E}\left[g^{(t)}(X)y\dot{\sigma}_+^{(t)}\sum_{l=1}^{L}\nu^{\top}W_{K+}^{(t)\top}q_{+l}^{(t)}p_l^{(+,t)\top}\mathrm{diag}\left(w_+^{\top}X - w_+^{\top}Xp_l^{(+,t)}\right)X^{\top}\mu\right]$$

$$+ \mathbb{E}\left[g^{(t)}(X)y\dot{\sigma}_+^{(t)}\sum_{l=1}^{L}\mu^{\top}W_{K+}^{(t)\top}q_{+l}^{(t)}p_l^{(+,t)\top}\mathrm{diag}\left(w_+^{\top}X - w_+^{\top}Xp_l^{(+,t)}\right)X^{\top}\nu\right]$$

$$\frac{\mathrm{d}\nu^{\top}W_{Q+}^{(t)\top}W_{Q+}^{(t)}\mu}{\mathrm{d}t} = \mathbb{E}\left[g^{(t)}(X)y\dot{\sigma}_+^{(t)}\sum_{l=1}^{L}\nu^{\top}W_{Q+}^{(t)\top}K_+^{(t)}\mathrm{diag}\left(X^{\top}w_+ - w_+^{\top}Xp_l^{(+,t)}\right)p_l^{(+,t)}x_l^{\top}\mu\right]$$

$$+ \mathbb{E}\left[g^{(t)}(X)y\dot{\sigma}_+^{(t)}\sum_{l=1}^{L}\mu^{\top}W_{Q+}^{(t)\top}K_+^{(t)}\mathrm{diag}\left(X^{\top}w_+ - w_+^{\top}Xp_l^{(+,t)}\right)p_l^{(+,t)}x_l^{\top}\nu\right]$$

## C SYMMETRY

In this section, we derive symmetry results for the infinite-width transformer model. We first characterize the initial values of the softmax attention weights:

**Lemma C.1.** *The initial values of the attention weight in the infinite-width transformer satisfy:*

$$\nu^{\top}W_{K\pm}^{(0)\top}W_{Q\pm}^{(0)}\mu = \mathbb{E}_{\widehat{W}_K,\widehat{W}_Q}\left[\nu^{\top}\widehat{W}_K^{\top}\widehat{W}_Q\mu\right] = 0$$

$$\nu^{\top}W_{K\pm}^{(0)\top}W_{K\pm}^{(0)}\mu = \begin{cases} \mathbb{E}_{\widehat{W}_K}\left[\nu^{\top}\widehat{W}_K^{\top}\widehat{W}_K\mu\right] = 0, & \nu \neq \mu \\ \mathbb{E}_{\widehat{W}_K}\left[\mu^{\top}\widehat{W}_K^{\top}\widehat{W}_K\mu\right] = \omega^2, & \nu = \mu \end{cases}$$

$$\nu^{\top}W_{Q\pm}^{(0)\top}W_{Q\pm}^{(0)}\mu = \begin{cases} \mathbb{E}_{\widehat{W}_Q}\left[\nu^{\top}\widehat{W}_Q^{\top}\widehat{W}_Q\mu\right] = 0, & \nu \neq \mu \\ \mathbb{E}_{\widehat{W}_Q}\left[\mu^{\top}\widehat{W}_Q^{\top}\widehat{W}_Q\mu\right] = \omega^2, & \nu = \mu \end{cases}$$

*where $\widehat{W}_K, \widehat{W}_Q \in \mathbb{R}^{m \times d}$ and $(\widehat{W}_K)_{ij}, (\widehat{W}_Q)_{ij} \sim \mathcal{N}(0, \omega^2\frac{1}{m})$.*

*Proof.* Let $(W_{K+})_i, (W_{Q+})_i$ denote the i-th row of $W_{K+}, W_{Q+}$, respectively. Then, we have $\nu^{\top}W_{K\pm}^{(0)\top}W_{Q\pm}^{(0)}\mu = \sum_{i=1}^{m}(W_{K+}^{(0)})_i^{\top}\nu(W_{Q+}^{(0)})_i^{\top}\mu$. Notice that $(W_{K+}^{(0)})_i^{\top}\nu \sim \mathcal{N}(0, \omega^2\frac{1}{m})$, $(W_{Q+}^{(0)})_i^{\top}\mu \sim \mathcal{N}(0, \omega^2\frac{1}{m})$. Thus, $\nu^{\top}W_{K\pm}^{(0)\top}W_{Q\pm}^{(0)}\mu$ has the same distribution as $\frac{1}{m}\sum_{i=1}^{m}a_ib_i$ where $a_i, b_i \overset{i.i.d.}{\sim} \mathcal{N}(0, \omega^2)$. By the Law of Large Number,

$$\lim_{m\to\infty}\nu^{\top}W_{K\pm}^{(0)\top}W_{Q\pm}^{(0)}\mu = \lim_{m\to\infty}\frac{1}{m}\sum_{i=1}^{m}a_ib_i = \mathbb{E}[a_1b_1] = \mathbb{E}[a_1]\mathbb{E}[b_1] = 0.$$

The rest can be proved similarly with an additional fact that for $w \sim \mathcal{N}(0, I)$, $w^{\top}\mu$ is independent of $w^{\top}\nu$ if $\mu \perp \nu$. $\qquad\square$

The symmetry results contain two parts: symmetry between two groups and symmetry between two attention heads.

**Lemma C.2** (Symmetry between two groups). *Under the infinite-width transformer model in Definition 5.1, let $\mu_1, \nu_1 \in \{c_1, v_1\}$ and $\mu_2, \nu_2 \in \{c_2, v_2\}$ such that $\mu_1, \mu_2$ are the same type signals (i.e., either $\mu_1, \mu_2 \in \mathcal{C}$ or $\mu_1, \mu_2 \in \mathcal{V}$) and $\nu_1, \nu_2$ are the same type signals. For all $t$,*

$$w_+^{(t)\top} \mu_1 = w_+^{(t)\top} \mu_2 \tag{3}$$

$$\nu_1^\top W_{K+}^{(t)\top} W_{Q+}^{(t)} \mu_1 = \nu_2^\top W_{K+}^{(t)\top} W_{Q+}^{(t)} \mu_2, \quad \nu_2^\top W_{K+}^{(t)\top} W_{Q+}^{(t)} \mu_1 = \nu_1^\top W_{K+}^{(t)\top} W_{Q+}^{(t)} \mu_2 \tag{4}$$

$$\nu_1^\top W_{K+}^{(t)\top} W_{K+}^{(t)} \mu_1 = \nu_2^\top W_{K+}^{(t)\top} W_{K+}^{(t)} \mu_2, \quad \nu_1^\top W_{K+}^{(t)\top} W_{K+}^{(t)} \mu_2 = \nu_2^\top W_{K+}^{(t)\top} W_{K+}^{(t)} \mu_1 \tag{5}$$

$$\nu_1^\top W_{Q+}^{(t)\top} W_{Q+}^{(t)} \mu_1 = \nu_2^\top W_{Q+}^{(t)\top} W_{Q+}^{(t)} \mu_2, \quad \nu_1^\top W_{Q+}^{(t)\top} W_{Q+}^{(t)} \mu_2 = \nu_2^\top W_{Q+}^{(t)\top} W_{Q+}^{(t)} \mu_1 \tag{6}$$

*The symmetry between the two groups also holds for the negative head.*

**Corollary C.3.** *Let $s, s' \in \{\pm 1\}$. For all $t$,*

$$p^{(+,t)}(X_{1,s,s'}) = p^{(+,t)}(X_{2,s,s'}), \quad p^{(-,t)}(X_{1,s,s'}) = p^{(-,t)}(X_{2,s,s'})$$

$$H_+^{(t)}(X_{1,s,s'}) = H_+^{(t)}(X_{2,s,s'}), \quad H_-^{(t)}(X_{1,s,s'}) = H_-^{(t)}(X_{2,s,s'})$$

$$f^{(t)}(X_{1,s,s'}) = f^{(t)}(X_{2,s,s'})$$

*Proof of Lemma C.2.* First of all, by Definition 5.1, all the equalities hold at $t = 0$. Thus, we only need to show the gradient flow updates satisfy all the equalities if all the equalities hold at time $t$.

Assume all the equalities hold at time $t$. This implies that $p^{(+,t)}(X_{1,s,s'}) = p^{(+,t)}(X_{2,s,s'})$ (recall that since our model doesn't have any positional encoding, we can consider ordered inputs in our analysis), which further implies that $H_+^{(t)}(X_{1,s,s'}) = H_+^{(t)}(X_{2,s,s'})$, $H_-^{(t)}(X_{1,s,s'}) = H_-^{(t)}(X_{2,s,s'})$ and $f^{(t)}(X_{1,s,s'}) = f^{(t)}(X_{2,s,s'})$ for all $s, s' \in \{\pm 1\}$.

First, we prove Equation (3):

$$\frac{\mathrm{d}w_+^{(t)\top} \mu_1}{\mathrm{d}t} = \mathbb{E}\left[g^{(t)} y \dot{\sigma}_+^{(t)} \sum_{l=1}^{L} \left(X p_l^{(+,t)}\right)^\top\right] \mu_1$$

$$= \mathbb{E}_{s,s'}\left[\frac{1}{2} g^{(t)}(X_{1,s,s'}) s \dot{\sigma}_+^{(t)}(X_{1,s,s'}) \sum_{l=1}^{L} \left(X_{1,s,s'} p_l^{(+,t)}(X_{1,s,s'})\right)^\top \mu_1\right.$$

$$\left. + \frac{1}{2} g^{(t)}(X_{2,s,s'}) s \dot{\sigma}_+^{(t)}(X_{2,s,s'}) \sum_{l=1}^{L} \left(X_{2,s,s'} p_l^{(+,t)}(X_{2,s,s'})\right)^\top \mu_1\right]$$

$$= \mathbb{E}_{s,s'}\left[\frac{1}{2} g^{(t)}(X_{2,s,s'}) s \dot{\sigma}_+^{(t)}(X_{2,s,s'}) \sum_{l=1}^{L} \left(X_{2,s,s'} p_l^{(+,t)}(X_{2,s,s'})\right)^\top \mu_2\right.$$

$$\left. + \frac{1}{2} g^{(t)}(X_{1,s,s'}) s \dot{\sigma}_+^{(t)}(X_{1,s,s'}) \sum_{l=1}^{L} \left(X_{1,s,s'} p_l^{(+,t)}(X_{1,s,s'})\right)^\top \mu_2\right]$$

$$\text{(by } p^{(+,t)}(X_{1,s,s'}) = p^{(+,t)}(X_{2,s,s'}))$$

$$= \mathbb{E}\left[g^{(t)} y \dot{\sigma}_+^{(t)} \sum_{l=1}^{L} \left(X p_l^{(+,t)}\right)^\top\right] \mu_2 = \frac{\mathrm{d}w_+^{(t)\top} \mu_2}{\mathrm{d}t}$$

Next, we prove the first equality in Equation (4):

$$\frac{\mathrm{d}\nu_1^\top W_{K+}^{(t)\top} W_{Q+}^{(t)} \mu_1}{\mathrm{d}t}$$

$$= \frac{1}{2} \mathbb{E}_{s,s'}\left[g^{(t)} s \dot{\sigma}_+^{(t)} \sum_{l=1}^{L} \mu_1^\top W_{Q+}^{(t)\top} q_{+l}^{(t)} p_l^{(+,t)\top} \mathrm{diag}\left(w_+^{(t)\top} X_{1,s,s'} - w_+^{(t)\top} X_{1,s,s'} p_l^{(+,t)}\right) X_{1,s,s'}^\top \nu_1\right]$$

$$+ \frac{1}{2}\mathbb{E}_{s,s'}\left[g^{(t)}s\dot\sigma_+^{(t)}\sum_{l=1}^L \mu_1^\top W_{Q+}^{(t)\top} q_{+l}^{(t)} p_l^{(+,t)\top}\mathrm{diag}\left(w_+^{(t)\top}X_{2,s,s'} - w_+^{(t)\top}X_{2,s,s'}p_l^{(+,t)}\right)X_{2,s,s'}^\top \nu_1\right]$$

$$+ \frac{1}{2}\mathbb{E}_{s,s'}\left[g^{(t)}s\dot\sigma_+^{(t)}\sum_{l=1}^L \nu_1^\top W_{K+}^{(t)\top} K_+^{(t)}\mathrm{diag}\left(X_{1,s,s'}^\top w_+ - w_+^\top X_{1,s,s'}p_l^{(+,t)}\right)p_l^{(+,t)}(X_{1,s,s'})_l^\top \mu_1\right]$$

$$+ \frac{1}{2}\mathbb{E}_{s,s'}\left[g^{(t)}s\dot\sigma_+^{(t)}\sum_{l=1}^L \nu_1^\top W_{K+}^{(t)\top} K_+^{(t)}\mathrm{diag}\left(X_{2,s,s'}^\top w_+ - w_+^\top X_{2,s,s'}p_l^{(+,t)}\right)p_l^{(+,t)}(X_{2,s,s'})_l^\top \mu_1\right]$$

$$= \frac{1}{2}\mathbb{E}_{s,s'}\left[g^{(t)}s\dot\sigma_+^{(t)}\sum_{l=1}^L \mu_2^\top W_{Q+}^{(t)\top} q_{+l}^{(t)} p_l^{(+,t)\top}\mathrm{diag}\left(w_+^{(t)\top}X_{2,s,s'} - w_+^{(t)\top}X_{2,s,s'}p_l^{(+,t)}\right)X_{2,s,s'}^\top \nu_2\right]$$

$$+ \frac{1}{2}\mathbb{E}_{s,s'}\left[g^{(t)}s\dot\sigma_+^{(t)}\sum_{l=1}^L \mu_2^\top W_{Q+}^{(t)\top} q_{+l}^{(t)} p_l^{(+,t)\top}\mathrm{diag}\left(w_+^{(t)\top}X_{1,s,s'} - w_+^{(t)\top}X_{1,s,s'}p_l^{(+,t)}\right)X_{1,s,s'}^\top \nu_2\right]$$

$$+ \frac{1}{2}\mathbb{E}_{s,s'}\left[g^{(t)}s\dot\sigma_+^{(t)}\sum_{l=1}^L \nu_2^\top W_{K+}^{(t)\top} K_+^{(t)}\mathrm{diag}\left(X_{2,s,s'}^\top w_+ - w_+^\top X_{2,s,s'}p_l^{(+,t)}\right)p_l^{(+,t)}(X_{2,s,s'})_l^\top \mu_2\right]$$

$$+ \frac{1}{2}\mathbb{E}_{s,s'}\left[g^{(t)}s\dot\sigma_+^{(t)}\sum_{l=1}^L \nu_2^\top W_{K+}^{(t)\top} K_+^{(t)}\mathrm{diag}\left(X_{1,s,s'}^\top w_+ - w_+^\top X_{1,s,s'}p_l^{(+,t)}\right)p_l^{(+,t)}(X_{1,s,s'})_l^\top \mu_2\right]$$

$$= \frac{\mathrm{d}\nu_2^\top W_{K+}^{(t)\top} W_{Q+}^{(t)}\mu_2}{\mathrm{d}t}$$

The rest equalities in Equation (4), Equation (5) and Equation (6) can be proved similarly. □

**Symmetry between two heads:** notice that for the negative head, we can view it as training the model $-f(X)$ on the dataset where the label is $-y$ and the set of classification signals is $\{-v_1, -v_2\}$.

**Lemma C.4** (Symmetry of two heads). *Suppose Definition 5.1 holds. Let $\mu_1, \mu_2 \in \mathcal{V} \cup \mathcal{C}$. For all $t$,*

$$w_+^{(t)\top}\mu_1 = w_-^{(t)\top}\mu_1 \cdot (-1)^{\mathbb{I}(\mu_1 \in \mathcal{V})} \tag{7}$$

$$\mu_2^\top W_{K+}^{(t)\top} W_{Q+}^{(t)}\mu_1 = \mu_2^\top W_{K-}^{(t)\top} W_{Q-}^{(t)}\mu_1 \cdot (-1)^{\mathbb{I}(\mu_2 \in \mathcal{V})}(-1)^{\mathbb{I}(\mu_1 \in \mathcal{V})} \tag{8}$$

$$\mu_2^\top W_{K+}^{(t)\top} W_{K+}^{(t)}\mu_1 = \mu_2^\top W_{K-}^{(t)\top} W_{K-}^{(t)}\mu_1 \cdot (-1)^{\mathbb{I}(\mu_2 \in \mathcal{V})}(-1)^{\mathbb{I}(\mu_1 \in \mathcal{V})} \tag{9}$$

$$\mu_2^\top W_{Q+}^{(t)\top} W_{Q+}^{(t)}\mu_1 = \mu_2^\top W_{Q-}^{(t)\top} W_{Q-}^{(t)}\mu_1 \cdot (-1)^{\mathbb{I}(\mu_2 \in \mathcal{V})}(-1)^{\mathbb{I}(\mu_1 \in \mathcal{V})} \tag{10}$$

**Corollary C.5.** *Let $s, s' \in \{\pm 1\}$. For all $t$,*

$$p^{(+,t)}(X_{k,s,s'}) = p^{(-,t)}(X_{k,-s,-s'})$$
$$H_+^{(t)}(X_{k,s,s'}) = H_-^{(t)}(X_{k,-s,-s'})$$
$$sf^{(t)}(X_{k,s,s'}) = -sf^{(t)}(X_{k,-s,-s'}).$$

*Proof of Lemma C.4.* First, all the equalities are satisfied when $t = 0$ by Definition 5.1. Assume all the equalities hold at time $t$ and we are going to show that the gradients with respect to $t$ also satisfy the equalities. Equation (8) implies that $p^{(+,t)}(X_{k,s,s'}) = p^{(-,t)}(X_{k,-s,-s'})$, which further implies $H_+^{(t)}(X_{k,s,s'}) = H_-^{(t)}(X_{k,-s,-s'})$ and $sf^{(t)}(X_{k,s,s'}) = -sf^{(t)}(X_{k,-s,-s'})$.

Define $\mu' := \mu \cdot (-1)^{\mathbb{I}(\mu \in \mathcal{V})}$ for $\mu \in \mathcal{V} \cup \mathcal{C}$ and $X'_{k,s,s'} := [c'_k, v'_k, v'_{k'}] = [c_k, -v_k, -v_{k'}] = X_{k,-s,-s'}$.

We first prove Equation (7):

$$\frac{\mathrm{d}w_+^{(t)\top}\mu_1}{\mathrm{d}t} = \mathbb{E}_{k,s,s'}\left[g^{(t)}s\dot\sigma_+^{(t)}\sum_{l=1}^L\left(X_{k,s,s'}p_l^{(+,t)}\right)^\top\right]\mu_1$$

$$\frac{\mathrm{d}w_-^{(t)\top}\mu_1'}{\mathrm{d}t} = -\mathbb{E}_{k,s,s'}\left[g^{(t)}s\dot{\sigma}_-^{(t)}\sum_{l=1}^{L}\left(X_{k,s,s'}p_l^{(-,t)}\right)^{\top}\right]\mu_1'$$

$$= \mathbb{E}_{k,s,s'}\left[g^{(t)}s\dot{\sigma}_-^{(t)}\sum_{l=1}^{L}\left(X'_{k,s,s'}p_l^{(-,t)}\right)^{\top}\right]\mu_1'$$

which implies

$$\frac{\mathrm{d}w_+^{(t)\top}\mu_1}{\mathrm{d}t} = \frac{\mathrm{d}w_-^{(t)\top}\mu_1'}{\mathrm{d}t}.$$

We next prove Equation (8):

$$\frac{\mathrm{d}\nu^{\top}W_{K+}^{(t)\top}W_{Q+}^{(t)}\mu}{\mathrm{d}t}$$

$$= \mathbb{E}_{k,s,s'}\left[g^{(t)}s\dot{\sigma}_+^{(t)}\sum_{l=1}^{L}\mu^{\top}W_{Q+}^{(t)\top}q_{+l}^{(t)}p_l^{(+,t)\top}\mathrm{diag}\left(w_+^{(t)\top}X_{k,s,s'} - w_+^{(t)\top}X_{k,s,s'}p_l^{(+,t)}\right)X_{k,s,s'}^{\top}\nu\right]$$

$$+ \mathbb{E}_{k,s,s'}\left[g^{(t)}s\dot{\sigma}_+^{(t)}\sum_{l=1}^{L}\nu^{\top}W_{K+}^{(t)\top}K_+^{(t)}\mathrm{diag}\left(X_{k,s,s'}^{\top}w_+ - w_+^{\top}X_{k,s,s'}p_l^{(+,t)}\right)p_l^{(+,t)}(X_{k,s,s'})_l^{\top}\mu\right]$$

$$\frac{\mathrm{d}(\nu')^{\top}W_{K-}^{(t)\top}W_{Q-}^{(t)}\mu'}{\mathrm{d}t}$$

$$= -\mathbb{E}_{k,s,s'}\left[g^{(t)}s\dot{\sigma}_-^{(t)}\sum_{l=1}^{L}(\mu')^{\top}W_{Q-}^{(t)\top}q_{-l}^{(t)}p_l^{(-,t)\top}\mathrm{diag}\left(w_-^{(t)\top}X_{k,s,s'} - w_-^{(t)\top}X_{k,s,s'}p_l^{(-,t)}\right)X_{k,s,s'}^{\top}\nu'\right]$$

$$- \mathbb{E}_{k,s,s'}\left[g^{(t)}s\dot{\sigma}_-^{(t)}\sum_{l=1}^{L}(\nu')^{\top}W_{K-}^{(t)\top}K_-^{(t)}\mathrm{diag}\left(X_{k,s,s'}^{\top}w_- - w_-^{\top}X_{k,s,s'}p_l^{(-,t)}\right)p_l^{(-,t)}(X_{k,s,s'})_l^{\top}\mu'\right]$$

$$= \mathbb{E}_{k,s,s'}\left[g^{(t)}s\dot{\sigma}_-^{(t)}\sum_{l=1}^{L}(\mu')^{\top}W_{Q-}^{(t)\top}q_{-l}^{(t)}p_l^{(-,t)\top}\mathrm{diag}\left(w_-^{(t)\top}X'_{k,s,s'} - w_-^{(t)\top}X'_{k,s,s'}p_l^{(-,t)}\right)X'^{\top}_{k,s,s'}\nu'\right]$$

$$+ \mathbb{E}_{k,s,s'}\left[g^{(t)}s\dot{\sigma}_-^{(t)}\sum_{l=1}^{L}(\nu')^{\top}W_{K-}^{(t)\top}K_-^{(t)}\mathrm{diag}\left(X'^{\top}_{k,s,s'}w_- - w_-^{\top}X'_{k,s,s'}p_l^{(-,t)}\right)p_l^{(-,t)}(X'_{k,s,s'})_l^{\top}\mu'\right]$$

which implies

$$\frac{\mathrm{d}\nu^{\top}W_{K+}^{(t)\top}W_{Q+}^{(t)}\mu}{\mathrm{d}t} = \frac{\mathrm{d}(\nu')^{\top}W_{K-}^{(t)\top}W_{Q-}^{(t)}\mu'}{\mathrm{d}t}.$$

The rest of equalities Equation (9) and Equation (10) can be proved similarly. $\qquad\square$

## D STAGE 1: TRAINING NEURONS TO LEARN THE SIGNALS

### D.1 STAGE 1.1: GROWTH OF NEURON ALIGNMENT WITH CLASSIFICATION SIGNALS

Due to our zero-initialization on the neuron weights, both attention heads are outputting zero. Since we add a small positive bias in the ReLU, for a short amount of time from the random initialization, we can ignore the ReLU activation.

**Lemma D.1** (Neuron alignment with the classification signal, stage 1.1). *Under Definition 5.1, for* $t \in [0, T_1]$*, if* $H_+^{(t)}(X), H_-^{(t)}(X) > -b$ *for all* $X \in \mathcal{D}$*, then for* $k \in \{1, 2\}$*,*

$$\frac{\mathrm{d}w_+^{(t)\top}v_k}{\mathrm{d}t} = \frac{1}{8}\left(\sum_{k'\in[2], s=s'}g^{(t)}(X_{k',s,s'})\right)$$

$$\frac{\mathrm{d}w_-^{(t)\top}v_k}{\mathrm{d}t} = -\frac{1}{8}\left(\sum_{k'\in[2],s=s'}g^{(t)}(X_{k',s,s'})\right).$$

*Proof.* Take $k=1$. By the gradient flow update, we have

$$\frac{\mathrm{d}w_+^{(t)\top}v_1}{\mathrm{d}t} = \mathbb{E}\left[g^{(t)}y\dot\sigma_+^{(t)}\sum_{l=1}^{L}\left(Xp_l^{(+,t)}\right)^\top\right]v_1$$

$$= \mathbb{E}\left[g^{(t)}y\sum_{l=1}^{L}\left(Xp_l^{(+,t)}\right)^\top\right]v_1$$

$$= \frac{1}{8}\left(\sum_{s=s'}g^{(t)}(X_{1,s,s'})\sum_{l=1}^{L}p_{q\leftarrow l,k\leftarrow v_1}^{(+,t)}(X_{1,s,s'})\right) + \frac{1}{8}\left(\sum_{s\neq s'}sg^{(t)}(X_{1,s,s'})\sum_{l=1}^{L}p_{q\leftarrow l,k\leftarrow sv_1}^{(+,t)}(X_{1,s,s'})\right)$$

$$+ \frac{1}{8}\left(\sum_{s=s'}g^{(t)}(X_{2,s,s'})\sum_{l=1}^{L}p_{q\leftarrow l,k\leftarrow v_1}^{(+,t)}(X_{2,s,s'})\right) - \frac{1}{8}\left(\sum_{s\neq s'}s'g^{(t)}(X_{2,s,s'})\sum_{l=1}^{L}p_{q\leftarrow l,k\leftarrow s'v_1}^{(+,t)}(X_{2,s,s'})\right)$$

$$= \frac{1}{8}\left(\sum_{k'\in[2],s=s'}g^{(t)}(X_{k',s,s'})\right). \hspace{2cm} \text{(by Corollary C.3)}$$

$\square$

**Lemma D.2** (No neuron alignment of group signals, stage 1.1)**.** *Under Definition 5.1, as long as $H_+^{(t)}(X), H_-^{(t)}(X) > -b$ for all $X$, and for $k \in \{1,2\}$,*

$$\frac{\mathrm{d}w_+^{(t)\top}c_k}{\mathrm{d}t} = 0, \quad \frac{\mathrm{d}w_-^{(t)\top}c_k}{\mathrm{d}t} = 0.$$

*Proof.* By the gradient flow update, we have

$$\frac{\mathrm{d}w_+^{(t)\top}c_k}{\mathrm{d}t} = \mathbb{E}\left[g^{(t)}y\dot\sigma_+^{(t)}\sum_{l=1}^{L}\left(Xp_l^{(+,t)}\right)^\top\right]c_k$$

$$= \mathbb{E}\left[g^{(t)}y\sum_{l=1}^{L}\left(Xp_l^{(+,t)}\right)^\top\right]c_k$$

$$= \frac{1}{8}\left(\sum_{s'}g^{(t)}(X_{k,+,s'})\sum_{l=1}^{L}p_{q\leftarrow l,k\leftarrow c_k}^{(+,t)} - \sum_{s'}g^{(t)}(X_{k,-,s'})\sum_{l=1}^{L}p_{q\leftarrow l,k\leftarrow c_k}^{(+,t)}\right)$$

$$= 0 \hspace{3cm} \text{(by and Corollary C.5)}$$

$\square$

**Corollary D.3** (Duration of stage 1.1)**.** *There exists a time $T_{0.5} = \Theta(1)$ such that $H_s^{(T_{0.5})}(X_{k,-s,-s}) \leq -b$ for $s \in \{\pm 1\}$.*

*Proof.* Lemma D.1 and Lemma D.2 imply that $g^{(t)}(X) = \Omega(1)$ for $t \leq O(1)$. Since $b$ is chosen to be a sufficiently small constant, there exists a time $T_{0.5} = \Theta(1)$ such that $H_s^{(T_{0.5})}(X_{k,-s,-s}) \leq -b$ for $s \in \{\pm 1\}$. $\square$

### D.2   STAGE 1.2: GROWTH OF NEURON ALIGNMENT WITH GROUP SIGNALS

In this sub-stage, we are going to show that (1) the neuron weight will keep positively aligning with the signed classification signal and (2) the neuron weights start to positively align with the group signal.

**Lemma D.4** (Growth of classification signals, stage 1.2). *Under Definition 5.1, for $t \in [0, T_1]$, if $H_+^{(t)}(X_{k,-,-}) < -b$ for all $k \in [2]$, then*

$$\frac{\mathrm{d}w_+^{(t)\top} v_k}{\mathrm{d}t} = \frac{1}{8} \left( \sum_{k' \in [2]} g^{(t)}(X_{k',+,+}) \right).$$

*Similarly, if $H_-^{(t)}(X_{k,+,+}) < -b$ for all $k \in [2]$, then*

$$\frac{\mathrm{d}w_-^{(t)\top} v_k}{\mathrm{d}t} = -\frac{1}{8} \left( \sum_{k' \in [2]} g^{(t)}(X_{k',-,-}) \right).$$

*Proof.* Take $k = 1$. If $H_+^{(t)}(X_{k',-,-}) < -b$ for all $k' \in [2]$, then

$$\frac{\mathrm{d}w_+^{(t)\top} v_1}{\mathrm{d}t} = \mathbb{E} \left[ g^{(t)} y \dot{\sigma}_+^{(t)} \sum_{l=1}^{L} \left( X p_l^{(+,t)} \right)^\top \right] v_1$$

$$= \frac{1}{8} \left( g^{(t)}(X_{1,+,+}) \sum_{l=1}^{L} p_{q \leftarrow l, k \leftarrow v_1}^{(+,t)}(X_{1,+,+}) \right) + \frac{1}{8} \left( \sum_{s \neq s'} s g^{(t)}(X_{1,s,s'}) \sum_{l=1}^{L} p_{q \leftarrow l, k \leftarrow s v_1}^{(+,t)}(X_{1,s,s'}) \right)$$

$$+ \frac{1}{8} \left( g^{(t)}(X_{2,+,+}) \sum_{l=1}^{L} p_{q \leftarrow l, k \leftarrow v_1}^{(+,t)}(X_{2,+,+}) \right) - \frac{1}{8} \left( \sum_{s \neq s'} s' g^{(t)}(X_{2,s,s'}) \sum_{l=1}^{L} p_{q \leftarrow l, k \leftarrow s' v_1}^{(+,t)}(X_{2,s,s'}) \right)$$

$$= \frac{1}{8} \left( \sum_{k' \in [2]} g^{(t)}(X_{k',+,+}) \right). \hspace{2cm} \text{(by Corollary C.3)}$$

$\square$

**Lemma D.5** (Growth of group signals, stage 1.2). *For $t \in [0, T_1]$, if $H_+^{(t)}(X_{k,-,-}) < -b$ for $k \in \{1, 2\}$, then*

$$\frac{\mathrm{d}w_+^{(t)\top} c_k}{\mathrm{d}t} = \frac{1}{8} g^{(t)}(X_{k,+,+}).$$

*Similarly, if $H_-^{(t)}(X_{k,+,+}) < -b$ for $k \in \{1, 2\}$, then*

$$\frac{\mathrm{d}w_-^{(t)\top} c_k}{\mathrm{d}t} = \frac{1}{8} g^{(t)}(X_{k,-,-}).$$

*Proof.* If $H_+^{(t)}(X_{k,-,-}) < -b$, then

$$\frac{\mathrm{d}w_+^{(t)\top} c_k}{\mathrm{d}t} = \mathbb{E} \left[ g^{(t)} y \dot{\sigma}_+^{(t)} \sum_{l=1}^{L} \left( X p_l^{(+,t)} \right)^\top \right] c_k$$

$$= \frac{1}{8} \left( \sum_{s'} g^{(t)}(X_{k,+,s'}) \sum_{l=1}^{L} p_{q \leftarrow l, k \leftarrow c_k}^{(+,t)}(X_{k,+,s'}) - g^{(t)}(X_{k,-,+}) \sum_{l=1}^{L} p_{q \leftarrow l, k \leftarrow c_k}^{(+,t)}(X_{k,-,+}) \right)$$

$$= \frac{1}{8} g^{(t)}(X_{k,+,+}). \hspace{2cm} \text{(by Corollary C.5)}$$

Similarly, if $H_-^{(t)}(X_{k,+,+}) < -b$ for $k \in [2]$, then

$$\frac{\mathrm{d}w_-^{(t)\top} c_k}{\mathrm{d}t} = -\mathbb{E} \left[ g^{(t)} y \dot{\sigma}_-^{(t)} \sum_{l=1}^{L} \left( X p_l^{(-,t)} \right)^\top \right] c_k$$

$$= -\frac{1}{8}\left(-\sum_{s'} g^{(t)}(X_{k,-,s'}) \sum_{l=1}^{L} p_{q \leftarrow l, k \leftarrow c_k}^{(-,t)}(X_{k,-,s'}) + g^{(t)}(X_{k,+,-}) \sum_{l=1}^{L} p_{q \leftarrow l, k \leftarrow c_k}^{(-,t)}(X_{k,+,-})\right)$$

$$= \frac{1}{8} g^{(t)}(X_{k,-,-}). \qquad \text{(by Corollary C.5)}$$

□

### D.3 PROPERTIES AT THE END OF STAGE 1

**Corollary D.6** (Activation of different types of samples). *For $t \in [T_{0.5}, T_1]$ and $s \in \{\pm 1\}$, $k \in [K]$, only $\dot{\sigma}_s^{(t)}(X_{k,-s,-s}) = 0$ and all the rest of cases equal 1.*

*Proof.* First of all, by the neuron signal alignment lemma stage 1.2 (Lemma D.4 and Lemma D.5), we have

$$\frac{\mathrm{d}w_+^{(t)\top}(v_1 + v_2)}{\mathrm{d}t} > \frac{\mathrm{d}w_+^{(t)\top} c_1}{\mathrm{d}t}.$$

Since the attention is uniform during stage 1, by Corollary D.3, we continue to have $\dot{\sigma}_+(X_{k,-,-}) = 0$ in stage 1.2. □

**Corollary D.7** (Neuron alignments at the end of stage 1). *There exists a time $T_1 = \Theta(1)$ such that $w_+^{(T_1)\top}\mu = \Theta(1)$ for all $\mu \in \mathcal{V} \cup \mathcal{C}$.*

*Proof.* This is a direct consequence of Lemma D.4 and Lemma D.5. □

As a result of stage 1, we can show that the transformer model now can produce a positive margin on the consistent samples but it fails to classify the conflicting samples.

**Theorem D.8** (Stage 1). *Under Definition 5.1, we have $f^{(T_1)}(X_{k,s,s}) = s \cdot \Omega(1)$ and $f^{(T_1)}(X_{k,s,-s}) = 0$.*

*Proof.* Since the attention during stage 1 is uniform, this is a direct consequence of Lemma D.1, Lemma D.4, Lemma D.5 and the symmetry between two heads in Lemma C.4. □

## E  STAGE 2: SELF-ATTENTION LEARNS FROM THE CONFLICTING (HARD) SAMPLES

### E.1  GROWTH OF SCORES

The following result states that there exists a range where (1) the growth of $c_k^\top W_{K+}^{(t)\top} W_{Q+}^{(t)} v_k$ and $-c_k^\top W_{K+}^{(t)\top} W_{Q+}^{(t)} v_{k'}$ dominates other score variables; (2) the growth of $v_k^\top W_{Q+}^{(t)\top} W_{Q+}^{(t)} v_k, -v_k^\top W_{Q+}^{(t)\top} W_{Q+}^{(t)} v_{k'}$ dominates other query self-score variables; and (3) the growth of $c_k^\top W_{K+}^{(t)\top} W_{K+}^{(t)} c_k, -c_k^\top W_{K+}^{(t)\top} W_{K+}^{(t)} c_{k'}$ dominates other key self-score variables.

**Lemma E.1.** *For $t \geq T_1$, there exists a constant $C > 0$ such that as long as $\max_{\mu,\nu \in \mathcal{V} \cup \mathcal{C}} |\mu^\top W_{K+}^{(t)\top} W_{Q+}^{(t)} \nu| \leq C$, then there exist a constant $0 < c < 1$ such that*

$$c \cdot \min_{k \neq k' \in [K]} \left(c_k^\top W_{K+}^{(t)\top} W_{Q+}^{(t)} v_k, -c_k^\top W_{K+}^{(t)\top} W_{Q+}^{(t)} v_{k'}\right)$$

$$\geq \max\left(\left(\bigcup_{\mu,\nu \in \mathcal{C} \cup \mathcal{V}} \left\{\left|\mu^\top W_{K+}^{(t)\top} W_{Q+}^{(t)} \nu\right|\right\}\right) \setminus \left(\bigcup_{k \neq k' \in [K]} \left\{\left|c_k^\top W_{K+}^{(t)\top} W_{Q+}^{(t)} v_k\right|, \left|c_k^\top W_{K+}^{(t)\top} W_{Q+}^{(t)} v_{k'}\right|\right\}\right)\right)$$

(11)

$$c \cdot \min_{k \neq k' \in [K]} \left(v_k^\top W_{Q+}^{(t)\top} W_{Q+}^{(t)} v_k, -v_k^\top W_{Q+}^{(t)\top} W_{Q+}^{(t)} v_{k'}\right)$$

$$\geq \max \left( \bigcup_{k \neq k' \in [K]} \left\{ \left| v_k^\top W_{Q+}^{(t)\top} W_{Q+}^{(t)} c_k \right|, \left| v_k^\top W_{Q+}^{(t)\top} W_{Q+}^{(t)} c_{k'} \right|, \left| c_k^\top W_{Q+}^{(t)\top} W_{Q+}^{(t)} c_{k'} \right| \right\} \right) \tag{12}$$

$$c \cdot \min_{k \neq k' \in [K]} \left( c_k^\top W_{K+}^{(t)\top} W_{K+}^{(t)} c_k, -c_k^\top W_{K+}^{(t)\top} W_{K+}^{(t)} c_{k'} \right)$$

$$\geq \max \left( \bigcup_{k \neq k' \in [K]} \left\{ \left| v_k^\top W_{K+}^{(t)\top} W_{K+}^{(t)} c_k \right|, \left| v_k^\top W_{K+}^{(t)\top} W_{K+}^{(t)} c_{k'} \right|, \left| v_k^\top W_{K+}^{(t)\top} W_{K+}^{(t)} v_{k'} \right| \right\} \right) \tag{13}$$

*and*

$$\|W_{Q+}^{(t)} v_k\|_2^2 \geq \|W_{Q+}^{(t)} c_k\|_2^2, \quad \|W_{K+}^{(t)} c_k\|_2^2 \geq \|W_{K+}^{(t)} v_k\|_2^2. \tag{14}$$

*Proof.* The proof is by inspecting the gradient flow updates in Appendix E.5 and proving by induction. At $t = T_1$, all the scores equal 0 and further, we have

$$\left. \frac{dc_k^\top W_{K+}^{(t)\top} W_{Q+}^{(t)} v_k}{dt} \right|_{t=T_1}, \quad -\left. \frac{dc_k^\top W_{K+}^{(t)\top} W_{Q+}^{(t)} v_{k'}}{dt} \right|_{t=T_1} = \frac{4}{3} g^{(T_1)} \omega^2 m \left( w_+^{(T_1)\top} c_1 - w_+^{(T_1)\top} X p_l^{(+,T_1)} \right)$$

$$= \frac{4}{3} g^{(T_1)} \omega^2 \frac{2}{3} w_+^{(T_1)\top} c_1$$

On the other hand, we have

$$\arg\max \left. \frac{d}{dt} \right|_{t=T_1} \left( \left( \bigcup_{\mu, \nu \in \mathcal{C} \cup \mathcal{V}} \left\{ \left| \mu^\top W_{K+}^{(t)\top} W_{Q+}^{(t)} \nu \right| \right\} \right) \setminus \left( \bigcup_{k \neq k' \in [K]} \left\{ \left| c_k^\top W_{K+}^{(t)\top} W_{Q+}^{(t)} v_k \right|, \left| c_k^\top W_{K+}^{(t)\top} W_{Q+}^{(t)} v_{k'} \right| \right\} \right) \right)$$

$$= \left. \frac{dv_1^\top W_{K+}^{(t)\top} W_{Q+}^{(t)} c_1}{dt} \right|_{t=T_1}$$

and

$$\left. \frac{dv_1^\top W_{K+}^{(t)\top} W_{Q+}^{(t)} c_1}{dt} \right|_{t=T_1} = \frac{4}{3} g^{(T_1)} \omega^2 w_+^{(T_1)\top} X p_l^{(+,T_1)} = \frac{4}{3} g^{(T_1)} \omega^2 \frac{1}{3} w_+^{(T_1)\top} c_1$$

Thus, Equation (11) is satisfied at $t = T_1$.

Next, we have $\|W_{Q+}^{(T_1)} v_k\|_2^2 = \omega^2$, $v_k^\top W_{Q+}^{(T_1)\top} W_{Q+}^{(T_1)} v_{k'} = 0$, $v_k^\top W_{Q+}^{(T_1)\top} W_{Q+}^{(T_1)} c_k = 0$, $v_k^\top W_{Q+}^{(T_1)\top} W_{Q+}^{(T_1)} c_{k'} = 0$ and $c_k^\top W_{Q+}^{(T_1)\top} W_{Q+}^{(T_1)} c_{k'} = 0$. Further,

$$\left. \frac{dv_k^\top W_{Q+}^{(t)\top} W_{Q+}^{(t)} v_{k'}}{dt} \right|_{t=T_1}, \quad \left. \frac{dv_k^\top W_{Q+}^{(t)\top} W_{Q+}^{(t)} c_k}{dt} \right|_{t=T_1}, \quad \left. \frac{dv_k^\top W_{Q+}^{(t)\top} W_{Q+}^{(t)} c_{k'}}{dt} \right|_{t=T_1}, \quad \left. \frac{dc_k^\top W_{Q+}^{(t)\top} W_{Q+}^{(t)} c_{k'}}{dt} \right|_{t=T_1} = 0$$

Hence, Equation (12) is satisfied at $t = T_1$.

Similarly, we have $\|W_{K+}^{(T_1)} c_k\|_2^2 = \omega^2$, $c_k^\top W_{K+}^{(T_1)\top} W_{K+}^{(T_1)} c_{k'} = 0$, $v_k^\top W_{K+}^{(T_1)\top} W_{K+}^{(T_1)} c_k = 0$, $v_k^\top W_{K+}^{(T_1)\top} W_{K+}^{(T_1)} c_{k'} = 0$ and $v_k^\top W_{K+}^{(T_1)\top} W_{K+}^{(T_1)} v_{k'} = 0$. In addition,

$$\left. \frac{dc_k^\top W_{K+}^{(t)\top} W_{K+}^{(t)} c_{k'}}{dt} \right|_{t=T_1}, \quad \left. \frac{dv_k^\top W_{K+}^{(t)\top} W_{K+}^{(t)} c_k}{dt} \right|_{t=T_1}, \quad \left. \frac{dv_k^\top W_{K+}^{(t)\top} W_{K+}^{(t)} c_{k'}}{dt} \right|_{t=T_1}, \quad \left. \frac{dv_k^\top W_{K+}^{(t)\top} W_{K+}^{(t)} v_{k'}}{dt} \right|_{t=T_1} = 0$$

Thus, Equation (13) is satisfied at $t = T_1$. Finally, Equation (14) is also satisfied at $t = T_1$.

Now, assume Equation (11), Equation (12) and Equation (13) hold at some $t > T_1$. We are going to show that the gradients at $t$ also satisfy those inequalities. We are going to prove the inequality between $c_k^\top W_{K+}^{(t)\top} W_{Q+}^{(t)} v_k$ and $v_1^\top W_{K+}^{(t)\top} W_{Q+}^{(t)} c_1$. The proof for the rest of cases is similar by checking the gradient flow dynamical system in Appendix E.5 as they all follow similar structures.

First of all, notice that if we replace $p^{(+,t)}$ with $p^{(+,T_1)}$ in $\frac{dc_1^\top W_{K+}^{(t)\top} W_{Q+}^{(t)} v_1}{dt}$ and $\frac{dv_1^\top W_{K+}^{(t)\top} W_{Q+}^{(t)} c_1}{dt}$, it holds that there exists a constant $0 < c' < 1$ such that

$$c' \cdot \frac{1}{3} g^{(t)} 2 \left( v_1^\top W_{Q+}^{(t)\top} W_{Q+}^{(t)} v_1 - v_1^\top W_{Q+}^{(t)\top} W_{Q+}^{(t)} v_2 \right) \left( w_+^{(t)\top} c_1 - w_+^{(t)\top} X p_l^{(+,T_1)} \right)$$

$$+ \frac{1}{3} g^{(t)} 2 \left( c_1^\top W_{K+}^{(t)\top} W_{K+}^{(t)} c_1 - c_1^\top W_{K+}^{(t)\top} W_{K+}^{(t)} c_2 \right) \left( w_+^{(t)\top} c_1 - w_+^{(t)\top} X p_l^{(+,T_1)} \right)$$

$$\geq \left( \frac{1}{3} g^{(t)} 2 \left( c_1^\top W_{Q+}^{(t)\top} W_{Q+}^{(t)} c_1 - c_1^\top W_{Q+}^{(t)\top} W_{Q+}^{(t)} c_2 \right) \left( -w_+^{(t)\top} X p_l^{(+,T_1)} \right) \right.$$

$$\left. + \frac{1}{3} g^{(t)} 2 \left( v_1^\top W_{K+}^{(t)\top} W_{K+}^{(t)} v_1 - v_1^\top W_{K+}^{(t)\top} W_{K+}^{(t)} v_2 \right) \left( -w_+^{(t)\top} X p_l^{(+,T_1)} \right) \right)$$

since $w_+^{(t)\top} c_1 - w_+^{(t)\top} X p_l^{(+,T_1)} = \frac{2}{3} w_+^{(t)\top} c_1$ and $w_+^{(t)\top} X p_l^{(+,T_1)} = \frac{1}{3} w_+^{(t)\top} c_1$. Next, we calculate the deviation of replacing $p^{(+,t)}$ with $p^{(+,T_1)}$ in the gradients

$$\frac{dc_1^\top W_{K+}^{(t)\top} W_{Q+}^{(t)} v_1}{dt}$$

$$- \left( \mathbb{E} \left[ g^{(t)} y \dot{\sigma}_+^{(t)} \sum_{l=1}^{L} v_1^\top W_{Q+}^{(t)\top} q_{+l}^{(t)} p_l^{(+,T_1)\top} \mathrm{diag} \left( w_+^{(t)\top} X - w_+^{(t)\top} X p_l^{(+,T_1)} \right) X^\top c_1 \right] \right.$$

$$\left. + \mathbb{E} \left[ g^{(t)} y \dot{\sigma}_+^{(t)} \sum_{l=1}^{L} c_1^\top W_{K+}^{(t)\top} K_+^{(t)} \mathrm{diag} \left( X^\top w_+ - w_+^\top X p_l^{(+,T_1)} \right) p_l^{(+,T_1)} x_l^\top v_1 \right] \right)$$

$$= \mathbb{E} \left[ g^{(t)} y \dot{\sigma}_+^{(t)} \sum_{l=1}^{L} v_1^\top W_{Q+}^{(t)\top} q_{+l}^{(t)} \left( p_l^{(+,t)\top} \mathrm{diag} \left( w_+^{(t)\top} X - w_+^{(t)\top} X p_l^{(+,t)} \right) \right. \right.$$

$$\left. \left. - p_l^{(+,T_1)\top} \mathrm{diag} \left( w_+^{(t)\top} X - w_+^{(t)\top} X p_l^{(+,T_1)} \right) \right) X^\top c_1 \right]$$

$$+ \mathbb{E} \left[ g^{(t)} y \dot{\sigma}_+^{(t)} \sum_{l=1}^{L} c_1^\top W_{K+}^{(t)\top} K_+^{(t)} \left( \mathrm{diag} \left( X^\top w_+ - w_+^\top X p_l^{(+,t)} \right) p_l^{(+,t)} \right. \right.$$

$$\left. \left. - \mathrm{diag} \left( X^\top w_+ - w_+^\top X p_l^{(+,T_1)} \right) p_l^{(+,T_1)} \right) x_l^\top v_1 \right]$$

Thus, there exists a constant $C > 0$ not too large such that $\max_{\mu,\nu \in \mathcal{V} \cup \mathcal{C}} |\mu^\top W_{K+}^{(t)\top} W_{Q+}^{(t)} \nu| \leq C$, and thus $\| p^{(+,t)} - p^{(+,T_1)} \|_\infty = O(1)$ and then

$$\left| \frac{dc_1^\top W_{K+}^{(t)\top} W_{Q+}^{(t)} v_1}{dt} - \left( \mathbb{E} \left[ g^{(t)} y \dot{\sigma}_+^{(t)} \sum_{l=1}^{L} v_1^\top W_{Q+}^{(t)\top} q_{+l}^{(t)} p_l^{(+,T_1)\top} \mathrm{diag} \left( w_+^{(t)\top} X - w_+^{(t)\top} X p_l^{(+,T_1)} \right) X^\top c_1 \right] \right. \right.$$

$$\left. \left. + \mathbb{E} \left[ g^{(t)} y \dot{\sigma}_+^{(t)} \sum_{l=1}^{L} c_1^\top W_{K+}^{(t)\top} K_+^{(t)} \mathrm{diag} \left( X^\top w_+ - w_+^\top X p_l^{(+,T_1)} \right) p_l^{(+,T_1)} x_l^\top v_1 \right] \right) \right|$$

$$\leq c'' \left( \frac{1}{3} g^{(t)} 2 \left( v_1^\top W_{Q+}^{(t)\top} W_{Q+}^{(t)} v_1 - v_1^\top W_{Q+}^{(t)\top} W_{Q+}^{(t)} v_2 \right) \left( w_+^{(t)\top} c_1 - w_+^{(t)\top} X p_l^{(+,T_1)} \right) \right.$$

$$\left. + \frac{1}{3} g^{(t)} 2 \left( c_1^\top W_{K+}^{(t)\top} W_{K+}^{(t)} c_1 - c_1^\top W_{K+}^{(t)\top} W_{K+}^{(t)} c_2 \right) \left( w_+^{(t)\top} c_1 - w_+^{(t)\top} X p_l^{(+,T_1)} \right) \right)$$

for some sufficiently small constant $c'' > 0$. This implies that if $c$ satisfies $c' + \Theta(1) < c < 1$ then $c \frac{dc_1^\top W_{K+}^{(t)\top} W_{Q+}^{(t)} v_1}{dt} \geq \frac{dv_1^\top W_{K+}^{(t)\top} W_{Q+}^{(t)} c_1}{dt}$. $\qquad \square$

Based on Lemma E.1, we can show that there exists a time range where the dynamical system exhibits exponential growth.

**Lemma E.2** (Score growth lemma). *For $t \geq T_1$, there exists a constant $C > 0$ such that as long as $\max_{\mu,\nu \in \mathcal{V} \cup \mathcal{C}} |\mu^\top W_{K+}^{(t)\top} W_{Q+}^{(t)} \nu| \leq C$, then there exists a constant $b > 0$ such that*

$$\frac{\mathrm{d}}{\mathrm{d}t} \begin{bmatrix} c_1^\top W_{K+}^{(t)\top} W_{Q+}^{(t)} v_1 \\ c_1^\top W_{K+}^{(t)\top} W_{K+}^{(t)} c_1 \\ v_1^\top W_{Q+}^{(t)\top} W_{Q+}^{(t)} v_1 \end{bmatrix} \geq b \begin{bmatrix} 0 & 1 & 1 \\ 1 & 0 & 0 \\ 1 & 0 & 0 \end{bmatrix} \begin{bmatrix} c_1^\top W_{K+}^{(t)\top} W_{Q+}^{(t)} v_1 \\ c_1^\top W_{K+}^{(t)\top} W_{K+}^{(t)} c_1 \\ v_1^\top W_{Q+}^{(t)\top} W_{Q+}^{(t)} v_1 \end{bmatrix}$$

*Proof.* First, as long as $\|p_l^{(+,t)}(X) - p_l^{(+,T_1)}(X)\|_\infty = O(1)$ for all $X$ and $l$, we have $g^{(t)}(X) \geq \Omega(1)$. Now, applying Lemma E.1 to the update of $c_1^\top W_{K+}^{(t)\top} W_{Q+}^{(t)} v_1$ in Appendix E.5.1, we can derive that there exists a constant $b_1 > 0$ such that

$$\frac{\mathrm{d}c_1^\top W_{K+}^{(t)\top} W_{Q+}^{(t)} v_1}{\mathrm{d}t} \geq b_1 \left( c_1^\top W_{K+}^{(t)\top} W_{K+}^{(t)} c_1 + v_1^\top W_{Q+}^{(t)\top} W_{Q+}^{(t)} v_1 \right)$$

and similarly, there exists $b_2, b_3$ such that

$$\frac{\mathrm{d}c_1^\top W_{K+}^{(t)\top} W_{K+}^{(t)} c_1}{\mathrm{d}t} \geq b_2 \cdot c_1^\top W_{K+}^{(t)\top} W_{Q+}^{(t)} v_1$$

$$\frac{\mathrm{d}v_1^\top W_{Q+}^{(t)\top} W_{Q+}^{(t)} v_1}{\mathrm{d}t} \geq b_3 \cdot c_1^\top W_{K+}^{(t)\top} W_{Q+}^{(t)} v_1$$

Take $b = \min(b_1, b_2, b_3)$ we finish the proof. $\square$

**Lemma E.3.** *There exists a $T_2$ such that $T_2 - T_1 = O(\log \frac{1}{\omega^2})$ and*

$$c_1^\top W_{K+}^{(T_2)\top} W_{Q+}^{(T_2)} v_1, -c_1^\top W_{K+}^{(T_2)\top} W_{Q+}^{(T_2)} v_2 = \Theta(1).$$

*Proof.* We can define a process $x(t)$ by

$$x(0) = \begin{bmatrix} 0 \\ \omega^2 \\ \omega^2 \end{bmatrix}, \quad \frac{\mathrm{d}x}{\mathrm{d}t} = \frac{b}{\sqrt{m}} Ax, \quad A = \begin{bmatrix} 0 & 1 & 1 \\ 1 & 0 & 0 \\ 1 & 0 & 0 \end{bmatrix}$$

and by Lemma E.2, we have

$$x(t) \leq \begin{bmatrix} c_1^\top W_{K+}^{(T_1+t)\top} W_{Q+}^{(T_1+t)} v_1 \\ c_1^\top W_{K+}^{(T_1+t)\top} W_{K+}^{(T_1+t)} c_1 \\ v_1^\top W_{Q+}^{(T_1+t)\top} W_{Q+}^{(T_1+t)} v_1 \end{bmatrix}.$$

Now, we analyze $x(t)$. We can compute the eigen-decomposition of $A$ as

$$\det(A - \lambda I) = \det\left( \begin{bmatrix} -\lambda & 1 & 1 \\ 1 & -\lambda & 0 \\ 1 & 0 & -\lambda \end{bmatrix} \right) = -\lambda^3 + 2\lambda = 0 \quad \Rightarrow \quad \lambda = 0, \pm\sqrt{2}.$$

The eigenvectors can be found by finding the null space of $A - \lambda I$:

$$\begin{bmatrix} -\lambda & 1 & 1 \\ 1 & -\lambda & 0 \\ 1 & 0 & -\lambda \end{bmatrix} \begin{bmatrix} x_1 \\ x_2 \\ x_3 \end{bmatrix} = \begin{bmatrix} 0 \\ 0 \\ 0 \end{bmatrix}$$

Thus, the eigenvector for $\lambda = 0$ is

$$C \cdot \begin{bmatrix} 0 \\ -1 \\ 1 \end{bmatrix}$$

and the eigenvectors for the non-zero eigenvalues are

$$C \cdot \begin{bmatrix} \lambda \\ 1 \\ 1 \end{bmatrix}$$

Therefore, plugging in the initial value $x(0)$, we have

$$x(t) = \frac{\omega^2}{2} \left( \exp\left(\sqrt{2}bt\right) \begin{bmatrix} \sqrt{2} \\ 1 \\ 1 \end{bmatrix} + \exp\left(-\sqrt{2}bt\right) \begin{bmatrix} -\sqrt{2} \\ 1 \\ 1 \end{bmatrix} \right)$$

Hence, there exists $T_2 - T_1 \leq O(\log \frac{1}{\omega^2})$ such that $c_1^\top W_{K+}^{(T_2)\top} W_{Q+}^{(T_2)} v_1 \geq x(T_2 - T_1) \geq \Omega(1)$.  □

### E.2 GROWTH OF ASYMMETRY

The following result shows that there exists a near-symmetry between the score and self-score variables.

**Lemma E.4.** *For* $t \geq T_1$, *there exists a constant* $C > 0$ *such that as long as* $\max_{\mu,\nu \in \mathcal{V} \cup \mathcal{C}} |\mu^\top W_{K+}^{(t)\top} W_{Q+}^{(t)} \nu| \leq C$, *if we define*

$$\Delta^{(t)} :=$$

$$\max_{\mu \in \mathcal{V} \cup \mathcal{C}} \left\{ \left| (c_1 + c_2)^\top W_{K+}^{(t)\top} W_{Q+}^{(t)} \mu \right|, \left| (v_1 + v_2)^\top W_{K+}^{(t)\top} W_{Q+}^{(t)} \mu \right|, \left| \mu^\top W_{K+}^{(t)\top} W_{Q+}^{(t)} (c_1 + c_2) \right|, \left| \mu^\top W_{K+}^{(t)\top} W_{Q+}^{(t)} (v_1 + v_2) \right| \right.$$

$$\left. \left| (c_1 + c_2)^\top W_{K+}^{(t)\top} W_{K+}^{(t)} \mu \right|, \left| (v_1 + v_2)^\top W_{K+}^{(t)\top} W_{K+}^{(t)} \mu \right|, \left| (c_1 + c_2)^\top W_{Q+}^{(t)\top} W_{Q+}^{(t)} \mu \right|, \left| (v_1 + v_2)^\top W_{Q+}^{(t)\top} W_{Q+}^{(t)} \mu \right| \right\}$$

*then*

$$\frac{\frac{d\Delta^{(t)}}{dt}}{\frac{dc_1^\top W_{K+}^{(T')\top} W_{Q+}^{(T')} v_1}{dt}} \leq \widetilde{O}(\omega).$$

*and*

$$\Delta^{(t)} \leq \widetilde{O}(\omega)$$

*Proof.* First of all, by the symmetry between two groups in Lemma C.2, we have

$$(c_1 + c_2)^\top W_{K+}^{(t)\top} W_{Q+}^{(t)} v_1 = c_1^\top W_{K+}^{(t)\top} W_{Q+}^{(t)} (v_1 + v_2)$$
$$(c_1 + c_2)^\top W_{K+}^{(t)\top} W_{Q+}^{(t)} c_1 = c_1^\top W_{K+}^{(t)\top} W_{Q+}^{(t)} (c_1 + c_2)$$
$$(v_1 + v_2)^\top W_{K+}^{(t)\top} W_{Q+}^{(t)} v_1 = v_1^\top W_{K+}^{(t)\top} W_{Q+}^{(t)} (v_1 + v_2)$$
$$(v_1 + v_2)^\top W_{K+}^{(t)\top} W_{Q+}^{(t)} c_1 = v_1^\top W_{K+}^{(t)\top} W_{Q+}^{(t)} (c_1 + c_2)$$

Thus,

$$\Delta^{(t)} = \max_{\mu \in \mathcal{V} \cup \mathcal{C}} \left\{ (c_1 + c_2)^\top W_{K+}^{(t)\top} W_{Q+}^{(t)} \mu, \; (v_1 + v_2)^\top W_{K+}^{(t)\top} W_{Q+}^{(t)} \mu, \right.$$

$$\left. (c_1 + c_2)^\top W_{K+}^{(t)\top} W_{K+}^{(t)} \mu, \; (v_1 + v_2)^\top W_{K+}^{(t)\top} W_{K+}^{(t)} \mu, (c_1 + c_2)^\top W_{Q+}^{(t)\top} W_{Q+}^{(t)} \mu, \; (v_1 + v_2)^\top W_{Q+}^{(t)\top} W_{Q+}^{(t)} \mu \right\}$$

At $t = T_1$, we have $\Delta^{(T_1)} = \omega^2$ and by the updates in Appendix E.5, we have

$$\left. \frac{d\Delta^{(t)}}{dt} \right|_{t=T_1} = 0$$

Define $\Delta p^{(t)} = \max_{l_1, l_2 \in [L], X \in \mathcal{D}} |p_{l_1, l_2}^{(t)}(X) - \frac{1}{3}|$. For $t \geq T_1$, from the gradients of the score variables in Appendix E.5 and Appendix E.6, we have

$$\left| \frac{d\Delta^{(t)}}{dt} \right| \leq O\left( \left| \Delta^{(t)} \Delta p^{(t)} \right| \right) \tag{15}$$

This can be seen from the following. Take $\frac{dc_1^\top W_{K+}^{(t)\top} W_{Q+}^{(t)} v_1}{dt}$ as an example: Define

$$
\left. \frac{dc_1^\top W_{K+}^{(t)\top} W_{Q+}^{(t)} v_1}{dt} \right|_{p^{(+,t)}=p^{(+,T_1)}}
$$

$$
:= \mathbb{E}\left[ g^{(t)} y \dot{\sigma}_+^{(t)} \sum_{l=1}^{L} v_1^\top W_{Q+}^{(t)\top} q_{+l}^{(t)} p_l^{(+,T_1)\top} \mathrm{diag}\left( w_+^{(t)\top} X - w_+^{(t)\top} X p_l^{(+,T_1)} \right) X^\top c_1 \middle| X_{k,s,-s} \right]
$$

$$
+ \mathbb{E}\left[ g^{(t)} y \dot{\sigma}_+^{(t)} \sum_{l=1}^{L} c_1^\top W_{K+}^{(t)\top} K_+^{(t)} \mathrm{diag}\left( X^\top w_+ - w_+^\top X p_l^{(+,T_1)} \right) p_l^{(+,T_1)} x_l^\top v_1 \middle| X_{k,s,-s} \right]
$$

Then, following similar steps in deriving Appendix E.5, it is not hard to derive that if we replace $p^{(+,t)}$ by $p^{(+,T_1)}$ in $\frac{dc_1^\top W_{K+}^{(t)\top} W_{Q+}^{(t)} v_1}{dt}$, we have

$$
\left. \frac{dc_1^\top W_{K+}^{(t)\top} W_{Q+}^{(t)} v_1}{dt} \right|_{p^{(+,t)}=p^{(+,T_1)}}
$$

$$
= \frac{1}{3} g^{(t)} 2 \left( v_1^\top W_{Q+}^{(t)\top} W_{Q+}^{(t)} v_1 - v_1^\top W_{Q+}^{(t)\top} W_{Q+}^{(t)} v_2 \right) \left( \frac{2}{3} w_+^{(T_1)\top} c_1 \right)
$$

$$
+ \frac{1}{3} g^{(t)} 2 \left( c_1^\top W_{K+}^{(t)\top} W_{K+}^{(t)} c_1 - c_1^\top W_{K+}^{(t)\top} W_{K+}^{(t)} c_2 \right) \left( \frac{2}{3} w_+^{(T_1)\top} c_1 \right)
$$

and thus,

$$
\left. \frac{dc_1^\top W_{K+}^{(t)\top} W_{Q+}^{(t)} (v_1 + v_2)}{dt} \right|_{p^{(+,t)}=p^{(+,T_1)}=\frac{1}{3}} = 0.
$$

Further, by the chain rule, we have

$$
\frac{dc_1^\top W_{K+}^{(t)\top} W_{Q+}^{(t)} (v_1 + v_2)}{dt} = c_1^\top W_{K+}^{(t)\top} \frac{dW_{Q+}^{(t)} (v_1 + v_2)}{dt} + \frac{dc_1^\top W_{K+}^{(t)\top}}{dt} W_{Q+}^{(t)} (v_1 + v_2)
$$

Therefore, expanding $\frac{dW_{Q+}^{(t)}(v_1+v_2)}{dt}$ via Appendix E.6, we have

$$
\left| \frac{dc_1^\top W_{K+}^{(t)\top} W_{Q+}^{(t)} (v_1 + v_2)}{dt} \right| = \left| \frac{dc_1^\top W_{K+}^{(t)\top} W_{Q+}^{(t)} (v_1 + v_2)}{dt} - \left. \frac{dc_1^\top W_{K+}^{(t)\top} W_{Q+}^{(t)} (v_1 + v_2)}{dt} \right|_{p^{(+,t)}=p^{(+,T_1)}=\frac{1}{3}} \right|
$$

$$
\leq O\left( \left| \Delta^{(t)} \Delta p^{(t)} \right| \right)
$$

Applying similar analysis to all other asymmetry variables, we get Equation (15).

Now, we analyze the growth of $\Delta^{(t)}$. Consider a specific time $T'$ such that $c_1^\top W_{K+}^{(T')\top} W_{Q+}^{(T')} v_1 = \Theta(\omega)$ for the first time and by Lemma E.1, we have $|\Delta p^{(t)}| \leq O(\omega)$ for all $t \leq T'$. Further, by Lemma E.2 and similar analysis in Lemma E.3, we have $T' - T_1 \leq O(\log(1/\omega))$. Thus,

$$
\left| \Delta^{(T')} \right| \leq \left| \Delta^{(T_1)} \right| + \int_{T_1}^{T'} \left| \frac{d\Delta^{(t)}}{dt} \right| dt \leq \omega^2 + \widetilde{O}(\omega^2).
$$

This implies that

$$
\frac{\Delta^{(T')}}{c_1^\top W_{K+}^{(T')\top} W_{Q+}^{(T')} v_1} \leq \frac{\widetilde{O}(\omega^2)}{\Theta(\omega)} \leq \widetilde{O}(\omega),
$$

Finally, comparing the gradient of $\Delta^{(t)}$ with the gradient of $c_1^\top W_{K+}^{(t)\top} W_{Q+}^{(t)} v_1$ for $t \in [T', T_2]$ (following similar analysis in Lemma E.1), we have

$$
\frac{\frac{d\Delta^{(t)}}{dt}}{\frac{dc_1^\top W_{K+}^{(T')\top} W_{Q+}^{(T')} v_1}{dt}} \leq \widetilde{O}(\omega).
$$

where the $\widetilde{O}$ is independent of $t$. Notice that this inequality on the ratio of gradients also holds for $t \leq T'$ since $\Delta p^{(t)} \leq O(\omega)$ for $t \leq T'$. This implies that $|\Delta^{(T_2)}| \leq |\Delta^{(T')}| + \widetilde{O}(\omega)|c_1^\top W_{K+}^{(T_2)\top} W_{Q+}^{(T_2)} v_1 - c_1^\top W_{K+}^{(T')\top} W_{Q+}^{(T')} v_1| \leq \widetilde{O}(\omega)$. $\qquad\square$

### E.3 WHEN THE QUERY IS THE GROUP SIGNAL

The main goal of this section is to analyze the group signal queries. To do this, we need to know the sign of the group signal queries. This can be analyzed by induction-like argument over all the scores and self-scores below.

**Lemma E.5** (Sign of all the score and self-score variables). *For $t \in [T_1, T_2]$,*

$$c_1 W_{K+}^{(t)\top} W_{Q+}^{(t)} v_1 \geq 0, \quad c_1 W_{K+}^{(t)\top} W_{Q+}^{(t)} v_2 \leq 0$$
$$v_1 W_{K+}^{(t)\top} W_{Q+}^{(t)} v_1 \leq 0, \quad v_1 W_{K+}^{(t)\top} W_{Q+}^{(t)} v_2 \geq 0$$
$$c_1 W_{K+}^{(t)\top} W_{Q+}^{(t)} c_1 \geq 0, \quad c_1 W_{K+}^{(t)\top} W_{Q+}^{(t)} c_2 \leq 0$$
$$v_1 W_{K+}^{(t)\top} W_{Q+}^{(t)} c_1 \leq 0, \quad v_1 W_{K+}^{(t)\top} W_{Q+}^{(t)} c_2 \geq 0$$
$$c_1 W_{Q+}^{(t)\top} W_{Q+}^{(t)} v_1 \geq 0, \quad c_1 W_{Q+}^{(t)\top} W_{Q+}^{(t)} v_2 \leq 0$$
$$c_1 W_{Q+}^{(t)\top} W_{Q+}^{(t)} c_1 \geq 0, \quad c_1 W_{Q+}^{(t)\top} W_{Q+}^{(t)} c_2 \leq 0$$
$$v_1 W_{Q+}^{(t)\top} W_{Q+}^{(t)} v_1 \geq 0, \quad v_1 W_{Q+}^{(t)\top} W_{Q+}^{(t)} v_2 \leq 0$$
$$c_1 W_{K+}^{(t)\top} W_{K+}^{(t)} v_1 \leq 0, \quad c_1 W_{K+}^{(t)\top} W_{K+}^{(t)} v_2 \geq 0$$
$$c_1 W_{K+}^{(t)\top} W_{K+}^{(t)} c_1 \geq 0, \quad c_1 W_{K+}^{(t)\top} W_{K+}^{(t)} c_2 \leq 0$$
$$v_1 W_{K+}^{(t)\top} W_{K+}^{(t)} v_1 \geq 0, \quad v_1 W_{K+}^{(t)\top} W_{K+}^{(t)} v_2 \leq 0$$

*Proof.* First of all, at $t = T_1$, all the variables are zero and thus the results are satisfied. Now, assume all the inequalities hold for some $t \geq T_1$, we show that the gradients of those score variables also satisfy those inequalities. We first prove the inequalities in the first column. First of all, by Lemma E.2, we have

$$\frac{dc_1 W_{K+}^{(t)\top} W_{Q+}^{(t)} v_1}{dt}, \quad \frac{dc_1^\top W_{K+}^{(t)\top} W_{K+}^{(t)} c_1}{dt}, \quad \frac{dv_1^\top W_{Q+}^{(t)\top} W_{Q+}^{(t)} v_1}{dt} \geq 0.$$

We now prove $v_1^\top W_{K+}^{(t)\top} W_{Q+}^{(t)} c_1 \leq 0$ as an example and the signs of all the other score variables can be analyzed similarly. By Appendix E.7, we have

$$\frac{dv_1^\top W_{K+}^{(t)\top} W_{Q+}^{(t)} c_1}{dt} = \mathbb{E}\left[ g^{(t)} y \dot\sigma_+^{(t)} \sum_{l=1}^L c_1^\top W_{Q+}^{(t)\top} q_{+l}^{(t)} p_l^{(+,t)\top} \mathrm{diag}\left( w_+^{(t)\top} X - w_+^{(t)\top} X p_l^{(+,t)} \right) X^\top v_1 \,\bigg|\, X_{k,+,-} \right]$$

$$+ \mathbb{E}\left[ g^{(t)} y \dot\sigma_+^{(t)} \sum_{l=1}^L c_1^\top W_{Q+}^{(t)\top} q_{+l}^{(t)} p_l^{(+,t)\top} \mathrm{diag}\left( w_+^{(t)\top} X - w_+^{(t)\top} X p_l^{(+,t)} \right) X^\top v_1 \,\bigg|\, X_{k,-,+} \right]$$

$$+ \mathbb{E}\left[ g^{(t)} y \dot\sigma_+^{(t)} \sum_{l=1}^L v_1^\top W_{K+}^{(t)\top} K_+^{(t)} \mathrm{diag}\left( X^\top w_+ - w_+^\top X p_l^{(+,t)} \right) p_l^{(+,t)} x_l^\top c_1 \,\bigg|\, X_{k,s,-s} \right]$$

We first analyze the sign of the sum of the first two terms.

$$\mathbb{E}\left[ g^{(t)} y \dot\sigma_+^{(t)} \sum_{l=1}^L c_1^\top W_{Q+}^{(t)\top} q_{+l}^{(t)} p_l^{(+,t)\top} \mathrm{diag}\left( w_+^{(t)\top} X - w_+^{(t)\top} X p_l^{(+,t)} \right) X^\top v_1 \,\bigg|\, X_{k,+,-} \right]$$

$$+ \mathbb{E}\left[ g^{(t)} y \dot\sigma_+^{(t)} \sum_{l=1}^L c_1^\top W_{Q+}^{(t)\top} q_{+l}^{(t)} p_l^{(+,t)\top} \mathrm{diag}\left( w_+^{(t)\top} X - w_+^{(t)\top} X p_l^{(+,t)} \right) X^\top v_1 \,\bigg|\, X_{k,-,+} \right]$$

$$= g^{(t)} \left( c_1^\top W_{Q+}^{(t)\top} W_{Q+}^{(t)} v_1 \left( p_{q\leftarrow v_1, k\leftarrow v_1}^{(+,t)} \left( -w_+^{(t)\top} X_{1,+,-} p_{q\leftarrow v_1}^{(+,t)} \right) + p_{q\leftarrow -v_1, k\leftarrow -v_1}^{(+,t)} \left( -w_+^{(t)\top} X_{2,+,-} p_{q\leftarrow -v_1}^{(+,t)} \right) \right) \right)$$

$$+ c_1^\top W_{Q+}^{(t)\top} W_{Q+}^{(t)} v_1 w_+^{(t)\top} v_1 \left( p_{q\leftarrow v_1, k\leftarrow v_1}^{(+,t)}(X_{1,+,-}) - p_{q\leftarrow -v_1, k\leftarrow -v_1}^{(+,t)}(X_{2,+,-}) \right)$$

$$+ c_1^\top W_{Q+}^{(t)\top} W_{Q+}^{(t)} c_1 p_{q\leftarrow c_1, k\leftarrow v_1}^{(+,t)} \left( -w_+^{(t)\top} X_{1,+,-} p_{q\leftarrow c_1}^{(+,t)} \right) - c_1^\top W_{Q+}^{(t)\top} W_{Q+}^{(t)} c_2 p_{q\leftarrow c_2, k\leftarrow -v_1}^{(+,t)} \left( -w_+^{(t)\top} X_{2,+,-} p_{q\leftarrow c_2}^{(+,t)} \right)$$

$$+ c_1^\top W_{Q+}^{(t)\top} W_{Q+}^{(t)}(-v_2) \left( p_{q\leftarrow -v_2, k\leftarrow v_1}^{(+,t)} \left( -w_+^{(t)\top} X_{1,+,-} p_{q\leftarrow -v_2}^{(+,t)} \right) + p_{q\leftarrow v_2, k\leftarrow -v_1}^{(+,t)} \left( -w_+^{(t)\top} X_{2,+,-} p_{q\leftarrow v_2}^{(+,t)} \right) \right)$$

$$+ c_1^\top W_{Q+}^{(t)\top} W_{Q+}^{(t)} v_2 w_+^{(t)\top} v_1 \left( -p_{q\leftarrow -v_2, k\leftarrow v_1}^{(+,t)}(X_{1,+,-}) + p_{q\leftarrow v_2, k\leftarrow -v_1}^{(+,t)}(X_{2,+,-}) \right)$$

$$+ c_1^\top W_{Q+}^{(t)\top} W_{Q+}^{(t)} c_1 p_{q\leftarrow c_1, k\leftarrow v_1}^{(+,t)}(X_{1,+,-}) w_+^{(t)\top} v_1 + c_1^\top W_{Q+}^{(t)\top} W_{Q+}^{(t)} c_2 p_{q\leftarrow c_2, k\leftarrow -v_1}^{(+,t)}(X_{2,+,-}) w_+^{(t)\top} v_1 \Bigg)$$

$$+ g^{(t)} \Bigg( c_1^\top W_{Q+}^{(t)\top} W_{Q+}^{(t)}(-v_1) \left( p_{q\leftarrow -v_1, k\leftarrow -v_1}^{(+,t)} \left( -w_+^{(t)\top} X_{1,-,+} p_{q\leftarrow -v_1}^{(+,t)} \right) + p_{q\leftarrow v_1, k\leftarrow v_1}^{(+,t)} \left( -w_+^{(t)\top} X_{2,-,+} p_{q\leftarrow v_1}^{(+,t)} \right) \right)$$

$$+ c_1^\top W_{Q+}^{(t)\top} W_{Q+}^{(t)}(-v_1) w_+^{(t)\top}(-v_1) \left( p_{q\leftarrow -v_1, k\leftarrow -v_1}^{(+,t)}(X_{1,-,+}) - p_{q\leftarrow v_1, k\leftarrow v_1}^{(+,t)}(X_{2,-,+}) \right)$$

$$+ c_1^\top W_{Q+}^{(t)\top} W_{Q+}^{(t)} c_1 p_{q\leftarrow c_1, k\leftarrow -v_1}^{(+,t)} \left( -w_+^{(t)\top} X_{1,-,+} p_{q\leftarrow c_1}^{(+,t)} \right) - c_1^\top W_{Q+}^{(t)\top} W_{Q+}^{(t)} c_2 p_{q\leftarrow c_2, k\leftarrow v_1}^{(+,t)} \left( -w_+^{(t)\top} X_{2,-,+} p_{q\leftarrow c_2}^{(+,t)} \right)$$

$$+ c_1^\top W_{Q+}^{(t)\top} W_{Q+}^{(t)} v_2 \left( p_{q\leftarrow v_2, k\leftarrow -v_1}^{(+,t)} \left( -w_+^{(t)\top} X_{1,-,+} p_{q\leftarrow v_2}^{(+,t)} \right) + p_{q\leftarrow -v_2, k\leftarrow v_1}^{(+,t)} \left( -w_+^{(t)\top} X_{2,-,+} p_{q\leftarrow -v_2}^{(+,t)} \right) \right)$$

$$+ c_1^\top W_{Q+}^{(t)\top} W_{Q+}^{(t)} v_2 \cdot w_+^{(t)\top}(-v_1) \left( p_{q\leftarrow v_2, k\leftarrow -v_1}^{(+,t)}(X_{1,-,+}) - p_{q\leftarrow -v_2, k\leftarrow v_1}^{(+,t)}(X_{2,-,+}) \right)$$

$$+ c_1^\top W_{Q+}^{(t)\top} W_{Q+}^{(t)} c_1 p_{q\leftarrow c_1, k\leftarrow -v_1}^{(+,t)}(X_{1,-,+}) w_+^{(t)\top}(-v_1) - c_1^\top W_{Q+}^{(t)\top} W_{Q+}^{(t)} c_2 p_{q\leftarrow c_2, k\leftarrow v_1}^{(+,t)}(X_{2,-,+}) w_+^{(t)\top} v_1 \Bigg)$$

Define

$$\Delta s^{(t)} := \max_{\mu \in \mathcal{V} \cup \mathcal{C}} \left\{ \left| (c_1 + c_2)^\top W_{K+}^{(t)\top} W_{Q+}^{(t)} \mu \right|, \ \left| (v_1 + v_2)^\top W_{K+}^{(t)\top} W_{Q+}^{(t)} \mu \right|, \right.$$
$$\left. \left| \mu^\top W_{K+}^{(t)\top} W_{Q+}^{(t)}(c_1 + c_2) \right|, \ \left| \mu^\top W_{K+}^{(t)\top} W_{Q+}^{(t)}(v_1 + v_2) \right| \right\}$$

Notice that

$$p_{q\leftarrow v_1, k\leftarrow v_1}^{(+,t)}(X_{1,+,-}) - p_{q\leftarrow -v_1, k\leftarrow -v_1}^{(+,t)}(X_{2,+,-}) = O(\Delta s^{(t)})$$
$$p_{q\leftarrow v_1, k\leftarrow v_1}^{(+,t)} w_+^{(t)\top} X_{1,+,-} p_{q\leftarrow v_1}^{(+,t)} = p_{q\leftarrow v_1, k\leftarrow c_1}^{(+,t)}(X_{1,+,-})(1 - p_{q\leftarrow v_1, k\leftarrow c_1}^{(+,t)}(X_{1,+,-}))/2 + O(\Delta s^{(t)})$$
$$p_{q\leftarrow -v_1, k\leftarrow -v_1}^{(+,t)} w_+^{(t)\top} X_{1,-,+} p_{q\leftarrow -v_1}^{(+,t)} = p_{q\leftarrow -v_1, k\leftarrow c_1}^{(+,t)}(X_{1,-,+})(1 - p_{q\leftarrow -v_1, k\leftarrow c_1}^{(+,t)}(X_{1,-,+}))/2 + O(\Delta s^{(t)})$$

Now consider the function $y(x) = (1-x)x/2$. Notice that if $x_1 < \frac{1}{3}$ and $\frac{1}{3} < x_2 < \frac{2}{3}$, then $y(x_2) > y(x_1)$. Therefore, there exists a constant $C > 0$ such that if $|c_1^\top W_{K+}^{(t)\top} W_{Q+}^{(t)} v_1| \leq C$, we have

$$p_{q\leftarrow v_1, k\leftarrow c_1}^{(+,t)}(X_{1,+,-})(1 - p_{q\leftarrow v_1, k\leftarrow c_1}^{(+,t)}(X_{1,+,-}))/2 > p_{q\leftarrow -v_1, k\leftarrow c_1}^{(+,t)}(X_{1,-,+})(1 - p_{q\leftarrow -v_1, k\leftarrow c_1}^{(+,t)}(X_{1,-,+}))/2$$

Next,

$$p_{q\leftarrow v_1, k\leftarrow c_1}^{(+,t)}(X_{1,+,-})(1 - p_{q\leftarrow v_1, k\leftarrow c_1}^{(+,t)}(X_{1,+,-}))/2 - p_{q\leftarrow -v_1, k\leftarrow c_1}^{(+,t)}(X_{1,-,+})(1 - p_{q\leftarrow -v_1, k\leftarrow c_1}^{(+,t)}(X_{1,-,+}))/2$$
$$= \Theta(c_1^\top W_{K+}^{(t)\top} W_{Q+}^{(t)} v_1) > O(\Delta s^{(t)})$$

where the inequality is by Lemma E.4. Thus,

$$c_1^\top W_{Q+}^{(t)\top} W_{Q+}^{(t)} v_1 \left( p_{q\leftarrow v_1, k\leftarrow v_1}^{(+,t)} \left( -w_+^{(t)\top} X_{1,+,-} p_{q\leftarrow v_1}^{(+,t)} \right) + p_{q\leftarrow -v_1, k\leftarrow -v_1}^{(+,t)} \left( -w_+^{(t)\top} X_{2,+,-} p_{q\leftarrow -v_1}^{(+,t)} \right) \right)$$

$$+ c_1^\top W_{Q+}^{(t)\top} W_{Q+}^{(t)} v_1 w_+^{(t)\top} v_1 \left( p_{q\leftarrow v_1, k\leftarrow v_1}^{(+,t)}(X_{1,+,-}) - p_{q\leftarrow -v_1, k\leftarrow -v_1}^{(+,t)}(X_{2,+,-}) \right)$$

$$+ c_1^\top W_{Q+}^{(t)\top} W_{Q+}^{(t)}(-v_1) \left( p_{q\leftarrow -v_1, k\leftarrow -v_1}^{(+,t)} \left( -w_+^{(t)\top} X_{1,-,+} p_{q\leftarrow -v_1}^{(+,t)} \right) + p_{q\leftarrow v_1, k\leftarrow v_1}^{(+,t)} \left( -w_+^{(t)\top} X_{2,-,+} p_{q\leftarrow v_1}^{(+,t)} \right) \right)$$

$$+ c_1^\top W_{Q+}^{(t)\top} W_{Q+}^{(t)}(-v_1) w_+^{(t)\top}(-v_1) \left( p_{q\leftarrow -v_1, k\leftarrow -v_1}^{(+,t)}(X_{1,-,+}) - p_{q\leftarrow v_1, k\leftarrow v_1}^{(+,t)}(X_{2,-,+}) \right)$$

$\leq 0$

Similarly,

$$c_1^\top W_{Q+}^{(t)\top} W_{Q+}^{(t)}(-v_2)\left(p_{q\leftarrow -v_2,k\leftarrow v_1}^{(+,t)}\left(-w_+^{(t)\top}X_{1,+,-}p_{q\leftarrow -v_2}^{(+,t)}\right)+p_{q\leftarrow v_2,k\leftarrow -v_1}^{(+,t)}\left(-w_+^{(t)\top}X_{2,+,-}p_{q\leftarrow v_2}^{(+,t)}\right)\right)$$
$$+c_1^\top W_{Q+}^{(t)\top} W_{Q+}^{(t)}v_2 w_+^{(t)\top}v_1\left(-p_{q\leftarrow -v_2,k\leftarrow v_1}^{(+,t)}(X_{1,+,-})+p_{q\leftarrow v_2,k\leftarrow -v_1}^{(+,t)}(X_{2,+,-})\right)$$
$$+c_1^\top W_{Q+}^{(t)\top} W_{Q+}^{(t)}v_2\left(p_{q\leftarrow -v_2,k\leftarrow -v_1}^{(+,t)}\left(-w_+^{(t)\top}X_{1,-,+}p_{q\leftarrow v_2}^{(+,t)}\right)+p_{q\leftarrow -v_2,k\leftarrow v_1}^{(+,t)}\left(-w_+^{(t)\top}X_{2,-,+}p_{q\leftarrow -v_2}^{(+,t)}\right)\right)$$
$$+c_1^\top W_{Q+}^{(t)\top} W_{Q+}^{(t)}v_2\cdot w_+^{(t)\top}(-v_1)\left(p_{q\leftarrow v_2,k\leftarrow -v_1}^{(+,t)}(X_{1,-,+})-p_{q\leftarrow -v_2,k\leftarrow v_1}^{(+,t)}(X_{2,-,+})\right)$$
$$\leq 0$$

Finally, since $v_1^\top W_{K+}^{(t)\top} W_{Q+}^{(t)}c_1 \leq 0, v_2^\top W_{K+}^{(t)\top} W_{Q+}^{(t)}c_1 \geq 0$,

$$c_1^\top W_{Q+}^{(t)\top} W_{Q+}^{(t)}c_1 p_{q\leftarrow c_1,k\leftarrow v_1}^{(+,t)}(X_{1,+,-})w_+^{(t)\top}v_1 + c_1^\top W_{Q+}^{(t)\top} W_{Q+}^{(t)}c_2 p_{q\leftarrow c_2,k\leftarrow -v_1}^{(+,t)}(X_{2,+,-})w_+^{(t)\top}v_1$$
$$+c_1^\top W_{Q+}^{(t)\top} W_{Q+}^{(t)}c_1 p_{q\leftarrow c_1,k\leftarrow -v_1}^{(+,t)}(X_{1,-,+})w_+^{(t)\top}(-v_1) - c_1^\top W_{Q+}^{(t)\top} W_{Q+}^{(t)}c_2 p_{q\leftarrow c_2,k\leftarrow v_1}^{(+,t)}(X_{2,-,+})w_+^{(t)\top}v_1$$
$$\leq 0$$

Thus,

$$\frac{\mathrm{d}v_1^\top W_{K+}^{(t)\top} W_{Q+}^{(t)}c_1}{\mathrm{d}t} \leq 0.$$

The inequalities in the second column can be proved by noticing the symmetric structure of the gradients of score variables in Appendix E.6. Thus, the gradient of the variables in the second column has the exact opposite sign of the corresponding variables in the first column. □

### E.4 Properties at the end of stage 2

**Lemma E.6** (Activation in stage 2). *For $t \in [T_1, T_2]$,*

$$\forall k \in [2], s \in \{\pm 1\}:\ \sigma_+^{(t)}(X_{k,s,-s}) > 0$$

*Proof.* This is a direct consequences of Corollary D.7 and Lemma E.4. □

**Theorem E.7** (Attention at the end of stage 2). *There exists a $T_2 = O(\log(1/\omega))$ such that*

- *For samples $X_{1,+,+}$ and $X_{1,-,-}$,*
  - *when the query token is the group signal $c_1$,*
  
  $$\left|p_{q\leftarrow c_1,k\leftarrow c_1}^{(+,T_2)}(X_{1,+,+}) - p_{q\leftarrow c_1,k\leftarrow c_1}^{(+,T_2)}(X_{1,-,-})\right| \leq \widetilde{O}(\omega)$$
  $$\left|p_{q\leftarrow c_1,k\leftarrow v_1}^{(+,T_2)}(X_{1,+,+}) - p_{q\leftarrow c_1,k\leftarrow -v_2}^{(+,T_2)}(X_{1,-,-})\right| \leq \widetilde{O}(\omega)$$
  $$\left|p_{q\leftarrow c_1,k\leftarrow v_2}^{(+,T_2)}(X_{1,+,+}) - p_{q\leftarrow c_1,k\leftarrow -v_1}^{(+,T_2)}(X_{1,-,-})\right| \leq \widetilde{O}(\omega)$$
  
  - *when the query token is the classification signal,*
  
  $$\left|p_{q\leftarrow v_1,k\leftarrow c_1}^{(+,T_2)}(X_{1,+,+}) - p_{q\leftarrow -v_2,k\leftarrow c_1}^{(+,T_2)}(X_{1,-,-})\right| \leq \widetilde{O}(\omega)$$
  $$\left|p_{q\leftarrow v_1,k\leftarrow v_1}^{(+,T_2)}(X_{1,+,+}) - p_{q\leftarrow -v_2,k\leftarrow -v_2}^{(+,T_2)}(X_{1,-,-})\right| \leq \widetilde{O}(\omega)$$
  $$\left|p_{q\leftarrow v_1,k\leftarrow v_2}^{(+,T_2)}(X_{1,+,+}) - p_{q\leftarrow -v_2,k\leftarrow -v_1}^{(+,T_2)}(X_{1,-,-})\right| \leq \widetilde{O}(\omega)$$
  $$\left|p_{q\leftarrow v_2,k\leftarrow c_1}^{(+,T_2)}(X_{1,+,+}) - p_{q\leftarrow -v_1,k\leftarrow c_1}^{(+,T_2)}(X_{1,-,-})\right| \leq \widetilde{O}(\omega)$$
  $$\left|p_{q\leftarrow v_2,k\leftarrow v_2}^{(+,T_2)}(X_{1,+,+}) - p_{q\leftarrow -v_1,k\leftarrow -v_1}^{(+,T_2)}(X_{1,-,-})\right| \leq \widetilde{O}(\omega)$$
  $$\left|p_{q\leftarrow v_2,k\leftarrow v_1}^{(+,T_2)}(X_{1,+,+}) - p_{q\leftarrow -v_1,k\leftarrow -v_2}^{(+,T_2)}(X_{1,-,-})\right| \leq \widetilde{O}(\omega)$$

- *For samples $X_{1,+,-}$ and $X_{1,-,+}$,*

  - *when the query is the group signal $c_1$,*

$$p_{q\leftarrow c_1, k\leftarrow c_1}^{(+,T_2)}(X_{1,+,-}) > p_{q\leftarrow c_1, k\leftarrow c_1}^{(+,T_2)}(X_{1,-,+})$$

$$\left| p_{q\leftarrow c_1, k\leftarrow v_1}^{(+,T_2)}(X_{1,+,-}) - p_{q\leftarrow c_1, k\leftarrow -v_2}^{(+,T_2)}(X_{1,+,-}) \right| \leq \widetilde{O}(\omega)$$

$$\left| p_{q\leftarrow c_1, k\leftarrow -v_1}^{(+,T_2)}(X_{1,-,+}) - p_{q\leftarrow c_1, k\leftarrow v_2}^{(+,T_2)}(X_{1,-,+}) \right| \leq \widetilde{O}(\omega)$$

  - *when the query is the classification signal,*

$$p_{q\leftarrow v_1, k\leftarrow c_1}^{(+,t)}(X_{1,+,-}) - p_{q\leftarrow -v_1, k\leftarrow c_1}^{(+,t)}(X_{1,-,+}) = \Theta(1)$$

$$p_{q\leftarrow -v_2, k\leftarrow c_1}^{(+,t)}(X_{1,+,-}) - p_{q\leftarrow v_2, k\leftarrow c_1}^{(+,t)}(X_{1,-,+}) = \Theta(1)$$

$$\left| p_{q\leftarrow v_1, k\leftarrow v_1}^{(+,t)}(X_{1,+,-}) - p_{q\leftarrow v_1, k\leftarrow -v_2}^{(+,t)}(X_{1,+,-}) \right| \leq \widetilde{O}(\omega)$$

$$\left| p_{q\leftarrow -v_2, k\leftarrow v_1}^{(+,t)}(X_{1,+,-}) - p_{q\leftarrow -v_2, k\leftarrow -v_2}^{(+,t)}(X_{1,+,-}) \right| \leq \widetilde{O}(\omega)$$

$$\left| p_{q\leftarrow -v_2, k\leftarrow c_1}^{(+,t)}(X_{1,+,-}) - p_{q\leftarrow v_1, k\leftarrow c_1}^{(+,t)}(X_{1,+,-}) \right| \leq \widetilde{O}(\omega)$$

*Proof.* The proof is a direct consequence of Lemma E.3, Lemma E.4 and Lemma E.5. □

## E.5 THE UPDATES OF THE SOFTMAX ATTENTION AT $t = T_1$

In this section, we give out the explicit updates for the scores and self-scores at $t = T_1$.

The equations in this section are derived by the following:

1. Notice that $p_{l_1,l_2}^{(+,T_1)} = \frac{1}{3}$ for $l_1, l_2 \in [L]$ and thus $w_+^{(T_1)} X_{k,s,-s} p_l^{(+,T_1)} = \frac{1}{3} w_+^{(T_1)\top} c_k$ where $s \in \{\pm 1\}$.

2. Therefore, by Corollary D.7, $w_+^{(T_1)\top} c_k = \Omega(1)$ and thus $\dot{\sigma}_+^{(T_1)}(X_{1,+,-}) = \dot{\sigma}_+^{(T_1)}(X_{1,-,+}) = \dot{\sigma}_+^{(T_1)}(X_{2,+,-}) = \dot{\sigma}_+^{(T_1)}(X_{2,-,+}) = 1$.

3. By symmetry between two groups and two heads in Lemma C.2 and Lemma C.4, we have $w_+^{(T_1)\top} c_1 = w_+^{(T_1)\top} c_2$ and $g^{(T_1)}(X_{1,+,-}) = g^{(T_1)}(X_{1,-,+}) = g^{(T_1)}(X_{2,+,-}) = g^{(T_1)}(X_{2,-,+})$. Thus, we denote $g^{(T_1)} = g^{(T_1)}(X_{1,+,-})$.

### E.5.1 THE SCORES

$$\left. \frac{\mathrm{d} c_1^\top W_{K+}^{(t)\top} W_{Q+}^{(t)} v_1}{\mathrm{d}t} \right|_{t=T_1}$$

$$= \mathbb{E}\left[ g^{(t)} y \dot{\sigma}_+^{(t)} \sum_{l=1}^L v_1^\top W_{Q+}^{(t)\top} q_{+l}^{(t)} p_l^{(+,t)\top} \mathrm{diag}\left( w_+^{(t)\top} X - w_+^{(t)\top} X p_l^{(+,t)} \right) X^\top c_1 \,\Big|\, X_{k,s,-s} \right]\Bigg|_{t=T_1}$$

$$+ \mathbb{E}\left[ g^{(t)} y \dot{\sigma}_+^{(t)} \sum_{l=1}^L c_1^\top W_{K+}^{(t)\top} K_+^{(t)} \mathrm{diag}\left( X^\top w_+ - w_+^\top X p_l^{(+,t)} \right) p_l^{(+,t)} x_l^\top v_1 \,\Big|\, X_{k,s,-s} \right]\Bigg|_{t=T_1}$$

$$= \frac{1}{3} g^{(T_1)} 2 \left( v_1^\top W_{Q+}^{(T_1)\top} W_{Q+}^{(T_1)} v_1 - v_1^\top W_{Q+}^{(T_1)\top} W_{Q+}^{(T_1)} v_2 \right) \left( w_+^{(T_1)\top} c_1 - w_+^{(T_1)\top} X p_l^{(+,T_1)} \right)$$

$$+ \frac{1}{3} g^{(T_1)} 2 \left( c_1^\top W_{K+}^{(T_1)\top} W_{K+}^{(T_1)} c_1 - c_1^\top W_{K+}^{(T_1)\top} W_{K+}^{(T_1)} c_2 \right) \left( w_+^{(T_1)\top} c_1 - w_+^{(T_1)\top} X p_l^{(+,T_1)} \right)$$

$$= \frac{1}{3} g^{(T_1)} 2 \left( v_1^\top W_{Q+}^{(T_1)\top} W_{Q+}^{(T_1)} v_1 - v_1^\top W_{Q+}^{(T_1)\top} W_{Q+}^{(T_1)} v_2 \right) \left( \frac{2}{3} w_+^{(T_1)\top} c_1 \right)$$

$$+ \frac{1}{3} g^{(T_1)} 2 \left( c_1^\top W_{K+}^{(T_1)\top} W_{K+}^{(T_1)} c_1 - c_1^\top W_{K+}^{(T_1)\top} W_{K+}^{(T_1)} c_2 \right) \left( \frac{2}{3} w_+^{(T_1)\top} c_1 \right)$$

$$\frac{\mathrm{d}c_1^\top W_{K+}^{(t)\top} W_{Q+}^{(t)} v_2}{\mathrm{d}t}\bigg|_{t=T_1}$$

$$= \frac{1}{3} g^{(T_1)} 2 \left( v_2^\top W_{Q+}^{(T_1)\top} W_{Q+}^{(T_1)} v_1 - v_2^\top W_{Q+}^{(T_1)\top} W_{Q+}^{(T_1)} v_2 \right) \left( w_+^{(T_1)\top} c_1 - w_+^{(T_1)\top} X p_l^{(+,T_1)} \right)$$

$$+ \frac{1}{3} g^{(T_1)} 2 \left( -c_1^\top W_{K+}^{(T_1)\top} W_{K+}^{(T_1)} c_1 + c_1^\top W_{K+}^{(T_1)\top} W_{K+}^{(T_1)} c_2 \right) \left( w_+^{(T_1)\top} c_1 - w_+^{(T_1)\top} X p_l^{(+,t)} \right)$$

$$= \frac{1}{3} g^{(T_1)} 2 \left( v_2^\top W_{Q+}^{(T_1)\top} W_{Q+}^{(T_1)} v_1 - v_2^\top W_{Q+}^{(T_1)\top} W_{Q+}^{(T_1)} v_2 \right) \left( \frac{2}{3} w_+^{(T_1)\top} c_1 \right)$$

$$+ \frac{1}{3} g^{(T_1)} 2 \left( -c_1^\top W_{K+}^{(T_1)\top} W_{K+}^{(T_1)} c_1 + c_1^\top W_{K+}^{(T_1)\top} W_{K+}^{(T_1)} c_2 \right) \left( \frac{2}{3} w_+^{(T_1)\top} c_1 \right)$$

$$\frac{\mathrm{d}c_1^\top W_{K+}^{(t)\top} W_{Q+}^{(t)} c_1}{\mathrm{d}t}\bigg|_{t=T_1}$$

$$= \mathbb{E} \left[ g^{(t)} y \dot{\sigma}_+^{(t)} \sum_{l=1}^L c_1^\top W_{Q+}^{(t)\top} q_{+l}^{(t)} p_l^{(+,t)\top} \mathrm{diag} \left( w_+^{(t)\top} X - w_+^{(t)\top} X p_l^{(+,t)} \right) X^\top c_1 \bigg| X_{k,s,-s} \right]\bigg|_{t=T_1}$$

$$+ \mathbb{E} \left[ g^{(t)} y \dot{\sigma}_+^{(t)} \sum_{l=1}^L c_1^\top W_{K+}^{(t)\top} K_+^{(t)} \mathrm{diag} \left( X^\top w_+ - w_+^\top X p_l^{(+,t)} \right) p_l^{(+,t)} x_l^\top c_1 \bigg| X_{k,s,-s} \right]\bigg|_{t=T_1}$$

$$= \frac{1}{3} g^{(T_1)} 2 \left( c_1^\top W_{Q+}^{(T_1)\top} W_{Q+}^{(T_1)} v_1 - c_1^\top W_{Q+}^{(T_1)\top} W_{Q+}^{(T_1)} v_2 \right) \left( w_+^{(T_1)} c_1 - w_+^{(T_1)\top} X p_l^{(+,T_1)} \right)$$

$$+ \frac{1}{3} g^{(T_1)} 2 \left( c_1^\top W_{K+}^{(T_1)\top} W_{K+}^{(T_1)} v_1 - c_1^\top W_{K+}^{(T_1)\top} W_{K+}^{(T_1)} v_2 \right) \left( -w_+^{(T_1)\top} X p_l^{(+,T_1)} \right)$$

$$= \frac{1}{3} g^{(T_1)} 2 \left( c_1^\top W_{Q+}^{(T_1)\top} W_{Q+}^{(T_1)} v_1 - c_1^\top W_{Q+}^{(T_1)\top} W_{Q+}^{(T_1)} v_2 \right) \left( \frac{2}{3} w_+^{(T_1)\top} c_1 \right)$$

$$+ \frac{1}{3} g^{(T_1)} 2 \left( c_1^\top W_{K+}^{(T_1)\top} W_{K+}^{(T_1)} v_1 - c_1^\top W_{K+}^{(T_1)\top} W_{K+}^{(T_1)} v_2 \right) \left( -\frac{1}{3} w_+^{(T_1)\top} c_1 \right)$$

$$\frac{\mathrm{d}c_1^\top W_{K+}^{(t)\top} W_{Q+}^{(t)} c_2}{\mathrm{d}t}\bigg|_{t=T_1}$$

$$= \frac{1}{3} g^{(t)} 2 \left( c_1^\top W_{Q+}^{(T_1)\top} W_{Q+}^{(T_1)} v_2 - c_1^\top W_{Q+}^{(T_1)\top} W_{Q+}^{(T_1)} v_1 \right) \left( w_+^{(T_1)} c_2 - w_+^{(T_1)\top} X p_l^{(+,T_1)} \right)$$

$$+ \frac{1}{3} g^{(T_1)} 2 \left( c_1^\top W_{K+}^{(T_1)\top} W_{K+}^{(T_1)} v_2 - c_1^\top W_{K+}^{(T_1)\top} W_{K+}^{(T_1)} v_1 \right) \left( -w_+^{(T_1)\top} X p_l^{(+,T_1)} \right)$$

$$= \frac{1}{3} g^{(t)} 2 \left( c_1^\top W_{Q+}^{(T_1)\top} W_{Q+}^{(T_1)} v_2 - c_1^\top W_{Q+}^{(T_1)\top} W_{Q+}^{(T_1)} v_1 \right) \left( \frac{2}{3} w_+^{(T_1)\top} c_2 \right)$$

$$+ \frac{1}{3} g^{(T_1)} 2 \left( c_1^\top W_{K+}^{(T_1)\top} W_{K+}^{(T_1)} v_2 - c_1^\top W_{K+}^{(T_1)\top} W_{K+}^{(T_1)} v_1 \right) \left( -\frac{1}{3} w_+^{(T_1)\top} c_1 \right)$$

$$\frac{\mathrm{d}v_1^\top W_{K+}^{(t)\top} W_{Q+}^{(t)} c_1}{\mathrm{d}t}\bigg|_{t=T_1}$$

$$= \mathbb{E} \left[ g^{(t)} y \dot{\sigma}_+^{(t)} \sum_{l=1}^L c_1^\top W_{Q+}^{(t)\top} q_{+l}^{(t)} p_l^{(+,t)\top} \mathrm{diag} \left( w_+^{(t)\top} X - w_+^{(t)\top} X p_l^{(+,t)} \right) X^\top v_1 \bigg| X_{k,s,-s} \right]\bigg|_{t=T_1}$$

$$+ \mathbb{E} \left[ g^{(t)} y \dot{\sigma}_+^{(t)} \sum_{l=1}^L v_1^\top W_{K+}^{(t)\top} K_+^{(t)} \mathrm{diag} \left( X^\top w_+ - w_+^\top X p_l^{(+,t)} \right) p_l^{(+,t)} x_l^\top c_1 \bigg| X_{k,s,-s} \right]\bigg|_{t=T_1}$$

$$= \frac{1}{3} g^{(T_1)} 2 \left( c_1^\top W_{Q+}^{(T_1)\top} W_{Q+}^{(T_1)} c_1 - c_1^\top W_{Q+}^{(T_1)\top} W_{Q+}^{(T_1)} c_2 \right) \left( -w_+^{(T_1)\top} X p_l^{(+,T_1)} \right)$$

$$+ \frac{1}{3} g^{(T_1)} 2 \left( v_1^\top W_{K+}^{(T_1)\top} W_{K+}^{(T_1)} v_1 - v_1^\top W_{K+}^{(T_1)\top} W_{K+}^{(T_1)} v_2 \right) \left( -w_+^{(T_1)\top} X p_l^{(+,T_1)} \right)$$

$$= \frac{1}{3} g^{(T_1)} 2 \left( c_1^\top W_{Q+}^{(T_1)\top} W_{Q+}^{(T_1)} c_1 - c_1^\top W_{Q+}^{(T_1)\top} W_{Q+}^{(T_1)} c_2 \right) \left( -\frac{1}{3} w_+^{(T_1)\top} c_1 \right)$$

$$+ \frac{1}{3} g^{(T_1)} 2 \left( v_1^\top W_{K+}^{(T_1)\top} W_{K+}^{(T_1)} v_1 - v_1^\top W_{K+}^{(T_1)\top} W_{K+}^{(T_1)} v_2 \right) \left( -\frac{1}{3} w_+^{(T_1)\top} c_1 \right)$$

$$\left. \frac{\mathrm{d} v_2^\top W_{K+}^{(t)\top} W_{Q+}^{(t)} c_1}{\mathrm{d}t} \right|_{t=T_1}$$

$$= \frac{1}{3} g^{(T_1)} 2 \left( -c_1^\top W_{Q+}^{(T_1)\top} W_{Q+}^{(T_1)} c_1 + c_1^\top W_{Q+}^{(T_1)\top} W_{Q+}^{(T_1)} c_2 \right) \left( -w_+^{(T_1)\top} X p_l^{(+,T_1)} \right)$$

$$+ \frac{1}{3} g^{(T_1)} 2 \left( v_2^\top W_{K+}^{(T_1)\top} W_{K+}^{(T_1)} v_1 - v_2^\top W_{K+}^{(T_1)\top} W_{K+}^{(T_1)} v_2 \right) \left( -w_+^{(T_1)\top} X p_l^{(+,T_1)} \right)$$

$$= \frac{1}{3} g^{(T_1)} 2 \left( -c_1^\top W_{Q+}^{(T_1)\top} W_{Q+}^{(T_1)} c_1 + c_1^\top W_{Q+}^{(T_1)\top} W_{Q+}^{(T_1)} c_2 \right) \left( -\frac{1}{3} w_+^{(T_1)\top} c_1 \right)$$

$$+ \frac{1}{3} g^{(T_1)} 2 \left( v_2^\top W_{K+}^{(T_1)\top} W_{K+}^{(T_1)} v_1 - v_2^\top W_{K+}^{(T_1)\top} W_{K+}^{(T_1)} v_2 \right) \left( -\frac{1}{3} w_+^{(T_1)\top} c_1 \right)$$

$$\left. \frac{\mathrm{d} v_1^\top W_{K+}^{(t)\top} W_{Q+}^{(t)} v_1}{\mathrm{d}t} \right|_{t=T_1}$$

$$= \mathbb{E} \left[ g^{(t)} y \dot{\sigma}_+^{(t)} \sum_{l=1}^L v_1^\top W_{Q+}^{(t)\top} q_{+l}^{(t)} p_l^{(+,t)\top} \mathrm{diag} \left( w_+^{(t)\top} X - w_+^{(t)\top} X p_l^{(+,t)} \right) X^\top v_1 \,\middle|\, X_{k,s,-s} \right]\Bigg|_{t=T_1}$$

$$+ \mathbb{E} \left[ g^{(t)} y \dot{\sigma}_+^{(t)} \sum_{l=1}^L v_1^\top W_{K+}^{(t)\top} K_+^{(t)} \mathrm{diag} \left( X^\top w_+ - w_+^\top X p_l^{(+,t)} \right) p_l^{(+,t)} x_l^\top v_1 \,\middle|\, X_{k,s,-s} \right]\Bigg|_{t=T_1}$$

$$= \frac{1}{3} g^{(T_1)} 2 \left( v_1^\top W_{Q+}^{(T_1)\top} W_{Q+}^{(T_1)} c_1 - v_1^\top W_{Q+}^{(T_1)\top} W_{Q+}^{(T_1)} c_2 \right) \left( -w_+^{(T_1)\top} X p_l^{(+,T_1)} \right)$$

$$+ \frac{1}{3} g^{(T_1)} 2 \left( v_1^\top W_{K+}^{(T_1)\top} W_{K+}^{(T_1)} c_1 - v_1^\top W_{K+}^{(T_1)\top} W_{K+}^{(T_1)} c_2 \right) \left( w_+^{(T_1)\top} c_1 - w_+^{(T_1)\top} X p_l^{(+,T_1)} \right)$$

$$= \frac{1}{3} g^{(T_1)} 2 \left( v_1^\top W_{Q+}^{(T_1)\top} W_{Q+}^{(T_1)} c_1 - v_1^\top W_{Q+}^{(T_1)\top} W_{Q+}^{(T_1)} c_2 \right) \left( -\frac{1}{3} w_+^{(T_1)\top} c_1 \right)$$

$$+ \frac{1}{3} g^{(T_1)} 2 \left( v_1^\top W_{K+}^{(T_1)\top} W_{K+}^{(T_1)} c_1 - v_1^\top W_{K+}^{(T_1)\top} W_{K+}^{(T_1)} c_2 \right) \left( \frac{2}{3} w_+^{(T_1)\top} c_1 \right)$$

$$\left. \frac{\mathrm{d} v_1^\top W_{K+}^{(t)\top} W_{Q+}^{(t)} v_2}{\mathrm{d}t} \right|_{t=T_1}$$

$$= \frac{1}{3} g^{(T_1)} 2 \left( v_2^\top W_{Q+}^{(T_1)\top} W_{Q+}^{(T_1)} c_1 - v_2^\top W_{Q+}^{(T_1)\top} W_{Q+}^{(T_1)} c_2 \right) \left( -w_+^{(T_1)\top} X p_l^{(+,T_1)} \right)$$

$$+ \frac{1}{3} g^{(T_1)} 2 \left( -v_1^\top W_{K+}^{(T_1)\top} W_{K+}^{(T_1)} c_1 + v_1^\top W_{K+}^{(T_1)\top} W_{K+}^{(T_1)} c_2 \right) \left( w_+^{(T_1)\top} c_1 - w_+^{(T_1)\top} X p_l^{(+,T_1)} \right)$$

$$= \frac{1}{3} g^{(T_1)} 2 \left( v_2^\top W_{Q+}^{(T_1)\top} W_{Q+}^{(T_1)} c_1 - v_2^\top W_{Q+}^{(T_1)\top} W_{Q+}^{(T_1)} c_2 \right) \left( -\frac{1}{3} w_+^{(T_1)\top} c_1 \right)$$

$$+ \frac{1}{3} g^{(T_1)} 2 \left( -v_1^\top W_{K+}^{(T_1)\top} W_{K+}^{(T_1)} c_1 + v_1^\top W_{K+}^{(T_1)\top} W_{K+}^{(T_1)} c_2 \right) \left( \frac{2}{3} w_+^{(T_1)\top} c_1 \right)$$

### E.5.2 THE KEY SELF-SCORES

$$\left. \frac{\mathrm{d} v_1^\top W_{K+}^{(t)\top} W_{K+}^{(t)} v_2}{\mathrm{d}t} \right|_{t=T_1}$$

$$= \mathbb{E}\left[g^{(t)}y\dot{\sigma}_+^{(t)}\sum_{l=1}^{L}v_1^\top W_{K+}^{(t)\top}q_{+l}^{(t)}p_l^{(+,t)\top}\mathrm{diag}\left(w_+^\top X - w_+^\top X p_l^{(+,t)}\right)X^\top v_2 \,\middle|\, X_{k,s,-s}\right]\Bigg|_{t=T_1}$$

$$+ \mathbb{E}\left[g^{(t)}y\dot{\sigma}_+^{(t)}\sum_{l=1}^{L}v_2^\top W_{K+}^{(t)\top}q_{+l}^{(t)}p_l^{(+,t)\top}\mathrm{diag}\left(w_+^\top X - w_+^\top X p_l^{(+,t)}\right)X^\top v_1 \,\middle|\, X_{k,s,-s}\right]\Bigg|_{t=T_1}$$

$$= \frac{1}{3}g^{(T_1)}2\left(v_1^\top W_{K+}^{(T_1)\top}W_{Q+}^{(T_1)}c_1 - v_1^\top W_{K+}^{(T_1)\top}W_{Q+}^{(T_1)}c_2\right)w_+^{(T_1)\top}Xp_l^{(+,T_1)}$$

$$+ \frac{1}{3}g^{(T_1)}2\left(-v_2^\top W_{K+}^{(T_1)\top}W_{Q+}^{(T_1)}c_1 + v_2^\top W_{K+}^{(T_1)\top}W_{Q+}^{(T_1)}c_2\right)w_+^{(T_1)\top}Xp_l^{(+,T_1)}$$

$$= \frac{1}{3}g^{(T_1)}2\left(v_1^\top W_{K+}^{(T_1)\top}W_{Q+}^{(T_1)}c_1 - v_1^\top W_{K+}^{(T_1)\top}W_{Q+}^{(T_1)}c_2\right)\frac{1}{3}w_+^{(T_1)\top}c_1$$

$$+ \frac{1}{3}g^{(T_1)}2\left(-v_2^\top W_{K+}^{(T_1)\top}W_{Q+}^{(T_1)}c_1 + v_2^\top W_{K+}^{(T_1)\top}W_{Q+}^{(T_1)}c_2\right)\frac{1}{3}w_+^{(T_1)\top}c_1$$

$$\frac{\mathrm{d}v_1^\top W_{K+}^{(t)\top}W_{K+}^{(t)}v_1}{\mathrm{d}t}\Bigg|_{t=T_1}$$

$$= \frac{1}{3}g^{(T_1)}4\left(v_1^\top W_{K+}^{(T_1)\top}W_{Q+}^{(T_1)}c_1 - v_1^\top W_{K+}^{(T_1)\top}W_{Q+}^{(T_1)}c_2\right)\left(-w_+^{(T_1)\top}Xp_l^{(+,T_1)}\right)$$

$$= \frac{1}{3}g^{(T_1)}4\left(v_1^\top W_{K+}^{(T_1)\top}W_{Q+}^{(T_1)}c_1 - v_1^\top W_{K+}^{(T_1)\top}W_{Q+}^{(T_1)}c_2\right)\left(-\frac{1}{3}w_+^{(T_1)\top}c_1\right)$$

$$\frac{\mathrm{d}c_1^\top W_{K+}^{(t)\top}W_{K+}^{(t)}c_2}{\mathrm{d}t}\Bigg|_{t=T_1}$$

$$= \mathbb{E}\left[g^{(t)}y\dot{\sigma}_+^{(t)}\sum_{l=1}^{L}c_1^\top W_{K+}^{(t)\top}q_{+l}^{(t)}p_l^{(+,t)\top}\mathrm{diag}\left(w_+^\top X - w_+^\top X p_l^{(+,t)}\right)X^\top c_2 \,\middle|\, X_{k,s,-s}\right]\Bigg|_{t=T_1}$$

$$+ \mathbb{E}\left[g^{(t)}y\dot{\sigma}_+^{(t)}\sum_{l=1}^{L}c_2^\top W_{K+}^{(t)\top}q_{+l}^{(t)}p_l^{(+,t)\top}\mathrm{diag}\left(w_+^\top X - w_+^\top X p_l^{(+,t)}\right)X^\top c_1 \,\middle|\, X_{k,s,-s}\right]\Bigg|_{t=T_1}$$

$$= \frac{1}{3}g^{(T_1)}2\left(c_1^\top W_{K+}^{(T_1)\top}W_{Q+}^{(T_1)}v_2 - c_1^\top W_{K+}^{(T_1)\top}W_{Q+}^{(T_1)}v_1\right)\left(w_+^{(T_1)\top}c_2 - w_+^{(T_1)\top}Xp_l^{(+,T_1)}\right)$$

$$+ \frac{1}{3}g^{(T_1)}2\left(c_2^\top W_{K+}^{(T_1)\top}W_{Q+}^{(T_1)}v_1 - c_2^\top W_{K+}^{(T_1)\top}W_{Q+}^{(T_1)}v_2\right)\left(w_+^{(T_1)\top}c_1 - w_+^{(T_1)\top}Xp_l^{(+,T_1)}\right)$$

$$= \frac{1}{3}g^{(T_1)}2\left(c_1^\top W_{K+}^{(T_1)\top}W_{Q+}^{(T_1)}v_2 - c_1^\top W_{K+}^{(T_1)\top}W_{Q+}^{(T_1)}v_1\right)\left(\frac{2}{3}w_+^{(T_1)\top}c_2\right)$$

$$+ \frac{1}{3}g^{(T_1)}2\left(c_2^\top W_{K+}^{(T_1)\top}W_{Q+}^{(T_1)}v_1 - c_2^\top W_{K+}^{(T_1)\top}W_{Q+}^{(T_1)}v_2\right)\left(\frac{2}{3}w_+^{(T_1)\top}c_1\right)$$

$$\frac{\mathrm{d}c_1^\top W_{K+}^{(t)\top}W_{K+}^{(t)}c_1}{\mathrm{d}t}\Bigg|_{t=T_1}$$

$$= \frac{1}{3}g^{(T_1)}4\left(c_1^\top W_{K+}^{(T_1)\top}W_{Q+}^{(T_1)}v_1 - c_1^\top W_{K+}^{(T_1)\top}W_{Q+}^{(T_1)}v_2\right)\left(w_+^{(T_1)\top}c_1 - w_+^{(T_1)\top}Xp_l^{(+,T_1)}\right)$$

$$= \frac{1}{3}g^{(T_1)}4\left(c_1^\top W_{K+}^{(T_1)\top}W_{Q+}^{(T_1)}v_1 - c_1^\top W_{K+}^{(T_1)\top}W_{Q+}^{(T_1)}v_2\right)\left(\frac{2}{3}w_+^{(T_1)\top}c_1\right)$$

$$\frac{\mathrm{d}c_1^\top W_{K+}^{(t)\top}W_{K+}^{(t)}v_1}{\mathrm{d}t}\Bigg|_{t=T_1}$$

$$= \mathbb{E}\left[g^{(t)}y\dot{\sigma}_+^{(t)}\sum_{l=1}^{L}c_1^\top W_{K+}^{(t)\top}q_{+l}^{(t)}p_l^{(+,t)\top}\mathrm{diag}\left(w_+^\top X - w_+^\top X p_l^{(+,t)}\right)X^\top v_1 \,\middle|\, X_{k,s,-s}\right]\Bigg|_{t=T_1}$$

$$+ \mathbb{E}\left[g^{(t)}y\dot{\sigma}_+^{(t)}\sum_{l=1}^{L}v_1^\top W_{K+}^{(t)\top}q_{+l}^{(t)}p_l^{(+,t)\top}\mathrm{diag}\left(w_+^\top X - w_+^\top X p_l^{(+,t)}\right)X^\top c_1\Bigg|X_{k,s,-s}\right]\Bigg|_{t=T_1}$$

$$= \frac{1}{3}g^{(T_1)}2\left(c_1^\top W_{K+}^{(T_1)\top}W_{Q+}^{(T_1)}c_1 - c_1^\top W_{K+}^{(T_1)\top}W_{Q+}^{(T_1)}c_2\right)\left(-w_+^{(T_1)\top}Xp_l^{(+,T_1)}\right)$$

$$+ \frac{1}{3}g^{(T_1)}2\left(v_1^\top W_{K+}^{(T_1)\top}W_{Q+}^{(T_1)}v_1 - v_1^\top W_{K+}^{(T_1)\top}W_{Q+}^{(T_1)}v_2\right)\left(w_+^{(T_1)\top}c_1 - w_+^{(T_1)\top}Xp_l^{(+,T_1)}\right)$$

$$= \frac{1}{3}g^{(T_1)}2\left(c_1^\top W_{K+}^{(T_1)\top}W_{Q+}^{(T_1)}c_1 - c_1^\top W_{K+}^{(T_1)\top}W_{Q+}^{(T_1)}c_2\right)\left(-\frac{1}{3}w_+^{(T_1)\top}Xc_1\right)$$

$$+ \frac{1}{3}g^{(T_1)}2\left(v_1^\top W_{K+}^{(T_1)\top}W_{Q+}^{(T_1)}v_1 - v_1^\top W_{K+}^{(T_1)\top}W_{Q+}^{(T_1)}v_2\right)\left(\frac{2}{3}w_+^{(T_1)\top}c_1\right)$$

$$\frac{dc_2^\top W_{K+}^{(t)\top}W_{K+}^{(t)}v_1}{dt}\Bigg|_{t=T_1}$$

$$= \frac{1}{3}g^{(T_1)}2\left(c_2^\top W_{K+}^{(T_1)\top}W_{Q+}^{(T_1)}c_1 - c_2^\top W_{K+}^{(T_1)\top}W_{Q+}^{(T_1)}c_2\right)\left(-w_+^{(T_1)\top}Xp_l^{(+,T_1)}\right)$$

$$+ \frac{1}{3}g^{(T_1)}2\left(v_1^\top W_{K+}^{(T_1)\top}W_{Q+}^{(T_1)}v_2 - v_1^\top W_{K+}^{(T_1)\top}W_{Q+}^{(T_1)}v_1\right)\left(w_+^{(T_1)\top}c_2 - w_+^{(T_1)\top}Xp_l^{(+,T_1)}\right)$$

$$= \frac{1}{3}g^{(T_1)}2\left(c_2^\top W_{K+}^{(T_1)\top}W_{Q+}^{(T_1)}c_1 - c_2^\top W_{K+}^{(T_1)\top}W_{Q+}^{(T_1)}c_2\right)\left(-\frac{1}{3}w_+^{(T_1)\top}c_1\right)$$

$$+ \frac{1}{3}g^{(T_1)}2\left(v_1^\top W_{K+}^{(T_1)\top}W_{Q+}^{(T_1)}v_2 - v_1^\top W_{K+}^{(T_1)\top}W_{Q+}^{(T_1)}v_1\right)\left(\frac{2}{3}w_+^{(T_1)\top}c_2\right)$$

### E.5.3 THE QUERY SELF-SCORES

$$\frac{dv_1^\top W_{Q+}^{(t)\top}W_{Q+}^{(t)}v_2}{dt}\Bigg|_{t=T_1}$$

$$= \mathbb{E}\left[g^{(t)}y\dot{\sigma}_+^{(t)}\sum_{l=1}^{L}v_1^\top W_{Q+}^{(t)\top}K_+^{(t)}\mathrm{diag}\left(X^\top w_+ - w_+^\top X p_l^{(+,t)}\right)p_l^{(+,t)}x_l^\top v_2\Bigg|X_{k,s,-s}\right]\Bigg|_{t=T_1}$$

$$+ \mathbb{E}\left[g^{(t)}y\dot{\sigma}_+^{(t)}\sum_{l=1}^{L}v_2^\top W_{Q+}^{(t)\top}K_+^{(t)}\mathrm{diag}\left(X^\top w_+ - w_+^\top X p_l^{(+,t)}\right)p_l^{(+,t)}x_l^\top v_1\Bigg|X_{k,s,-s}\right]\Bigg|_{t=T_1}$$

$$= \frac{1}{3}g^{(T_1)}2\left(-v_1^\top W_{Q+}^{(T_1)\top}W_{K+}^{(T_1)}c_1 + v_1^\top W_{Q+}^{(T_1)\top}W_{K+}^{(T_1)}c_2\right)\left(w_+^{(T_1)\top}c_1 - w_+^{(T_1)\top}Xp_l^{(+,T_1)}\right)$$

$$+ \frac{1}{3}g^{(T_1)}2\left(v_2^\top W_{Q+}^{(T_1)\top}W_{K+}^{(T_1)}c_1 - v_2^\top W_{Q+}^{(T_1)\top}W_{K+}^{(T_1)}c_2\right)\left(w_+^{(T_1)\top}c_1 - w_+^{(T_1)\top}Xp_l^{(+,T_1)}\right)$$

$$= \frac{1}{3}g^{(T_1)}2\left(-v_1^\top W_{Q+}^{(T_1)\top}W_{K+}^{(T_1)}c_1 + v_1^\top W_{Q+}^{(T_1)\top}W_{K+}^{(T_1)}c_2\right)\left(\frac{2}{3}w_+^{(T_1)\top}c_1\right)$$

$$+ \frac{1}{3}g^{(T_1)}2\left(v_2^\top W_{Q+}^{(T_1)\top}W_{K+}^{(T_1)}c_1 - v_2^\top W_{Q+}^{(T_1)\top}W_{K+}^{(T_1)}c_2\right)\left(\frac{2}{3}w_+^{(T_1)\top}c_1\right)$$

$$\frac{dv_1^\top W_{Q+}^{(t)\top}W_{Q+}^{(t)}v_1}{dt}\Bigg|_{t=T_1}$$

$$= \frac{1}{3}g^{(T_1)}2\left(v_1^\top W_{Q+}^{(T_1)\top}W_{K+}^{(T_1)}c_1 - v_1^\top W_{Q+}^{(T_1)\top}W_{K+}^{(T_1)}c_2\right)\left(w_+^{(T_1)\top}c_1 - w_+^{(T_1)\top}Xp_l^{(+,T_1)}\right)$$

$$+ \frac{1}{3}g^{(T_1)}2\left(v_1^\top W_{Q+}^{(T_1)\top}W_{K+}^{(T_1)}c_1 - v_1^\top W_{Q+}^{(T_1)\top}W_{K+}^{(T_1)}c_2\right)\left(w_+^{(T_1)\top}c_1 - w_+^{(T_1)\top}Xp_l^{(+,T_1)}\right)$$

$$= \frac{1}{3}g^{(T_1)}2\left(v_1^\top W_{Q+}^{(T_1)\top}W_{K+}^{(T_1)}c_1 - v_1^\top W_{Q+}^{(T_1)\top}W_{K+}^{(T_1)}c_2\right)\left(\frac{2}{3}w_+^{(T_1)\top}c_1\right)$$

$$+ \frac{1}{3}g^{(T_1)}2\left(v_1^\top W_{Q+}^{(T_1)\top}W_{K+}^{(T_1)}c_1 - v_1^\top W_{Q+}^{(T_1)\top}W_{K+}^{(T_1)}c_2\right)\left(\frac{2}{3}w_+^{(T_1)\top}c_1\right)$$

$$\left.\frac{dc_1^\top W_{Q+}^{(t)\top}W_{Q+}^{(t)}c_2}{dt}\right|_{t=T_1}$$

$$= \mathbb{E}\left[\left.g^{(t)}y\dot\sigma_+^{(t)}\sum_{l=1}^{L}c_1^\top W_{Q+}^{(t)\top}K_+^{(t)}\mathrm{diag}\left(X^\top w_+ - w_+^\top X p_l^{(+,t)}\right)p_l^{(+,t)}x_l^\top c_2\right|X_{k,s,-s}\right]\Bigg|_{t=T_1}$$

$$+ \mathbb{E}\left[\left.g^{(t)}y\dot\sigma_+^{(t)}\sum_{l=1}^{L}c_2^\top W_{Q+}^{(t)\top}K_+^{(t)}\mathrm{diag}\left(X^\top w_+ - w_+^\top X p_l^{(+,t)}\right)p_l^{(+,t)}x_l^\top c_1\right|X_{k,s-s}\right]\Bigg|_{t=T_1}$$

$$= \frac{1}{3}g^{(T_1)}2\left(c_1^\top W_{Q+}^{(T_1)\top}W_{K+}^{(T_1)}v_2 - c_1^\top W_{Q+}^{(T_1)\top}W_{K+}^{(T_1)}v_1\right)\left(-w_+^{(T_1)\top}X p_l^{(+,T_1)}\right)$$

$$+ \frac{1}{3}g^{(T_1)}2\left(c_2^\top W_{Q+}^{(T_1)\top}W_{K+}^{(T_1)}v_1 - c_2^\top W_{Q+}^{(T_1)\top}W_{K+}^{(T_1)}v_2\right)\left(-w_+^{(T_1)\top}X p_l^{(+,T_1)}\right)$$

$$= \frac{1}{3}g^{(T_1)}2\left(c_1^\top W_{Q+}^{(T_1)\top}W_{K+}^{(T_1)}v_2 - c_1^\top W_{Q+}^{(T_1)\top}W_{K+}^{(T_1)}v_1\right)\left(-\frac{1}{3}w_+^{(T_1)\top}c_1\right)$$

$$+ \frac{1}{3}g^{(T_1)}2\left(c_2^\top W_{Q+}^{(T_1)\top}W_{K+}^{(T_1)}v_1 - c_2^\top W_{Q+}^{(T_1)\top}W_{K+}^{(T_1)}v_2\right)\left(-\frac{1}{3}w_+^{(T_1)\top}c_1\right)$$

$$\left.\frac{dc_1^\top W_{Q+}^{(t)\top}W_{Q+}^{(t)}c_1}{dt}\right|_{t=T_1}$$

$$= \frac{1}{3}g^{(T_1)}2\left(-c_1^\top W_{Q+}^{(T_1)\top}W_{K+}^{(T_1)}v_2 + c_1^\top W_{Q+}^{(T_1)\top}W_{K+}^{(T_1)}v_1\right)\left(-w_+^{(T_1)\top}X p_l^{(+,T_1)}\right)$$

$$+ \frac{1}{3}g^{(T_1)}2\left(c_1^\top W_{Q+}^{(T_1)\top}W_{K+}^{(T_1)}v_1 - c_1^\top W_{Q+}^{(T_1)\top}W_{K+}^{(T_1)}v_2\right)\left(-w_+^{(T_1)\top}X p_l^{(+,T_1)}\right)$$

$$= \frac{1}{3}g^{(T_1)}2\left(-c_1^\top W_{Q+}^{(T_1)\top}W_{K+}^{(T_1)}v_2 + c_1^\top W_{Q+}^{(T_1)\top}W_{K+}^{(T_1)}v_1\right)\left(-\frac{1}{3}w_+^{(T_1)\top}c_1\right)$$

$$+ \frac{1}{3}g^{(T_1)}2\left(c_1^\top W_{Q+}^{(T_1)\top}W_{K+}^{(T_1)}v_1 - c_1^\top W_{Q+}^{(T_1)\top}W_{K+}^{(T_1)}v_2\right)\left(-\frac{1}{3}w_+^{(T_1)\top}c_1\right)$$

$$\left.\frac{dc_1^\top W_{Q+}^{(t)\top}W_{Q+}^{(t)}v_2}{dt}\right|_{t=T_1}$$

$$= \mathbb{E}\left[\left.g^{(t)}y\dot\sigma_+^{(t)}\sum_{l=1}^{L}c_1^\top W_{Q+}^{(t)\top}K_+^{(t)}\mathrm{diag}\left(X^\top w_+ - w_+^\top X p_l^{(+,t)}\right)p_l^{(+,t)}x_l^\top v_2\right|X_{k,s,-s}\right]\Bigg|_{t=T_1}$$

$$+ \mathbb{E}\left[\left.g^{(t)}y\dot\sigma_+^{(t)}\sum_{l=1}^{L}v_2^\top W_{Q+}^{(t)\top}K_+^{(t)}\mathrm{diag}\left(X^\top w_+ - w_+^\top X p_l^{(+,t)}\right)p_l^{(+,t)}x_l^\top c_1\right|X_{k,s,-s}\right]\Bigg|_{t=T_1}$$

$$= \frac{1}{3}g^{(T_1)}2\left(c_1^\top W_{Q+}^{(T_1)\top}W_{K+}^{(T_1)}c_2 - c_1^\top W_{Q+}^{(T_1)\top}W_{K+}^{(T_1)}c_1\right)\left(w_+^{(T_1)\top}c_1 - w_+^{(T_1)\top}X p_l^{(+,T-1)}\right)$$

$$+ \frac{1}{3}g^{(T_1)}2\left(v_2^\top W_{Q+}^{(T_1)\top}W_{K+}^{(T_1)}v_1 - v_2^\top W_{Q+}^{(T_1)\top}W_{K+}^{(T_1)}v_2\right)\left(-w_+^{(T_1)\top}X p_l^{(+,T_1)}\right)$$

$$= \frac{1}{3}g^{(T_1)}2\left(c_1^\top W_{Q+}^{(T_1)\top}W_{K+}^{(T_1)}c_2 - c_1^\top W_{Q+}^{(T_1)\top}W_{K+}^{(T_1)}c_1\right)\left(\frac{2}{3}w_+^{(T_1)\top}c_1\right)$$

$$+ \frac{1}{3}g^{(T_1)}2\left(v_2^\top W_{Q+}^{(T_1)\top}W_{K+}^{(T_1)}v_1 - v_2^\top W_{Q+}^{(T_1)\top}W_{K+}^{(T_1)}v_2\right)\left(-\frac{1}{3}w_+^{(T_1)\top}c_1\right)$$

$$\left.\frac{dc_1^\top W_{Q+}^{(t)\top}W_{Q+}^{(t)}v_1}{dt}\right|_{t=T_1}$$

$$= \frac{1}{3} g^{(T_1)} 2 \left( c_1^\top W_{Q+}^{(T_1)\top} W_{K+}^{(T_1)} c_1 - c_1^\top W_{Q+}^{(T_1)\top} W_{K+}^{(T_1)} c_2 \right) \left( w_+^{(T_1)\top} c_1 - w_+^{(T_1)\top} X p_l^{(+,T_1)} \right)$$

$$+ \frac{1}{3} g^{(T_1)} 2 \left( v_1^\top W_{Q+}^{(T_1)\top} W_{K+}^{(T_1)} v_1 - v_1^\top W_{Q+}^{(T_1)\top} W_{K+}^{(T_1)} v_2 \right) \left( -w_+^{(T_1)\top} X p_l^{(+,T_1)} \right)$$

$$= \frac{1}{3} g^{(T_1)} 2 \left( c_1^\top W_{Q+}^{(T_1)\top} W_{K+}^{(T_1)} c_1 - c_1^\top W_{Q+}^{(T_1)\top} W_{K+}^{(T_1)} c_2 \right) \left( \frac{2}{3} w_+^{(T_1)\top} c_1 \right)$$

$$+ \frac{1}{3} g^{(T_1)} 2 \left( v_1^\top W_{Q+}^{(T_1)\top} W_{K+}^{(T_1)} v_1 - v_1^\top W_{Q+}^{(T_1)\top} W_{K+}^{(T_1)} v_2 \right) \left( -\frac{1}{3} w_+^{(T_1)\top} c_1 \right)$$

### E.6 SUM OF PARTIAL GRADIENTS

$$\frac{\mathrm{d} W_{Q+}^{(t)} v_1}{\mathrm{d} t} + \frac{\mathrm{d} W_{Q+}^{(t)} v_2}{\mathrm{d} t} :$$

$$\mathbb{E} \left[ g^{(t)} y \dot\sigma_+^{(t)} \sum_{l=1}^L K_+^{(t)} \mathrm{diag} \left( X^\top w_+^{(t)} - w_+^{(t)\top} X p_l^{(+,t)} \right) p_l^{(+,t)} x_l^\top v_1 \,\middle|\, X_{1,+,-} \right]$$

$$+ \mathbb{E} \left[ g^{(t)} y \dot\sigma_+^{(t)} \sum_{l=1}^L K_+^{(t)} \mathrm{diag} \left( X^\top w_+^{(t)} - w_+^{(t)\top} X p_l^{(+,t)} \right) p_l^{(+,t)} x_l^\top v_2 \,\middle|\, X_{2,+,-} \right]$$

$$= g^{(t)} \left( W_{K+}^{(t)} v_1 \left( w_+^{(t)\top} v_1 - w_+^{(t)\top} X_{1,+,-} p_{q\leftarrow v_1}^{(+,t)} \right) p_{q\leftarrow v_1, k\leftarrow v_1}^{(+,t)} \right.$$

$$+ W_{K+}^{(t)} c_1 \left( w_+^{(t)\top} c_1 - w_+^{(t)\top} X_{1,+,-} p_{q\leftarrow v_1}^{(+,t)} \right) p_{q\leftarrow v_1, k\leftarrow c_1}^{(+,t)}$$

$$+ W_{K+}^{(t)} (-v_2) \left( w_+^{(t)\top} (-v_2) - w_+^{(t)\top} X_{1,+,-} p_{q\leftarrow v_1}^{(+,t)} \right) p_{q\leftarrow v_1, k\leftarrow -v_2}^{(+,t)}$$

$$+ W_{K+}^{(t)} v_2 \left( w_+^{(t)\top} v_2 - w_+^{(t)\top} X_{2,+,-} p_{q\leftarrow v_2}^{(+,t)} \right) p_{q\leftarrow v_2, k\leftarrow v_2}^{(+,t)}$$

$$+ W_{K+}^{(t)} c_2 \left( w_+^{(t)\top} c_2 - w_+^{(t)\top} X_{2,+,-} p_{q\leftarrow v_2}^{(+,t)} \right) p_{q\leftarrow v_2, k\leftarrow c_2}^{(+,t)}$$

$$+ \left. W_{K+}^{(t)} (-v_1) \left( w_+^{(t)\top} (-v_1) - w_+^{(t)\top} X_{2,+,-} p_{q\leftarrow v_2}^{(+,t)} \right) p_{q\leftarrow v_2, k\leftarrow -v_1}^{(+,t)} \right)$$

$$\frac{\mathrm{d} W_{Q+}^{(t)} (c_1 + c_2)}{\mathrm{d} t} :$$

$$\mathbb{E} \left[ g^{(t)} y \dot\sigma_+^{(t)} \sum_{l=1}^L K_+^{(t)} \mathrm{diag} \left( X^\top w_+ - w_+^\top X p_l^{(+,t)} \right) p_l^{(+,t)} x_l^\top (c_1 + c_2) \,\middle|\, X_{k,+,-} \right]$$

$$= g^{(t)} \left( W_{K+}^{(t)} v_1 \left( w_+^{(t)\top} v_1 - w_+^{(t)\top} X_{1,+,-} p_{q\leftarrow c_1}^{(+,t)} \right) p_{q\leftarrow c_1, k\leftarrow v_1}^{(+,t)} \right.$$

$$+ W_{K+}^{(t)} c_1 \left( w_+^{(t)\top} c_1 - w_+^{(t)\top} X_{1,+,-} p_{q\leftarrow c_1}^{(+,t)} \right) p_{q\leftarrow c_1, k\leftarrow c_1}^{(+,t)}$$

$$+ W_{K+}^{(t)} (-v_2) \left( w_+^{(t)\top} (-v_2) - w_+^{(t)\top} X_{1,+,-} p_{q\leftarrow c_1}^{(+,t)} \right) p_{q\leftarrow c_1, k\leftarrow -v_2}^{(+,t)}$$

$$+ W_{K+}^{(t)} v_2 \left( w_+^{(t)\top} v_2 - w_+^{(t)\top} X_{2,+,-} p_{q\leftarrow c_2}^{(+,t)} \right) p_{q\leftarrow c_2, k\leftarrow v_2}^{(+,t)}$$

$$+ W_{K+}^{(t)} c_2 \left( w_+^{(t)\top} c_2 - w_+^{(t)\top} X_{2,+,-} p_{q\leftarrow c_2}^{(+,t)} \right) p_{q\leftarrow c_2, k\leftarrow c_2}^{(+,t)}$$

$$+ \left. W_{K+}^{(t)} (-v_1) \left( w_+^{(t)\top} (-v_1) - w_+^{(t)\top} X_{2,+,-} p_{q\leftarrow c_2}^{(+,t)} \right) p_{q\leftarrow c_2, k\leftarrow -v_1}^{(+,t)} \right)$$

$$\frac{\mathrm{d} W_{K+}^{(t)} v_1}{\mathrm{d} t} + \frac{\mathrm{d} W_{K+}^{(t)} v_2}{\mathrm{d} t} :$$

$$\mathbb{E}\left[g^{(t)}y\dot{\sigma}_+^{(t)}\sum_{l=1}^{L}q_{+l}^{(t)}p_l^{(+,t)\top}\text{diag}\left(w_+^\top X - w_+^\top X p_l^{(+,t)}\right)X^\top v_1 \,\bigg|\, X_{1,+,-}\right]$$

$$+\,\mathbb{E}\left[g^{(t)}y\dot{\sigma}_+^{(t)}\sum_{l=1}^{L}q_{+l}^{(t)}p_l^{(+,t)\top}\text{diag}\left(w_+^\top X - w_+^\top X p_l^{(+,t)}\right)X^\top v_2 \,\bigg|\, X_{2,+,-}\right]$$

$$=g^{(t)}\Bigg(W_{Q+}^{(t)}v_1 p_{q\leftarrow v_1, k\leftarrow v_1}^{(+,t)}\left(w_+^{(t)\top}v_1 - w_+^{(t)\top}X_{1,+,-}p_{q\leftarrow v_1}^{(+,t)}\right)$$

$$+\,W_{Q+}^{(t)}c_1 p_{q\leftarrow c_1, k\leftarrow v_1}^{(+,t)}\left(w_+^{(t)\top}v_1 - w_+^{(t)\top}X_{1,+,-}p_{q\leftarrow c_1}^{(+,t)}\right)$$

$$+\,W_{Q+}^{(t)}(-v_2) p_{q\leftarrow -v_2, k\leftarrow v_1}^{(+,t)}\left(w_+^{(t)\top}v_1 - w_+^{(t)\top}X_{1,+,-}p_{q\leftarrow -v_2}^{(+,t)}\right)$$

$$+\,W_{Q+}^{(t)}v_2 p_{q\leftarrow v_2, k\leftarrow v_2}^{(+,t)}\left(w_+^{(t)\top}v_2 - w_+^{(t)\top}X_{2,+,-}p_{q\leftarrow v_2}^{(+,t)}\right)$$

$$+\,W_{Q+}^{(t)}c_2 p_{q\leftarrow c_2, k\leftarrow v_2}^{(+,t)}\left(w_+^{(t)\top}v_2 - w_+^{(t)\top}X_{2,+,-}p_{q\leftarrow c_2}^{(+,t)}\right)$$

$$+\,W_{Q+}^{(t)}(-v_1) p_{q\leftarrow -v_1, k\leftarrow v_2}^{(+,t)}\left(w_+^{(t)\top}v_2 - w_+^{(t)\top}X_{2,+,-}p_{q\leftarrow -v_1}^{(+,t)}\right)\Bigg)$$

$$\frac{dW_{K+}^{(t)}(c_1+c_2)}{dt}:$$

$$\mathbb{E}\left[g^{(t)}y\dot{\sigma}_+^{(t)}\sum_{l=1}^{L}q_{+l}^{(t)}p_l^{(+,t)\top}\text{diag}\left(w_+^\top X - w_+^\top X p_l^{(+,t)}\right)X^\top (c_1+c_2) \,\bigg|\, X_{k,+,-}\right]$$

$$=g^{(t)}\Bigg(W_{Q+}^{(t)}v_1 p_{q\leftarrow v_1, k\leftarrow c_1}^{(+,t)}\left(w_+^{(t)\top}c_1 - w_+^{(t)\top}X_{1,+,-}p_{q\leftarrow v_1}^{(+,t)}\right)$$

$$+\,W_{Q+}^{(t)}c_1 p_{q\leftarrow c_1, k\leftarrow c_1}^{(+,t)}\left(w_+^{(t)\top}c_1 - w_+^{(t)\top}X_{1,+,-}p_{q\leftarrow c_1}^{(+,t)}\right)$$

$$+\,W_{Q+}^{(t)}(-v_2) p_{q\leftarrow -v_2, k\leftarrow c_1}^{(+,t)}\left(w_+^{(t)\top}c_1 - w_+^{(t)\top}X_{1,+,-}p_{q\leftarrow -v_2}^{(+,t)}\right)$$

$$+\,W_{Q+}^{(t)}v_2 p_{q\leftarrow v_2, k\leftarrow c_2}^{(+,t)}\left(w_+^{(t)\top}c_2 - w_+^{(t)\top}X_{2,+,-}p_{q\leftarrow v_2}^{(+,t)}\right)$$

$$+\,W_{Q+}^{(t)}c_2 p_{q\leftarrow c_2, k\leftarrow c_2}^{(+,t)}\left(w_+^{(t)\top}c_2 - w_+^{(t)\top}X_{2,+,-}p_{q\leftarrow c_2}^{(+,t)}\right)$$

$$+\,W_{Q+}^{(t)}(-v_1) p_{q\leftarrow -v_1, k\leftarrow c_2}^{(+,t)}\left(w_+^{(t)\top}c_2 - w_+^{(t)\top}X_{2,+,-}p_{q\leftarrow -v_1}^{(+,t)}\right)\Bigg)$$

### E.7 EXPANDING THE UPDATES

To help us analyze the sign of the scores and self-scores in Lemma E.5, we rearrange the gradient flow updates for those variables into forms that are convenient for such analysis.

#### E.7.1 THE KEY SELF-SCORE

$$\frac{dc_1^\top W_{K+}^{(t)\top}W_{K+}^{(t)}v_1}{dt} = \mathbb{E}\left[g^{(t)}y\dot{\sigma}_+^{(t)}\sum_{l=1}^{L}c_1^\top W_{K+}^{(t)\top}q_{+l}^{(t)}p_l^{(+,t)\top}\text{diag}\left(w_+^\top X - w_+^\top X p_l^{(+,t)}\right)X^\top v_1 \,\bigg|\, X_{k,s,-s}\right]$$

$$+\,\mathbb{E}\left[g^{(t)}y\dot{\sigma}_+^{(t)}\sum_{l=1}^{L}v_1^\top W_{K+}^{(t)\top}q_{+l}^{(t)}p_l^{(+,t)\top}\text{diag}\left(w_+^\top X - w_+^\top X p_l^{(+,t)}\right)X^\top c_1 \,\bigg|\, X_{k,s,-s}\right]$$

$$\mathbb{E}\left[g^{(t)}y\dot{\sigma}_+^{(t)}\sum_{l=1}^{L}c_1^\top W_{K+}^{(t)\top}q_{+l}^{(t)}p_l^{(+,t)\top}\text{diag}\left(w_+^\top X - w_+^\top X p_l^{(+,t)}\right)X^\top v_1 \,\bigg|\, X_{k,+,-}\right]$$

$$
= g^{(t)} \Bigg( c_1^\top W_{K+}^{(t)} W_{Q+}^{(t)} v_1 p_{q \leftarrow v_1, k \leftarrow v_1}^{(+,t)} \left( w_+^{(t)\top} v_1 - w_+^{(t)\top} X_{1,+,-} p_{q \leftarrow v_1}^{(+,t)} \right)
$$

$$
+ c_1^\top W_{K+}^{(t)} W_{Q+}^{(t)} c_1 p_{q \leftarrow c_1, k \leftarrow v_1}^{(+,t)} \left( w_+^{(t)\top} v_1 - w_+^{(t)\top} X_{1,+,-} p_{q \leftarrow c_1}^{(+,t)} \right)
$$

$$
+ c_1^\top W_{K+}^{(t)} W_{Q+}^{(t)} (-v_2) p_{q \leftarrow -v_2, k \leftarrow v_1}^{(+,t)} \left( w_+^{(t)\top} v_1 - w_+^{(t)\top} X_{1,+,-} p_{q \leftarrow -v_2}^{(+,t)} \right)
$$

$$
- c_1^\top W_{K+}^{(t)\top} W_{Q+}^{(t)} v_2 p_{q \leftarrow v_2, k \leftarrow -v_1}^{(+,t)} \left( w_+^{(t)\top} (-v_1) - w_+^{(t)\top} X_{2,+,-} p_{q \leftarrow v_2}^{(+,t)} \right)
$$

$$
- c_1^\top W_{K+}^{(t)} W_{Q+}^{(t)} c_2 p_{q \leftarrow c_2, k \leftarrow -v_1}^{(+,t)} \left( w_+^{(t)\top} (-v_1) - w_+^{(t)\top} X_{2,+,-} p_{q \leftarrow c_2}^{(+,t)} \right)
$$

$$
- c_1^\top W_{K+}^{(t)} W_{Q+}^{(t)} (-v_1) p_{q \leftarrow -v_1, k \leftarrow -v_1}^{(+,t)} \left( w_+^{(t)\top} (-v_1) - w_+^{(t)\top} X_{2,+,-} p_{q \leftarrow -v_1}^{(+,t)} \right) \Bigg)
$$

$$
= g^{(t)} \Bigg( c_1^\top W_{K+}^{(t)} W_{Q+}^{(t)} v_1 \left( p_{q \leftarrow v_1, k \leftarrow v_1}^{(+,t)} \left( -w_+^{(t)\top} X_{1,+,-} p_{q \leftarrow v_1}^{(+,t)} \right) + p_{q \leftarrow -v_1, k \leftarrow -v_1}^{(+,t)} \left( -w_+^{(t)\top} X_{2,+,-} p_{q \leftarrow -v_1}^{(+,t)} \right) \right)
$$

$$
+ c_1^\top W_{K+}^{(t)} W_{Q+}^{(t)} v_1 w_+^{(t)\top} v_1 \left( p_{q \leftarrow v_1, k \leftarrow v_1}^{(+,t)} (X_{1,+,-}) - p_{q \leftarrow -v_1, k \leftarrow -v_1}^{(+,t)} (X_{2,+,-}) \right)
$$

$$
+ c_1^\top W_{K+}^{(t)} W_{Q+}^{(t)} (-v_2) \left( p_{q \leftarrow -v_2, k \leftarrow v_1}^{(+,t)} \left( -w_+^{(t)\top} X_{1,+,-} p_{q \leftarrow -v_2}^{(+,t)} \right) + p_{q \leftarrow v_2, k \leftarrow -v_1}^{(+,t)} \left( -w_+^{(t)\top} X_{2,+,-} p_{q \leftarrow v_2}^{(+,t)} \right) \right)
$$

$$
+ c_1^\top W_{K+}^{(t)} W_{Q+}^{(t)} (-v_2) w_+^{(t)\top} v_1 \left( p_{q \leftarrow -v_2, k \leftarrow v_1}^{(+,t)} (X_{1,+,-}) - p_{q \leftarrow v_2, k \leftarrow -v_1}^{(+,t)} (X_{2,+,-}) \right)
$$

$$
+ c_1^\top W_{K+}^{(t)} W_{Q+}^{(t)} c_1 p_{q \leftarrow c_1, k \leftarrow v_1}^{(+,t)} \left( -w_+^{(t)\top} X_{1,+,-} p_{q \leftarrow c_1}^{(+,t)} \right) - c_1^\top W_{K+}^{(t)} W_{Q+}^{(t)} c_2 p_{q \leftarrow c_2, k \leftarrow -v_1}^{(+,t)} \left( -w_+^{(t)\top} X_{2,+,-} p_{q \leftarrow c_2}^{(+,t)} \right)
$$

$$
+ c_1^\top W_{K+}^{(t)} W_{Q+}^{(t)} c_1 p_{q \leftarrow c_1, k \leftarrow v_1}^{(+,t)} (X_{1,+,-}) w_+^{(t)\top} v_1 + c_1^\top W_{K+}^{(t)} W_{Q+}^{(t)} c_2 p_{q \leftarrow c_2, k \leftarrow -v_1}^{(+,t)} (X_{2,+,-}) w_+^{(t)\top} v_1 \Bigg)
$$

$$
\mathbb{E} \left[ g^{(t)} y \dot\sigma_+^{(t)} \sum_{l=1}^{L} c_1^\top W_{K+}^{(t)\top} q_{+l}^{(t)} p_l^{(+,t)\top} \operatorname{diag} \left( w_+^\top X - w_+^\top X p_l^{(+,t)} \right) X^\top v_1 \,\Bigg|\, X_{k,-,+} \right]
$$

$$
= g^{(t)} \Bigg( c_1^\top W_{K+}^{(t)\top} W_{Q+}^{(t)} (-v_1) p_{q \leftarrow -v_1, k \leftarrow -v_1}^{(+,t)} \left( w_+^{(t)\top} (-v_1) - w_+^{(t)\top} X_{1,-,+} p_{q \leftarrow -v_1}^{(+,t)} \right)
$$

$$
+ c_1^\top W_{K+}^{(t)\top} W_{Q+}^{(t)} c_1 p_{q \leftarrow c_1, k \leftarrow -v_1}^{(+,t)} \left( w_+^{(t)\top} (-v_1) - w_+^{(t)\top} X_{1,-,+} p_{q \leftarrow c_1}^{(+,t)} \right)
$$

$$
+ c_1^\top W_{K+}^{(t)\top} W_{Q+}^{(t)} v_2 p_{q \leftarrow v_2, k \leftarrow -v_1}^{(+,t)} \left( w_+^{(t)\top} (-v_1) - w_+^{(t)\top} X_{1,-,+} p_{q \leftarrow v_2}^{(+,t)} \right)
$$

$$
- c_1^\top W_{K+}^{(t)\top} W_{Q+}^{(t)} (-v_2) p_{q \leftarrow -v_2, k \leftarrow v_1}^{(+,t)} \left( w_+^{(t)\top} v_1 - w_+^{(t)\top} X_{2,-,+} p_{q \leftarrow -v_2}^{(+,t)} \right)
$$

$$
- c_1^\top W_{K+}^{(t)\top} W_{Q+}^{(t)} c_2 p_{q \leftarrow c_2, k \leftarrow v_1}^{(+,t)} \left( w_+^{(t)\top} v_1 - w_+^{(t)\top} X_{2,-,+} p_{q \leftarrow c_2}^{(+,t)} \right)
$$

$$
- c_1^\top W_{K+}^{(t)\top} W_{Q+}^{(t)} v_1 p_{q \leftarrow v_1, k \leftarrow v_1}^{(+,t)} \left( w_+^{(t)\top} v_1 - w_+^{(t)\top} X_{2,-,+} p_{q \leftarrow v_1}^{(+,t)} \right) \Bigg)
$$

$$
= g^{(t)} \Bigg( c_1^\top W_{K+}^{(t)\top} W_{Q+}^{(t)} (-v_1) \left( p_{q \leftarrow -v_1, k \leftarrow -v_1}^{(+,t)} \left( -w_+^{(t)\top} X_{1,-,+} p_{q \leftarrow -v_1}^{(+,t)} \right) + p_{q \leftarrow v_1, k \leftarrow v_1}^{(+,t)} \left( -w_+^{(t)\top} X_{2,-,+} p_{q \leftarrow v_1}^{(+,t)} \right) \right)
$$

$$
+ c_1^\top W_{K+}^{(t)\top} W_{Q+}^{(t)} v_1 w_+^{(t)\top} v_1 \left( p_{q \leftarrow -v_1, k \leftarrow -v_1}^{(+,t)} (X_{1,-,+}) - p_{q \leftarrow v_1, k \leftarrow v_1}^{(+,t)} (X_{2,-,+}) \right)
$$

$$
+ c_1^\top W_{K+}^{(t)\top} W_{Q+}^{(t)} v_2 \left( p_{q \leftarrow v_2, k \leftarrow -v_1}^{(+,t)} \left( -w_+^{(t)\top} X_{1,-,+} p_{q \leftarrow v_2}^{(+,t)} \right) + p_{q \leftarrow -v_2, k \leftarrow v_1}^{(+,t)} \left( -w_+^{(t)\top} X_{2,-,+} p_{q \leftarrow -v_2}^{(+,t)} \right) \right)
$$

$$
+ c_1^\top W_{K+}^{(t)\top} W_{Q+}^{(t)} v_2 w_+^{(t)\top} (-v_1) \left( p_{q \leftarrow v_2, k \leftarrow -v_1}^{(+,t)} (X_{1,-,+}) - p_{q \leftarrow -v_2, k \leftarrow v_1}^{(+,t)} (X_{2,-,+}) \right)
$$

$$
+ c_1^\top W_{K+}^{(t)\top} W_{Q+}^{(t)} c_1 p_{q \leftarrow c_1, k \leftarrow -v_1}^{(+,t)} \left( -w_+^{(t)\top} X_{1,-,+} p_{q \leftarrow c_1}^{(+,t)} \right) - c_1^\top W_{K+}^{(t)\top} W_{Q+}^{(t)} c_2 p_{q \leftarrow c_2, k \leftarrow v_1}^{(+,t)} \left( -w_+^{(t)\top} X_{2,-,+} p_{q \leftarrow c_2}^{(+,t)} \right)
$$

$$+ c_1^\top W_{K+}^{(t)\top} W_{Q+}^{(t)} c_1 w_+^{(t)\top} (-v_1) p_{q\leftarrow c_1, k\leftarrow -v_1}^{(+,t)} (X_{1,-,+}) - c_1^\top W_{K+}^{(t)\top} W_{Q+}^{(t)} c_2 w_+^{(t)\top} v_1 p_{q\leftarrow c_2, k\leftarrow v_1}^{(+,t)} (X_{2,-,+}) \bigg)$$

$$\mathbb{E}\left[ g^{(t)} y \dot\sigma_+^{(t)} \sum_{l=1}^{L} v_1^\top W_{K+}^{(t)\top} q_{+l}^{(t)} p_l^{(+,t)\top} \mathrm{diag}\left( w_+^\top X - w_+^\top X p_l^{(+,t)} \right) X^\top c_1 \,\bigg|\, X_{k,s,-s} \right]$$

$$= g^{(t)} \bigg( v_1^\top W_{K+}^{(t)\top} W_{Q+}^{(t)} v_1 p_{q\leftarrow v_1, k\leftarrow c_1}^{(+,t)} \left( w_+^{(t)\top} c_1 - w_+^{(t)\top} X_{1,+,-} p_{q\leftarrow v_1}^{(+,t)} \right)$$

$$+ v_1^\top W_{K+}^{(t)\top} W_{Q+}^{(t)} c_1 p_{q\leftarrow c_1, k\leftarrow c_1}^{(+,t)} \left( w_+^{(t)\top} c_1 - w_+^{(t)\top} X_{1,+,-} p_{q\leftarrow c_1}^{(+,t)} \right)$$

$$+ v_1^\top W_{K+}^{(t)\top} W_{Q+}^{(t)} (-v_2) p_{q\leftarrow -v_2, k\leftarrow c_1}^{(+,t)} \left( w_+^{(t)\top} c_1 - w_+^{(t)\top} X_{1,+,-} p_{q\leftarrow -v_2}^{(+,t)} \right)$$

$$- v_1^\top W_{K+}^{(t)\top} W_{Q+}^{(t)} (-v_1) p_{q\leftarrow -v_1, k\leftarrow c_1}^{(+,t)} \left( w_+^{(t)\top} c_1 - w_+^{(t)\top} X_{1,-,+} p_{q\leftarrow -v_1}^{(+,t)} \right)$$

$$- v_1^\top W_{K+}^{(t)\top} W_{Q+}^{(t)} c_1 p_{q\leftarrow c_1, k\leftarrow c_1}^{(+,t)} \left( w_+^{(t)\top} c_1 - w_+^{(t)\top} X_{1,-,+} p_{q\leftarrow c_1}^{(+,t)} \right)$$

$$- v_1^\top W_{K+}^{(t)\top} W_{Q+}^{(t)} v_2 p_{q\leftarrow v_2, k\leftarrow c_1}^{(+,t)} \left( w_+^{(t)\top} c_1 - w_+^{(t)\top} X_{1,-,+} p_{q\leftarrow v_2}^{(+,t)} \right) \bigg)$$

$$\frac{d v_1^\top W_{K+}^{(t)\top} W_{K+}^{(t)} v_1}{dt} = 2\mathbb{E}\left[ g^{(t)} y \dot\sigma_+^{(t)} \sum_{l=1}^{L} v_1^\top W_{K+}^{(t)\top} q_{+l}^{(t)} p_l^{(+,t)\top} \mathrm{diag}\left( w_+^\top X - w_+^\top X p_l^{(+,t)} \right) X^\top v_1 \,\bigg|\, X_{k,s,-s} \right]$$

$$\mathbb{E}\left[ g^{(t)} y \dot\sigma_+^{(t)} \sum_{l=1}^{L} v_1^\top W_{K+}^{(t)\top} q_{+l}^{(t)} p_l^{(+,t)\top} \mathrm{diag}\left( w_+^\top X - w_+^\top X p_l^{(+,t)} \right) X^\top v_1 \,\bigg|\, X_{k,+,-} \right]$$

$$= g^{(t)} \bigg( v_1^\top W_{K+}^{(t)} W_{Q+}^{(t)} v_1 p_{q\leftarrow v_1, k\leftarrow v_1}^{(+,t)} \left( w_+^{(t)\top} v_1 - w_+^{(t)\top} X_{1,+,-} p_{q\leftarrow v_1}^{(+,t)} \right)$$

$$+ v_1^\top W_{K+}^{(t)} W_{Q+}^{(t)} c_1 p_{q\leftarrow c_1, k\leftarrow v_1}^{(+,t)} \left( w_+^{(t)\top} v_1 - w_+^{(t)\top} X_{1,+,-} p_{q\leftarrow c_1}^{(+,t)} \right)$$

$$+ v_1^\top W_{K+}^{(t)} W_{Q+}^{(t)} (-v_2) p_{q\leftarrow -v_2, k\leftarrow v_1}^{(+,t)} \left( w_+^{(t)\top} v_1 - w_+^{(t)\top} X_{1,+,-} p_{q\leftarrow -v_2}^{(+,t)} \right)$$

$$- v_1^\top W_{K+}^{(t)\top} W_{Q+}^{(t)} v_2 p_{q\leftarrow v_2, k\leftarrow -v_1}^{(+,t)} \left( w_+^{(t)\top} (-v_1) - w_+^{(t)\top} X_{2,+,-} p_{q\leftarrow v_2}^{(+,t)} \right)$$

$$- v_1^\top W_{K+}^{(t)} W_{Q+}^{(t)} c_2 p_{q\leftarrow c_2, k\leftarrow -v_1}^{(+,t)} \left( w_+^{(t)\top} (-v_1) - w_+^{(t)\top} X_{2,+,-} p_{q\leftarrow c_2}^{(+,t)} \right)$$

$$- v_1^\top W_{K+}^{(t)} W_{Q+}^{(t)} (-v_1) p_{q\leftarrow -v_1, k\leftarrow -v_1}^{(+,t)} \left( w_+^{(t)\top} (-v_1) - w_+^{(t)\top} X_{2,+,-} p_{q\leftarrow -v_1}^{(+,t)} \right) \bigg)$$

$$= g^{(t)} \bigg( v_1^\top W_{K+}^{(t)} W_{Q+}^{(t)} v_1 \left( p_{q\leftarrow v_1, k\leftarrow v_1}^{(+,t)} \left( -w_+^{(t)\top} X_{1,+,-} p_{q\leftarrow v_1}^{(+,t)} \right) + p_{q\leftarrow -v_1, k\leftarrow -v_1}^{(+,t)} \left( -w_+^{(t)\top} X_{2,+,-} p_{q\leftarrow -v_1}^{(+,t)} \right) \right)$$

$$+ v_1^\top W_{K+}^{(t)} W_{Q+}^{(t)} v_1 w_+^{(t)\top} v_1 \left( p_{q\leftarrow v_1, k\leftarrow v_1}^{(+,t)} (X_{1,+,-}) - p_{q\leftarrow -v_1, k\leftarrow -v_1}^{(+,t)} (X_{2,+,-}) \right)$$

$$+ v_1^\top W_{K+}^{(t)} W_{Q+}^{(t)} (-v_2) \left( p_{q\leftarrow -v_2, k\leftarrow v_1}^{(+,t)} \left( -w_+^{(t)\top} X_{1,+,-} p_{q\leftarrow -v_2}^{(+,t)} \right) + p_{q\leftarrow v_2, k\leftarrow -v_1}^{(+,t)} \left( -w_+^{(t)\top} X_{2,+,-} p_{q\leftarrow v_2}^{(+,t)} \right) \right)$$

$$+ v_1^\top W_{K+}^{(t)} W_{Q+}^{(t)} (-v_2) w_+^{(t)\top} v_1 \left( p_{q\leftarrow -v_2, k\leftarrow v_1}^{(+,t)} (X_{1,+,-}) - p_{q\leftarrow -v_2, k\leftarrow -v_1}^{(+,t)} (X_{2,+,-}) \right)$$

$$+ v_1^\top W_{K+}^{(t)} W_{Q+}^{(t)} c_1 p_{q\leftarrow c_1, k\leftarrow v_1}^{(+,t)} \left( -w_+^{(t)\top} X_{1,+,-} p_{q\leftarrow c_1}^{(+,t)} \right) - v_1^\top W_{K+}^{(t)} W_{Q+}^{(t)} c_2 p_{q\leftarrow c_2, k\leftarrow -v_1}^{(+,t)} \left( -w_+^{(t)\top} X_{2,+,-} p_{q\leftarrow c_2}^{(+,t)} \right)$$

$$+ v_1^\top W_{K+}^{(t)} W_{Q+}^{(t)} c_1 p_{q\leftarrow c_1, k\leftarrow v_1}^{(+,t)} (X_{1,+,-}) w_+^{(t)\top} v_1 + v_1^\top W_{K+}^{(t)} W_{Q+}^{(t)} c_2 p_{q\leftarrow c_2, k\leftarrow -v_1}^{(+,t)} (X_{2,+,-}) w_+^{(t)\top} v_1 \bigg)$$

$$
\mathbb{E}\left[g^{(t)}y\dot{\sigma}_{+}^{(t)}\sum_{l=1}^{L}c_1^\top W_{K+}^{(t)\top}q_{+l}^{(t)}p_l^{(+,t)\top}\mathrm{diag}\left(w_+^\top X - w_+^\top X p_l^{(+,t)}\right)X^\top v_1 \,\middle|\, X_{k,-,+}\right]
$$

$$
= g^{(t)}\left(v_1^\top W_{K+}^{(t)\top}W_{Q+}^{(t)}(-v_1)p_{q\leftarrow -v_1,k\leftarrow -v_1}^{(+,t)}\left(w_+^{(t)\top}(-v_1) - w_+^{(t)\top}X_{1,-,+}p_{q\leftarrow -v_1}^{(+,t)}\right)\right.
$$

$$
+ v_1^\top W_{K+}^{(t)\top}W_{Q+}^{(t)}c_1 p_{q\leftarrow c_1,k\leftarrow -v_1}^{(+,t)}\left(w_+^{(t)\top}(-v_1) - w_+^{(t)\top}X_{1,-,+}p_{q\leftarrow c_1}^{(+,t)}\right)
$$

$$
+ v_1^\top W_{K+}^{(t)\top}W_{Q+}^{(t)}v_2 p_{q\leftarrow v_2,k\leftarrow -v_1}^{(+,t)}\left(w_+^{(t)\top}(-v_1) - w_+^{(t)\top}X_{1,-,+}p_{q\leftarrow v_2}^{(+,t)}\right)
$$

$$
- v_1^\top W_{K+}^{(t)\top}W_{Q+}^{(t)}(-v_2)p_{q\leftarrow -v_2,k\leftarrow v_1}^{(+,t)}\left(w_+^{(t)\top}v_1 - w_+^{(t)\top}X_{2,-,+}p_{q\leftarrow -v_2}^{(+,t)}\right)
$$

$$
- v_1^\top W_{K+}^{(t)\top}W_{Q+}^{(t)}c_2 p_{q\leftarrow c_2,k\leftarrow v_1}^{(+,t)}\left(w_+^{(t)\top}v_1 - w_+^{(t)\top}X_{2,-,+}p_{q\leftarrow c_2}^{(+,t)}\right)
$$

$$
\left.- v_1^\top W_{K+}^{(t)\top}W_{Q+}^{(t)}v_1 p_{q\leftarrow v_1,k\leftarrow v_1}^{(+,t)}\left(w_+^{(t)\top}v_1 - w_+^{(t)\top}X_{2,-,+}p_{q\leftarrow v_1}^{(+,t)}\right)\right)
$$

$$
= g^{(t)}\left(v_1^\top W_{K+}^{(t)\top}W_{Q+}^{(t)}(-v_1)\left(p_{q\leftarrow -v_1,k\leftarrow -v_1}^{(+,t)}\left(-w_+^{(t)\top}X_{1,-,+}p_{q\leftarrow -v_1}^{(+,t)}\right) + p_{q\leftarrow v_1,k\leftarrow v_1}^{(+,t)}\left(-w_+^{(t)\top}X_{2,-,+}p_{q\leftarrow v_1}^{(+,t)}\right)\right)\right.
$$

$$
+ v_1^\top W_{K+}^{(t)\top}W_{Q+}^{(t)}v_1 w_+^{(t)\top}v_1\left(p_{q\leftarrow -v_1,k\leftarrow -v_1}^{(+,t)}(X_{1,-,+}) - p_{q\leftarrow v_1,k\leftarrow v_1}^{(+,t)}(X_{2,-,+})\right)
$$

$$
+ v_1^\top W_{K+}^{(t)\top}W_{Q+}^{(t)}v_2\left(p_{q\leftarrow v_2,k\leftarrow -v_1}^{(+,t)}\left(-w_+^{(t)\top}X_{1,-,+}p_{q\leftarrow v_2}^{(+,t)}\right) + p_{q\leftarrow -v_2,k\leftarrow v_1}^{(+,t)}\left(-w_+^{(t)\top}X_{2,-,+}p_{q\leftarrow -v_2}^{(+,t)}\right)\right)
$$

$$
+ v_1^\top W_{K+}^{(t)\top}W_{Q+}^{(t)}v_2 w_+^{(t)\top}(-v_1)\left(p_{q\leftarrow v_2,k\leftarrow -v_1}^{(+,t)}(X_{1,-,+}) - p_{q\leftarrow -v_2,k\leftarrow v_1}^{(+,t)}(X_{2,-,+})\right)
$$

$$
+ v_1^\top W_{K+}^{(t)\top}W_{Q+}^{(t)}c_1 p_{q\leftarrow c_1,k\leftarrow -v_1}^{(+,t)}\left(-w_+^{(t)\top}X_{1,-,+}p_{q\leftarrow c_1}^{(+,t)}\right) - v_1^\top W_{K+}^{(t)\top}W_{Q+}^{(t)}c_2 p_{q\leftarrow c_2,k\leftarrow v_1}^{(+,t)}\left(-w_+^{(t)\top}X_{2,-,+}p_{q\leftarrow c_2}^{(+,t)}\right)
$$

$$
\left.+ v_1^\top W_{K+}^{(t)\top}W_{Q+}^{(t)}c_1 w_+^{(t)\top}(-v_1)p_{q\leftarrow c_1,k\leftarrow -v_1}^{(+,t)}(X_{1,-,+}) - v_1^\top W_{K+}^{(t)\top}W_{Q+}^{(t)}c_2 w_+^{(t)\top}v_1 p_{q\leftarrow c_2,k\leftarrow v_1}^{(+,t)}(X_{2,-,+})\right)
$$

### E.7.2 THE QUERY SELF-SCORE

$$
\frac{\mathrm{d}c_1^\top W_{Q+}^{(t)\top}W_{Q+}^{(t)}v_1}{\mathrm{d}t}
$$

$$
= \mathbb{E}\left[g^{(t)}y\dot{\sigma}_{+}^{(t)}\sum_{l=1}^{L}c_1^\top W_{Q+}^{(t)\top}K_+^{(t)}\mathrm{diag}\left(X^\top w_+ - w_+^\top X p_l^{(+,t)}\right)p_l^{(+,t)}x_l^\top v_1 \,\middle|\, X_{k,s,-s}\right]
$$

$$
+ \mathbb{E}\left[g^{(t)}y\dot{\sigma}_{+}^{(t)}\sum_{l=1}^{L}v_1^\top W_{Q+}^{(t)\top}K_+^{(t)}\mathrm{diag}\left(X^\top w_+ - w_+^\top X p_l^{(+,t)}\right)p_l^{(+,t)}x_l^\top c_1 \,\middle|\, X_{k,s,-s}\right]
$$

$$
\mathbb{E}\left[g^{(t)}y\dot{\sigma}_{+}^{(t)}\sum_{l=1}^{L}c_1^\top W_{Q+}^{(t)\top}K_+^{(t)}\mathrm{diag}\left(X^\top w_+ - w_+^\top X p_l^{(+,t)}\right)p_l^{(+,t)}x_l^\top v_1 \,\middle|\, X_{k,+,-}\right]
$$

$$
= g^{(t)}\left(c_1^\top W_{Q+}^{(t)\top}W_{K+}^{(t)}v_1\left(w_+^{(t)\top}v_1 - w_+^{(t)\top}X_{1,+,-}p_{q\leftarrow v_1}^{(+,t)}\right)p_{q\leftarrow v_1,k\leftarrow v_1}^{(+,t)}\right.
$$

$$
+ c_1^\top W_{Q+}^{(t)\top}W_{K+}^{(t)}c_1\left(w_+^{(t)\top}c_1 - w_+^{(t)\top}X_{1,+,-}p_{q\leftarrow v_1}^{(+,t)}\right)p_{q\leftarrow v_1,k\leftarrow c_1}^{(+,t)}
$$

$$
+ c_1^\top W_{Q+}^{(t)\top}W_{K+}^{(t)}(-v_2)\left(w_+^{(t)\top}(-v_2) - w_+^{(t)\top}X_{1,+,-}p_{q\leftarrow v_1}^{(+,t)}\right)p_{q\leftarrow v_1,k\leftarrow -v_2}^{(+,t)}
$$

$$
- c_1^\top W_{Q+}^{(t)\top}W_{K+}^{(t)}v_2\left(w_+^{(t)\top}v_2 - w_+^{(t)\top}X_{2,+,-}p_{q\leftarrow -v_1}^{(+,t)}\right)p_{q\leftarrow -v_1,k\leftarrow v_2}^{(+,t)}
$$

$$
- c_1^\top W_{Q+}^{(t)\top}W_{K+}^{(t)}c_2\left(w_+^{(t)\top}c_2 - w_+^{(t)\top}X_{2,+,-}p_{q\leftarrow -v_1}^{(+,t)}\right)p_{q\leftarrow -v_1,k\leftarrow c_2}^{(+,t)}
$$

$$- c_1^\top W_{Q+}^{(t)\top} W_{K+}^{(t)}(-v_1)\left(w_+^{(t)\top}(-v_1) - w_+^{(t)\top}X_{2,+,-}p_{q\leftarrow -v_1}^{(+,t)}\right)p_{q\leftarrow -v_1, k\leftarrow -v_1}^{(+,t)}\Bigg)$$

$$= g^{(t)}\Bigg(c_1^\top W_{Q+}^{(t)\top} W_{K+}^{(t)}v_1\left(\left(-w_+^{(t)\top}X_{1,+,-}p_{q\leftarrow v_1}^{(+,t)}\right)p_{q\leftarrow v_1, k\leftarrow v_1}^{(+,t)} + \left(-w_+^{(t)\top}X_{2,+,-}p_{q\leftarrow -v_1}^{(+,t)}\right)p_{q\leftarrow -v_1, k\leftarrow -v_1}^{(+,t)}\right)$$

$$+ c_1^\top W_{Q+}^{(t)\top} W_{K+}^{(t)}v_1 w_+^{(t)\top}v_1\left(p_{q\leftarrow v_1, k\leftarrow v_1}^{(+,t)}(X_{1,+,-}) - p_{q\leftarrow -v_1, k\leftarrow -v_1}^{(+,t)}(X_{2,+,-})\right)$$

$$+ c_1^\top W_{Q+}^{(t)\top} W_{K+}^{(t)}(-v_2)\left(\left(-w_+^{(t)\top}X_{1,+,-}p_{q\leftarrow v_1}^{(+,t)}\right)p_{q\leftarrow v_1, k\leftarrow -v_2}^{(+,t)} + \left(-w_+^{(t)\top}X_{2,+,-}p_{q\leftarrow -v_1}^{(+,t)}\right)p_{q\leftarrow -v_1, k\leftarrow -v_2}^{(+,t)}\right)$$

$$+ c_1^\top W_{Q+}^{(t)\top} W_{K+}^{(t)}v_2 w_+^{(t)\top}v_2\left(p_{q\leftarrow v_1, k\leftarrow -v_2}^{(+,t)}(X_{1,+,-}) - p_{q\leftarrow -v_1, k\leftarrow -v_2}^{(+,t)}(X_{2,+,-})\right)$$

$$+ c_1^\top W_{Q+}^{(t)\top} W_{K+}^{(t)}c_1\left(w_+^{(t)\top}c_1 - w_+^{(t)\top}X_{1,+,-}p_{q\leftarrow v_1}^{(+,t)}\right)p_{q\leftarrow v_1, k\leftarrow c_1}^{(+,t)}$$

$$- c_1^\top W_{Q+}^{(t)\top} W_{K+}^{(t)}c_2\left(w_+^{(t)\top}c_2 - w_+^{(t)\top}X_{2,+,-}p_{q\leftarrow -v_1}^{(+,t)}\right)p_{q\leftarrow -v_1, k\leftarrow c_2}^{(+,t)}\Bigg)$$

$$\mathbb{E}\left[g^{(t)}y\dot\sigma_+^{(t)}\sum_{l=1}^L c_1^\top W_{Q+}^{(t)\top} K_+^{(t)}\mathrm{diag}\left(X^\top w_+ - w_+^\top X p_l^{(+,t)}\right)p_l^{(+,t)}x_l^\top v_1 \bigg| X_{k,-,+}\right]$$

$$= g^{(t)}\Bigg(c_1^\top W_{Q+}^{(t)\top} W_{K+}^{(t)}(-v_1)\left(w_+^{(t)\top}(-v_1) - w_+^{(t)\top}X_{1,-,+}p_{q\leftarrow -v_1}^{(+,t)}\right)p_{q\leftarrow -v_1, k\leftarrow -v_1}^{(+,t)}$$

$$+ c_1^\top W_{Q+}^{(t)\top} W_{K+}^{(t)}c_1\left(w_+^{(t)\top}c_1 - w_+^{(t)\top}X_{1,-,+}p_{q\leftarrow -v_1}^{(+,t)}\right)p_{q\leftarrow -v_1, k\leftarrow c_1}^{(+,t)}$$

$$+ c_1^\top W_{Q+}^{(t)\top} W_{K+}^{(t)}v_2\left(w_+^{(t)\top}v_2 - w_+^{(t)\top}X_{1,-,+}p_{q\leftarrow -v_1}^{(+,t)}\right)p_{q\leftarrow -v_1, k\leftarrow v_2}^{(+,t)}$$

$$- c_1^\top W_{Q+}^{(t)\top} W_{K+}^{(t)}(-v_2)\left(w_+^{(t)\top}(-v_2) - w_+^{(t)\top}X_{2,-,+}p_{q\leftarrow v_1}^{(+,t)}\right)p_{q\leftarrow v_1, k\leftarrow -v_2}^{(+,t)}$$

$$- c_1^\top W_{Q+}^{(t)\top} W_{K+}^{(t)}c_2\left(w_+^{(t)\top}c_2 - w_+^{(t)\top}X_{2,-,+}p_{q\leftarrow v_1}^{(+,t)}\right)p_{q\leftarrow v_1, k\leftarrow c_2}^{(+,t)}$$

$$- c_1^\top W_{Q+}^{(t)\top} W_{K+}^{(t)}v_1\left(w_+^{(t)\top}v_1 - w_+^{(t)\top}X_{2,-,+}p_{q\leftarrow v_1}^{(+,t)}\right)p_{q\leftarrow v_1, k\leftarrow v_1}^{(+,t)}\Bigg)$$

$$= g^{(t)}\Bigg(c_1^\top W_{Q+}^{(t)\top} W_{K+}^{(t)}(-v_1)\left(\left(-w_+^{(t)\top}X_{1,-,+}p_{q\leftarrow -v_1}^{(+,t)}\right)p_{q\leftarrow -v_1, k\leftarrow -v_1}^{(+,t)} + \left(-w_+^{(t)\top}X_{2,-,+}p_{q\leftarrow v_1}^{(+,t)}\right)p_{q\leftarrow v_1, k\leftarrow v_1}^{(+,t)}\right)$$

$$+ c_1^\top W_{Q+}^{(t)\top} W_{K+}^{(t)}v_1 w_+^{(t)\top}v_1\left(p_{q\leftarrow -v_1, k\leftarrow -v_1}^{(+,t)}(X_{1,-,+}) - p_{q\leftarrow v_1, k\leftarrow v_1}^{(+,t)}(X_{2,-,+})\right)$$

$$+ c_1^\top W_{Q+}^{(t)\top} W_{K+}^{(t)}v_2\left(\left(-w_+^{(t)\top}X_{1,-,+}p_{q\leftarrow -v_1}^{(+,t)}\right)p_{q\leftarrow -v_1, k\leftarrow v_2}^{(+,t)} + \left(-w_+^{(t)\top}X_{2,-,+}p_{q\leftarrow v_1}^{(+,t)}\right)p_{q\leftarrow v_1, k\leftarrow -v_2}^{(+,t)}\right)$$

$$+ c_1^\top W_{Q+}^{(t)\top} W_{K+}^{(t)}v_2 w_+^{(t)\top}v_2\left(p_{q\leftarrow -v_1, k\leftarrow v_2}^{(+,t)}(X_{1,-,+}) - p_{q\leftarrow v_1, k\leftarrow -v_2}^{(+,t)}(X_{2,-,+})\right)$$

$$+ c_1^\top W_{Q+}^{(t)\top} W_{K+}^{(t)}c_1\left(w_+^{(t)\top}c_1 - w_+^{(t)\top}X_{1,-,+}p_{q\leftarrow -v_1}^{(+,t)}\right)p_{q\leftarrow -v_1, k\leftarrow c_1}^{(+,t)}$$

$$- c_1^\top W_{Q+}^{(t)\top} W_{K+}^{(t)}c_2\left(w_+^{(t)\top}c_2 - w_+^{(t)\top}X_{2,-,+}p_{q\leftarrow v_1}^{(+,t)}\right)p_{q\leftarrow v_1, k\leftarrow c_2}^{(+,t)}\Bigg)$$

$$\mathbb{E}\left[g^{(t)}y\dot\sigma_+^{(t)}\sum_{l=1}^L v_1^\top W_{Q+}^{(t)\top} K_+^{(t)}\mathrm{diag}\left(X^\top w_+ - w_+^\top X p_l^{(+,t)}\right)p_l^{(+,t)}x_l^\top c_1 \bigg| X_{k,s,-s}\right]$$

$$= g^{(t)}\Bigg(v_1^\top W_{Q+}^{(t)\top} W_{K+}^{(t)}v_1\left(w_+^{(t)\top}v_1 - w_+^{(t)\top}X_{1,+,-}p_{q\leftarrow c_1}^{(+,t)}\right)p_{q\leftarrow c_1, k\leftarrow v_1}^{(+,t)}$$

$$+ v_1^\top W_{Q+}^{(t)\top} W_{K+}^{(t)}c_1\left(w_+^{(t)\top}c_1 - w_+^{(t)\top}X_{1,+,-}p_{q\leftarrow c_1}^{(+,t)}\right)p_{q\leftarrow c_1, k\leftarrow c_1}^{(+,t)}$$

$$+ v_1^\top W_{Q+}^{(t)\top} W_{K+}^{(t)}(-v_2)\left(w_+^{(t)\top}(-v_2) - w_+^{(t)\top} X_{1,+,-} p_{q\leftarrow c_1}^{(+,t)}\right) p_{q\leftarrow c_1, k\leftarrow -v_2}^{(+,t)}$$

$$- v_1^\top W_{Q+}^{(t)\top} W_{K+}^{(t)}(-v_1)\left(w_+^{(t)\top}(-v_1) - w_+^{(t)\top} X_{1,-,+} p_{q\leftarrow c_1}^{(+,t)}\right) p_{q\leftarrow c_1, k\leftarrow -v_1}^{(+,t)}$$

$$- v_1^\top W_{Q+}^{(t)\top} W_{K+}^{(t)} c_1\left(w_+^{(t)\top} c_1 - w_+^{(t)\top} X_{1,-,+} p_{q\leftarrow c_1}^{(+,t)}\right) p_{q\leftarrow c_1, k\leftarrow c_1}^{(+,t)}$$

$$- v_1^\top W_{Q+}^{(t)\top} W_{K+}^{(t)} v_2\left(w_+^{(t)\top} v_2 - w_+^{(t)\top} X_{1,-,+} p_{q\leftarrow c_1}^{(+,t)}\right) p_{q\leftarrow c_1, k\leftarrow v_2}^{(+,t)}$$

$$= g^{(t)}\left( v_1^\top W_{Q+}^{(t)\top} W_{K+}^{(t)} v_1\left(\left(-w_+^{(t)\top} X_{1,+,-} p_{q\leftarrow c_1}^{(+,t)}\right) p_{q\leftarrow c_1, k\leftarrow v_1}^{(+,t)} + \left(-w_+^{(t)\top} X_{1,-,+} p_{q\leftarrow c_1}^{(+,t)}\right) p_{q\leftarrow c_1, k\leftarrow -v_1}^{(+,t)}\right)\right.$$

$$+ v_1^\top W_{Q+}^{(t)\top} W_{K+}^{(t)} v_1 w_+^{(t)\top} v_1\left(p_{q\leftarrow c_1, k\leftarrow v_1}^{(+,t)}(X_{1,+,-}) - p_{q\leftarrow c_1, k\leftarrow -v_1}^{(+,t)}(X_{1,-,+})\right)$$

$$+ v_1^\top W_{Q+}^{(t)\top} W_{K+}^{(t)}(-v_2)\left(\left(-w_+^{(t)\top} X_{1,+,-} p_{q\leftarrow c_1}^{(+,t)}\right) p_{q\leftarrow c_1, k\leftarrow -v_2}^{(+,t)} + \left(-w_+^{(t)\top} X_{1,-,+} p_{q\leftarrow c_1}^{(+,t)}\right) p_{q\leftarrow c_1, k\leftarrow v_2}^{(+,t)}\right)$$

$$+ v_1^\top W_{Q+}^{(t)\top} W_{K+}^{(t)} v_2 w_+^{(t)\top} v_2\left(p_{q\leftarrow c_1, k\leftarrow -v_2}^{(+,t)}(X_{1,+,-}) - p_{q\leftarrow c_1, k\leftarrow v_2}^{(+,t)}(X_{1,-,+})\right)$$

$$+ v_1^\top W_{Q+}^{(t)\top} W_{K+}^{(t)} c_1\left(w_+^{(t)\top} c_1 - w_+^{(t)\top} X_{1,+,-} p_{q\leftarrow c_1}^{(+,t)}\right) p_{q\leftarrow c_1, k\leftarrow c_1}^{(+,t)}$$

$$\left. - v_1^\top W_{Q+}^{(t)\top} W_{K+}^{(t)} c_1\left(w_+^{(t)\top} c_1 - w_+^{(t)\top} X_{1,-,+} p_{q\leftarrow c_1}^{(+,t)}\right) p_{q\leftarrow c_1, k\leftarrow c_1}^{(+,t)}\right)$$

$$\frac{d c_1^\top W_{Q+}^{(t)\top} W_{Q+}^{(t)} c_1}{dt} = 2\mathbb{E}\left[g^{(t)} y \dot\sigma_+^{(t)} \sum_{l=1}^{L} c_1^\top W_{Q+}^{(t)\top} K_+^{(t)} \mathrm{diag}\left(X^\top w_+ - w_+^\top X p_l^{(+,t)}\right) p_l^{(+,t)} x_l^\top c_1 \,\middle|\, X_{k,s,-s}\right]$$

$$\mathbb{E}\left[g^{(t)} y \dot\sigma_+^{(t)} \sum_{l=1}^{L} c_1^\top W_{Q+}^{(t)\top} K_+^{(t)} \mathrm{diag}\left(X^\top w_+ - w_+^\top X p_l^{(+,t)}\right) p_l^{(+,t)} x_l^\top c_1 \,\middle|\, X_{k,s,-s}\right]$$

$$= g^{(t)}\left( c_1^\top W_{Q+}^{(t)\top} W_{K+}^{(t)} v_1\left(w_+^{(t)\top} v_1 - w_+^{(t)\top} X_{1,+,-} p_{q\leftarrow c_1}^{(+,t)}\right) p_{q\leftarrow c_1, k\leftarrow v_1}^{(+,t)}\right.$$

$$+ c_1^\top W_{Q+}^{(t)\top} W_{K+}^{(t)} c_1\left(w_+^{(t)\top} c_1 - w_+^{(t)\top} X_{1,+,-} p_{q\leftarrow c_1}^{(+,t)}\right) p_{q\leftarrow c_1, k\leftarrow c_1}^{(+,t)}$$

$$+ c_1^\top W_{Q+}^{(t)\top} W_{K+}^{(t)}(-v_2)\left(w_+^{(t)\top}(-v_2) - w_+^{(t)\top} X_{1,+,-} p_{q\leftarrow c_1}^{(+,t)}\right) p_{q\leftarrow c_1, k\leftarrow -v_2}^{(+,t)}$$

$$- c_1^\top W_{Q+}^{(t)\top} W_{K+}^{(t)}(-v_1)\left(w_+^{(t)\top}(-v_1) - w_+^{(t)\top} X_{1,-,+} p_{q\leftarrow c_1}^{(+,t)}\right) p_{q\leftarrow c_1, k\leftarrow -v_1}^{(+,t)}$$

$$- c_1^\top W_{Q+}^{(t)\top} W_{K+}^{(t)} c_1\left(w_+^{(t)\top} c_1 - w_+^{(t)\top} X_{1,-,+} p_{q\leftarrow c_1}^{(+,t)}\right) p_{q\leftarrow c_1, k\leftarrow c_1}^{(+,t)}$$

$$\left. - c_1^\top W_{Q+}^{(t)\top} W_{K+}^{(t)} v_2\left(w_+^{(t)\top} v_2 - w_+^{(t)\top} X_{1,-,+} p_{q\leftarrow c_1}^{(+,t)}\right) p_{q\leftarrow c_1, k\leftarrow v_2}^{(+,t)}\right)$$

$$= g^{(t)}\left( c_1^\top W_{Q+}^{(t)\top} W_{K+}^{(t)} v_1\left(\left(-w_+^{(t)\top} X_{1,+,-} p_{q\leftarrow c_1}^{(+,t)}\right) p_{q\leftarrow c_1, k\leftarrow v_1}^{(+,t)} + \left(-w_+^{(t)\top} X_{1,-,+} p_{q\leftarrow c_1}^{(+,t)}\right) p_{q\leftarrow c_1, k\leftarrow -v_1}^{(+,t)}\right)\right.$$

$$+ c_1^\top W_{Q+}^{(t)\top} W_{K+}^{(t)} v_1 w_+^{(t)\top} v_1\left(p_{q\leftarrow c_1, k\leftarrow v_1}^{(+,t)}(X_{1,+,-}) - p_{q\leftarrow c_1, k\leftarrow -v_1}^{(+,t)}(X_{1,-,+})\right)$$

$$+ c_1^\top W_{Q+}^{(t)\top} W_{K+}^{(t)}(-v_2)\left(\left(-w_+^{(t)\top} X_{1,+,-} p_{q\leftarrow c_1}^{(+,t)}\right) p_{q\leftarrow c_1, k\leftarrow -v_2}^{(+,t)} + \left(-w_+^{(t)\top} X_{1,-,+} p_{q\leftarrow c_1}^{(+,t)}\right) p_{q\leftarrow c_1, k\leftarrow v_2}^{(+,t)}\right)$$

$$+ c_1^\top W_{Q+}^{(t)\top} W_{K+}^{(t)} v_2 w_+^{(t)\top} v_2\left(p_{q\leftarrow c_1, k\leftarrow -v_2}^{(+,t)}(X_{1,+,-}) - p_{q\leftarrow c_1, k\leftarrow v_2}^{(+,t)}(X_{1,-,+})\right)$$

$$+ c_1^\top W_{Q+}^{(t)\top} W_{K+}^{(t)} c_1\left(w_+^{(t)\top} c_1 - w_+^{(t)\top} X_{1,+,-} p_{q\leftarrow c_1}^{(+,t)}\right) p_{q\leftarrow c_1, k\leftarrow c_1}^{(+,t)}$$

$$\left. - c_1^\top W_{Q+}^{(t)\top} W_{K+}^{(t)} c_1\left(w_+^{(t)\top} c_1 - w_+^{(t)\top} X_{1,-,+} p_{q\leftarrow c_1}^{(+,t)}\right) p_{q\leftarrow c_1, k\leftarrow c_1}^{(+,t)}\right)$$

### E.7.3 The Scores

$$\frac{dc_1^\top W_{K+}^{(t)\top} W_{Q+}^{(t)} c_1}{dt} = \mathbb{E}\left[ g^{(t)} y \dot\sigma_+^{(t)} \sum_{l=1}^L c_1^\top W_{Q+}^{(t)\top} q_{+l}^{(t)} p_l^{(+,t)\top} \mathrm{diag}\left( w_+^{(t)\top} X - w_+^{(t)\top} X p_l^{(+,t)} \right) X^\top c_1 \,\middle|\, X_{k,s,-s} \right]$$

$$+ \mathbb{E}\left[ g^{(t)} y \dot\sigma_+^{(t)} \sum_{l=1}^L c_1^\top W_{K+}^{(t)\top} K_+^{(t)} \mathrm{diag}\left( X^\top w_+ - w_+^\top X p_l^{(+,t)} \right) p_l^{(+,t)} x_l^\top c_1 \,\middle|\, X_{k,s,-s} \right]$$

$$\mathbb{E}\left[ g^{(t)} y \dot\sigma_+^{(t)} \sum_{l=1}^L c_1^\top W_{Q+}^{(t)\top} q_{+l}^{(t)} p_l^{(+,t)\top} \mathrm{diag}\left( w_+^{(t)\top} X - w_+^{(t)\top} X p_l^{(+,t)} \right) X^\top c_1 \,\middle|\, X_{k,s,-s} \right]$$

$$= g^{(t)} \Bigg( c_1^\top W_{Q+}^{(t)\top} W_{Q+}^{(t)} v_1 p_{q\leftarrow v_1, k\leftarrow c_1}^{(+,t)} \left( w_+^{(t)\top} c_1 - w_+^{(t)\top} X_{1,+,-} p_{q\leftarrow v_1}^{(+,t)} \right)$$

$$+ c_1^\top W_{Q+}^{(t)\top} W_{Q+}^{(t)} c_1 p_{q\leftarrow c_1, k\leftarrow c_1}^{(+,t)} \left( w_+^{(t)\top} c_1 - w_+^{(t)\top} X_{1,+,-} p_{q\leftarrow c_1}^{(+,t)} \right)$$

$$+ c_1^\top W_{Q+}^{(t)\top} W_{Q+}^{(t)} (-v_2) p_{q\leftarrow -v_2, k\leftarrow c_1}^{(+,t)} \left( w_+^{(t)\top} c_1 - w_+^{(t)\top} X_{1,+,-} p_{q\leftarrow -v_2}^{(+,t)} \right)$$

$$- c_1^\top W_{Q+}^{(t)\top} W_{Q+}^{(t)} (-v_1) p_{q\leftarrow (-v_1), k\leftarrow c_1}^{(+,t)} \left( w_+^{(t)\top} c_1 - w_+^{(t)\top} X_{1,-,+} p_{q\leftarrow -v_1}^{(+,t)} \right)$$

$$- c_1^\top W_{Q+}^{(t)\top} W_{Q+}^{(t)} c_1 p_{q\leftarrow c_1, k\leftarrow c_1}^{(+,t)} \left( w_+^{(t)\top} c_1 - w_+^{(t)\top} X_{1,-,+} p_{q\leftarrow c_1}^{(+,t)} \right)$$

$$- c_1^\top W_{Q+}^{(t)\top} W_{Q+}^{(t)} v_2 p_{q\leftarrow v_2, k\leftarrow c_1}^{(+,t)} \left( w_+^{(t)\top} c_1 - w_+^{(t)\top} X_{1,-,+} p_{q\leftarrow v_2}^{(+,t)} \right) \Bigg)$$

$$\mathbb{E}\left[ g^{(t)} y \dot\sigma_+^{(t)} \sum_{l=1}^L c_1^\top W_{K+}^{(t)\top} K_+^{(t)} \mathrm{diag}\left( X^\top w_+ - w_+^\top X p_l^{(+,t)} \right) p_l^{(+,t)} x_l^\top c_1 \,\middle|\, X_{k,s,-s} \right]$$

$$= g^{(t)} \Bigg( c_1^\top W_{K+}^{(t)\top} W_{K+}^{(t)} v_1 \left( w_+^{(t)\top} v_1 - w_+^{(t)\top} X_{1,+,-} p_{q\leftarrow c_1}^{(+,t)} \right) p_{q\leftarrow c_1, k\leftarrow v_1}^{(+,t)}$$

$$+ c_1^\top W_{K+}^{(t)\top} W_{K+}^{(t)} c_1 \left( w_+^{(t)\top} c_1 - w_+^{(t)\top} X_{1,+,-} p_{q\leftarrow c_1}^{(+,t)} \right) p_{q\leftarrow c_1, k\leftarrow c_1}^{(+,t)}$$

$$+ c_1^\top W_{K+}^{(t)\top} W_{K+}^{(t)} (-v_2) \left( w_+^{(t)\top} (-v_2) - w_+^{(t)\top} X_{1,+,-} p_{q\leftarrow c_1}^{(+,t)} \right) p_{q\leftarrow c_1, k\leftarrow -v_2}^{(+,t)}$$

$$- c_1^\top W_{K+}^{(t)\top} W_{K+}^{(t)} (-v_1) \left( w_+^{(t)\top} (-v_1) - w_+^{(t)\top} X_{1,-,+} p_{q\leftarrow c_1}^{(+,t)} \right) p_{q\leftarrow c_1, k\leftarrow -v_1}^{(+,t)}$$

$$- c_1^\top W_{K+}^{(t)\top} W_{K+}^{(t)} c_1 \left( w_+^{(t)\top} c_1 - w_+^{(t)\top} X_{1,-,+} p_{q\leftarrow c_1}^{(+,t)} \right) p_{q\leftarrow c_1, k\leftarrow c_1}^{(+,t)}$$

$$- c_1^\top W_{K+}^{(t)\top} W_{K+}^{(t)} v_2 \left( w_+^{(t)\top} v_2 - w_+^{(t)\top} X_{1,-,+} p_{q\leftarrow c_1}^{(+,t)} \right) p_{q\leftarrow c_1, k\leftarrow v_2}^{(+,t)} \Bigg)$$

$$= g^{(t)} \Bigg( c_1^\top W_{K+}^{(t)\top} W_{K+}^{(t)} v_1 \left( \left( -w_+^{(t)\top} X_{1,+,-} p_{q\leftarrow c_1}^{(+,t)} \right) p_{q\leftarrow c_1, k\leftarrow v_1}^{(+,t)} + \left( -w_+^{(t)\top} X_{1,-,+} p_{q\leftarrow c_1}^{(+,t)} \right) p_{q\leftarrow c_1, k\leftarrow -v_1}^{(+,t)} \right)$$

$$+ c_1^\top W_{K+}^{(t)\top} W_{K+}^{(t)} v_1 w_+^{(t)\top} v_1 \left( p_{q\leftarrow c_1, k\leftarrow v_1}^{(+,t)} (X_{1,+,-}) - p_{q\leftarrow c_1, k\leftarrow -v_1}^{(+,t)} (X_{1,-,+}) \right)$$

$$+ c_1^\top W_{K+}^{(t)\top} W_{K+}^{(t)} (-v_2) \left( \left( -w_+^{(t)\top} X_{1,+,-} p_{q\leftarrow c_1}^{(+,t)} \right) p_{q\leftarrow c_1, k\leftarrow -v_2}^{(+,t)} + \left( -w_+^{(t)\top} X_{1,-,+} p_{q\leftarrow c_1}^{(+,t)} \right) p_{q\leftarrow c_1, k\leftarrow v_2}^{(+,t)} \right)$$

$$+ c_1^\top W_{K+}^{(t)\top} W_{K+}^{(t)} v_2 w_+^{(t)\top} v_2 \left( p_{q\leftarrow c_1, k\leftarrow -v_2}^{(+,t)} (X_{1,+,-}) - p_{q\leftarrow c_1, k\leftarrow v_2}^{(+,t)} (X_{1,-,+}) \right)$$

$$+ c_1^\top W_{K+}^{(t)\top} W_{K+}^{(t)} c_1 \left( w_+^{(t)\top} c_1 - w_+^{(t)\top} X_{1,+,-} p_{q\leftarrow c_1}^{(+,t)} \right) p_{q\leftarrow c_1, k\leftarrow c_1}^{(+,t)}$$

$$- c_1^\top W_{K+}^{(t)\top} W_{K+}^{(t)} c_1 \left( w_+^{(t)\top} c_1 - w_+^{(t)\top} X_{1,-,+} p_{q\leftarrow c_1}^{(+,t)} \right) p_{q\leftarrow c_1, k\leftarrow c_1}^{(+,t)} \Bigg)$$

$$\frac{dv_1^\top W_{K+}^{(t)\top} W_{Q+}^{(t)} v_1}{dt} = \mathbb{E}\left[g^{(t)} y \dot{\sigma}_+^{(t)} \sum_{l=1}^{L} v_1^\top W_{Q+}^{(t)\top} q_{+l}^{(t)} p_l^{(+,t)\top} \mathrm{diag}\left(w_+^{(t)\top} X - w_+^{(t)\top} X p_l^{(+,t)}\right) X^\top v_1 \middle| X_{k,s,-s}\right]$$

$$+ \mathbb{E}\left[g^{(t)} y \dot{\sigma}_+^{(t)} \sum_{l=1}^{L} v_1^\top W_{K+}^{(t)\top} K_+^{(t)} \mathrm{diag}\left(X^\top w_+ - w_+^\top X p_l^{(+,t)}\right) p_l^{(+,t)} x_l^\top v_1 \middle| X_{k,s,-s}\right]$$

$$\mathbb{E}\left[g^{(t)} y \dot{\sigma}_+^{(t)} \sum_{l=1}^{L} v_1^\top W_{Q+}^{(t)\top} q_{+l}^{(t)} p_l^{(+,t)\top} \mathrm{diag}\left(w_+^{(t)\top} X - w_+^{(t)\top} X p_l^{(+,t)}\right) X^\top v_1 \middle| X_{k,+,-}\right]$$

$$= g^{(t)}\left( v_1^\top W_{Q+}^{(t)\top} W_{Q+}^{(t)} v_1 p_{q\leftarrow v_1, k\leftarrow v_1}^{(+,t)} \left(w_+^{(t)\top} v_1 - w_+^{(t)\top} X_{1,+,-} p_{q\leftarrow v_1}^{(+,t)}\right)\right.$$

$$+ v_1^\top W_{Q+}^{(t)\top} W_{Q+}^{(t)} c_1 p_{q\leftarrow c_1, k\leftarrow v_1}^{(+,t)} \left(w_+^{(t)\top} v_1 - w_+^{(t)\top} X_{1,+,-} p_{q\leftarrow c_1}^{(+,t)}\right)$$

$$+ v_1^\top W_{Q+}^{(t)\top} W_{Q+}^{(t)} (-v_2) p_{q\leftarrow -v_2, k\leftarrow v_1}^{(+,t)} \left(w_+^{(t)\top} v_1 - w_+^{(t)\top} X_{1,+,-} p_{q\leftarrow -v_2}^{(+,t)}\right)$$

$$- v_1^\top W_{Q+}^{(t)\top} W_{Q+}^{(t)} v_2 p_{q\leftarrow v_2, k\leftarrow -v_1}^{(+,t)} \left(w_+^{(t)\top} (-v_1) - w_+^{(t)\top} X_{2,+,-} p_{q\leftarrow v_2}^{(+,t)}\right)$$

$$- v_1^\top W_{Q+}^{(t)\top} W_{Q+}^{(t)} c_2 p_{q\leftarrow c_2, k\leftarrow -v_1}^{(+,t)} \left(w_+^{(t)\top} (-v_1) - w_+^{(t)\top} X_{2,+,-} p_{q\leftarrow c_2}^{(+,t)}\right)$$

$$\left. - v_1^\top W_{Q+}^{(t)\top} W_{Q+}^{(t)} (-v_1) p_{q\leftarrow -v_1, k\leftarrow -v_1}^{(+,t)} \left(w_+^{(t)\top} (-v_1) - w_+^{(t)\top} X_{2,+,-} p_{q\leftarrow -v_1}^{(+,t)}\right)\right)$$

$$= g^{(t)}\left( v_1^\top W_{Q+}^{(t)\top} W_{Q+}^{(t)} v_1 \left(p_{q\leftarrow v_1, k\leftarrow v_1}^{(+,t)} \left(-w_+^{(t)\top} X_{1,+,-} p_{q\leftarrow v_1}^{(+,t)}\right) + p_{q\leftarrow -v_1, k\leftarrow -v_1}^{(+,t)} \left(-w_+^{(t)\top} X_{2,+,-} p_{q\leftarrow -v_1}^{(+,t)}\right)\right)\right.$$

$$+ v_1^\top W_{Q+}^{(t)\top} W_{Q+}^{(t)} v_1 w_+^{(t)\top} v_1 \left(p_{q\leftarrow v_1, k\leftarrow v_1}^{(+,t)} (X_{1,+,-}) - p_{q\leftarrow -v_1, k\leftarrow -v_1}^{(+,t)} (X_{2,+,-})\right)$$

$$+ v_1^\top W_{Q+}^{(t)\top} W_{Q+}^{(t)} (-v_2) \left(p_{q\leftarrow -v_2, k\leftarrow v_1}^{(+,t)} \left(-w_+^{(t)\top} X_{1,+,-} p_{q\leftarrow -v_2}^{(+,t)}\right) + p_{q\leftarrow v_2, k\leftarrow -v_1}^{(+,t)} \left(-w_+^{(t)\top} X_{2,+,-} p_{q\leftarrow v_2}^{(+,t)}\right)\right)$$

$$+ v_1^\top W_{Q+}^{(t)\top} W_{Q+}^{(t)} (-v_2) w_+^{(t)\top} v_1 \left(p_{q\leftarrow -v_2, k\leftarrow v_1}^{(+,t)} (X_{1,+,-}) - p_{q\leftarrow v_2, k\leftarrow -v_1}^{(+,t)} (X_{2,+,-})\right)$$

$$+ \left(v_1^\top W_{Q+}^{(t)\top} W_{Q+}^{(t)} c_1 p_{q\leftarrow c_1, k\leftarrow v_1}^{(+,t)} + v_1^\top W_{Q+}^{(t)\top} W_{Q+}^{(t)} c_2 p_{q\leftarrow c_2, k\leftarrow -v_1}^{(+,t)}\right) w_+^{(t)\top} v_1$$

$$+ v_1^\top W_{Q+}^{(t)\top} W_{Q+}^{(t)} c_1 p_{q\leftarrow c_1, k\leftarrow v_1}^{(+,t)} \left(-w_+^{(t)\top} X_{1,+,-} p_{q\leftarrow c_1}^{(+,t)}\right)$$

$$\left. - v_1^\top W_{Q+}^{(t)\top} W_{Q+}^{(t)} c_2 p_{q\leftarrow c_2, k\leftarrow -v_1}^{(+,t)} \left(-w_+^{(t)\top} X_{2,+,-} p_{q\leftarrow c_2}^{(+,t)}\right)\right)$$

$$\mathbb{E}\left[g^{(t)} y \dot{\sigma}_+^{(t)} \sum_{l=1}^{L} v_1^\top W_{Q+}^{(t)\top} q_{+l}^{(t)} p_l^{(+,t)\top} \mathrm{diag}\left(w_+^{(t)\top} X - w_+^{(t)\top} X p_l^{(+,t)}\right) X^\top v_1 \middle| X_{k,-,+}\right]$$

$$= g^{(t)}\left( v_1^\top W_{Q+}^{(t)\top} W_{Q+}^{(t)} (-v_1) p_{q\leftarrow -v_1, k\leftarrow -v_1}^{(+,t)} \left(w_+^{(t)\top} (-v_1) - w_+^{(t)} X_{1,-,+} p_{q\leftarrow -v_1}^{(+,t)}\right)\right.$$

$$+ v_1^\top W_{Q+}^{(t)\top} W_{Q+}^{(t)} c_1 p_{q\leftarrow c_1, k\leftarrow -v_1}^{(+,t)} \left(w_+^{(t)\top} (-v_1) - w_+^{(t)} X_{1,-,+} p_{q\leftarrow c_1}^{(+,t)}\right)$$

$$+ v_1^\top W_{Q+}^{(t)\top} W_{Q+}^{(t)} v_2 p_{q\leftarrow v_2, k\leftarrow -v_1}^{(+,t)} \left(w_+^{(t)\top} (-v_1) - w_+^{(t)} X_{1,-,+} p_{q\leftarrow v_2}^{(+,t)}\right)$$

$$- v_1^\top W_{Q+}^{(t)\top} W_{Q+}^{(t)} (-v_2) p_{q\leftarrow -v_2, k\leftarrow v_1}^{(+,t)} \left(w_+^{(t)\top} v_1 - w_+^{(t)} X_{2,-,+} p_{q\leftarrow -v_2}^{(+,t)}\right)$$

$$- v_1^\top W_{Q+}^{(t)\top} W_{Q+}^{(t)} c_2 p_{q\leftarrow c_2, k\leftarrow v_1}^{(+,t)} \left(w_+^{(t)\top} v_1 - w_+^{(t)} X_{2,-,+} p_{q\leftarrow c_2}^{(+,t)}\right)$$

$$\left. - v_1^\top W_{Q+}^{(t)\top} W_{Q+}^{(t)} v_1 p_{q\leftarrow v_1, k\leftarrow v_1}^{(+,t)} \left(w_+^{(t)\top} v_1 - w_+^{(t)} X_{2,-,+} p_{q\leftarrow v_1}^{(+,t)}\right)\right)$$

$$
\begin{aligned}
= g^{(t)} &\bigg( v_1^\top W_{Q+}^{(t)\top} W_{Q+}^{(t)} (-v_1) \left( p_{q\leftarrow -v_1, k\leftarrow -v_1}^{(+,t)} \left( -w_+^{(t)} X_{1,-,+} p_{q\leftarrow -v_1}^{(+,t)} \right) + p_{q\leftarrow v_1, k\leftarrow v_1}^{(+,t)} \left( -w_+^{(t)} X_{2,-,+} p_{q\leftarrow v_1}^{(+,t)} \right) \right) \\
&+ v_1^\top W_{Q+}^{(t)\top} W_{Q+}^{(t)} v_1 w_+^{(t)\top} v_1 \left( p_{q\leftarrow -v_1, k\leftarrow -v_1}^{(+,t)} (X_{1,-,+}) - p_{q\leftarrow v_1, k\leftarrow v_1}^{(+,t)} (X_{2,-,+}) \right) \\
&+ v_1^\top W_{Q+}^{(t)\top} W_{Q+}^{(t)} v_2 \left( p_{q\leftarrow v_2, k\leftarrow -v_1}^{(+,t)} \left( -w_+^{(t)} X_{1,-,+} p_{q\leftarrow v_2}^{(+,t)} \right) + p_{q\leftarrow -v_2, k\leftarrow v_1}^{(+,t)} \left( -w_+^{(t)} X_{2,-,+} p_{q\leftarrow -v_2}^{(+,t)} \right) \right) \\
&+ v_1^\top W_{Q+}^{(t)\top} W_{Q+}^{(t)} v_2 w_+^{(t)\top} (-v_1) \left( p_{q\leftarrow v_2, k\leftarrow -v_1}^{(+,t)} (X_{1,-,+}) - p_{q\leftarrow -v_2, k\leftarrow v_1}^{(+,t)} (X_{2,-,+}) \right) \\
&+ \left( v_1^\top W_{Q+}^{(t)\top} W_{Q+}^{(t)} c_1 p_{q\leftarrow c_1, k\leftarrow -v_1}^{(+,t)} + v_1^\top W_{Q+}^{(t)\top} W_{Q+}^{(t)} c_2 p_{q\leftarrow c_2, k\leftarrow v_1}^{(+,t)} \right) w_+^{(t)\top} (-v_1) \\
&+ v_1^\top W_{Q+}^{(t)\top} W_{Q+}^{(t)} c_1 p_{q\leftarrow c_1, k\leftarrow -v_1}^{(+,t)} \left( -w_+^{(t)} X_{1,-,+} p_{q\leftarrow c_1}^{(+,t)} \right) \\
&- v_1^\top W_{Q+}^{(t)\top} W_{Q+}^{(t)} c_2 p_{q\leftarrow c_2, k\leftarrow v_1}^{(+,t)} \left( -w_+^{(t)} X_{2,-,+} p_{q\leftarrow c_2}^{(+,t)} \right) \bigg)
\end{aligned}
$$

$$
\mathbb{E}\left[ g^{(t)} y \dot{\sigma}_+^{(t)} \sum_{l=1}^L v_1^\top W_{K+}^{(t)\top} K_+^{(t)} \mathrm{diag}\left( X^\top w_+ - w_+^\top X p_l^{(+,t)} \right) p_l^{(+,t)} x_l^\top v_1 \,\middle|\, X_{k,+,-} \right]
$$

$$
\begin{aligned}
= g^{(t)} &\bigg( v_1^\top W_{K+}^{(t)\top} W_{K+}^{(t)} v_1 \left( w_+^{(t)} v_1 - w_+^{(t)\top} X_{1,+,-} p_{q\leftarrow v_1}^{(+,t)} \right) p_{q\leftarrow v_1, k\leftarrow v_1}^{(+,t)} \\
&+ v_1^\top W_{K+}^{(t)\top} W_{K+}^{(t)} c_1 \left( w_+^{(t)} c_1 - w_+^{(t)\top} X_{1,+,-} p_{q\leftarrow v_1}^{(+,t)} \right) p_{q\leftarrow v_1, k\leftarrow c_1}^{(+,t)} \\
&+ v_1^\top W_{K+}^{(t)\top} W_{K+}^{(t)} (-v_2) \left( w_+^{(t)} (-v_2) - w_+^{(t)\top} X_{1,+,-} p_{q\leftarrow v_1}^{(+,t)} \right) p_{q\leftarrow v_1, k\leftarrow -v_2}^{(+,t)} \\
&- v_1^\top W_{K+}^{(t)\top} W_{K+}^{(t)} v_2 \left( w_+^{(t)} v_2 - w_+^{(t)\top} X_{2,+,-} p_{q\leftarrow -v_1}^{(+,t)} \right) p_{q\leftarrow -v_1, k\leftarrow v_2}^{(+,t)} \\
&- v_1^\top W_{K+}^{(t)\top} W_{K+}^{(t)} c_2 \left( w_+^{(t)} c_2 - w_+^{(t)\top} X_{2,+,-} p_{q\leftarrow -v_1}^{(+,t)} \right) p_{q\leftarrow -v_1, k\leftarrow c_2}^{(+,t)} \\
&- v_1^\top W_{K+}^{(t)\top} W_{K+}^{(t)} (-v_1) \left( w_+^{(t)} (-v_1) - w_+^{(t)\top} X_{2,+,-} p_{q\leftarrow -v_1}^{(+,t)} \right) p_{q\leftarrow -v_1, k\leftarrow -v_1}^{(+,t)} \bigg)
\end{aligned}
$$

$$
\begin{aligned}
= g^{(t)} &\bigg( v_1^\top W_{K+}^{(t)\top} W_{K+}^{(t)} v_1 \left( \left( -w_+^{(t)\top} X_{1,+,-} p_{q\leftarrow v_1}^{(+,t)} \right) p_{q\leftarrow v_1, k\leftarrow v_1}^{(+,t)} + \left( -w_+^{(t)\top} X_{2,+,-} p_{q\leftarrow -v_1}^{(+,t)} \right) p_{q\leftarrow -v_1, k\leftarrow -v_1}^{(+,t)} \right) \\
&+ v_1^\top W_{K+}^{(t)\top} W_{K+}^{(t)} v_1 w_+^{(t)} v_1 \left( p_{q\leftarrow v_1, k\leftarrow v_1}^{(+,t)} (X_{1,+,-}) - p_{q\leftarrow -v_1, k\leftarrow -v_1}^{(+,t)} (X_{2,+,-}) \right) \\
&- v_1^\top W_{K+}^{(t)\top} W_{K+}^{(t)} v_2 \left( \left( -w_+^{(t)\top} X_{1,+,-} p_{q\leftarrow v_1}^{(+,t)} \right) p_{q\leftarrow v_1, k\leftarrow -v_2}^{(+,t)} + \left( -w_+^{(t)\top} X_{2,+,-} p_{q\leftarrow -v_1}^{(+,t)} \right) p_{q\leftarrow -v_1, k\leftarrow v_2}^{(+,t)} \right) \\
&+ v_1^\top W_{K+}^{(t)\top} W_{K+}^{(t)} v_2 w_+^{(t)} v_2 \left( p_{q\leftarrow v_1, k\leftarrow -v_2}^{(+,t)} (X_{1,+,-}) - p_{q\leftarrow -v_1, k\leftarrow v_2}^{(+,t)} (X_{2,+,-}) \right) \\
&+ v_1^\top W_{K+}^{(t)\top} W_{K+}^{(t)} c_1 \left( w_+^{(t)} c_1 - w_+^{(t)\top} X_{1,+,-} p_{q\leftarrow v_1}^{(+,t)} \right) p_{q\leftarrow v_1, k\leftarrow c_1}^{(+,t)} \\
&- v_1^\top W_{K+}^{(t)\top} W_{K+}^{(t)} c_2 \left( w_+^{(t)} c_2 - w_+^{(t)\top} X_{2,+,-} p_{q\leftarrow -v_1}^{(+,t)} \right) p_{q\leftarrow -v_1, k\leftarrow c_2}^{(+,t)} \bigg)
\end{aligned}
$$

$$
\mathbb{E}\left[ g^{(t)} y \dot{\sigma}_+^{(t)} \sum_{l=1}^L v_1^\top W_{K+}^{(t)\top} K_+^{(t)} \mathrm{diag}\left( X^\top w_+ - w_+^\top X p_l^{(+,t)} \right) p_l^{(+,t)} x_l^\top v_1 \,\middle|\, X_{k,-,+} \right]
$$

$$
\begin{aligned}
= g^{(t)} &\bigg( v_1^\top W_{K+}^{(t)\top} W_{K+}^{(t)} (-v_1) \left( w_+^{(t)\top} (-v_1) - w_+^{(t)\top} X_{1,-,+} p_{q\leftarrow -v_1}^{(+,t)} \right) p_{q\leftarrow -v_1, k\leftarrow -v_1}^{(+,t)} \\
&+ v_1^\top W_{K+}^{(t)\top} W_{K+}^{(t)} c_1 \left( w_+^{(t)\top} c_1 - w_+^{(t)\top} X_{1,-,+} p_{q\leftarrow -v_1}^{(+,t)} \right) p_{q\leftarrow -v_1, k\leftarrow c_1}^{(+,t)} \\
&+ v_1^\top W_{K+}^{(t)\top} W_{K+}^{(t)} v_2 \left( w_+^{(t)\top} v_2 - w_+^{(t)\top} X_{1,-,+} p_{q\leftarrow -v_1}^{(+,t)} \right) p_{q\leftarrow -v_1, k\leftarrow v_2}^{(+,t)}
\end{aligned}
$$

$$- v_1^\top W_{K+}^{(t)\top} W_{K+}^{(t)} (-v_2) \left( w_+^{(t)\top}(-v_2) - w_+^{(t)\top} X_{2,-,+} p_{q\leftarrow v_1}^{(+,t)} \right) p_{q\leftarrow v_1, k\leftarrow -v_2}^{(+,t)}$$

$$- v_1^\top W_{K+}^{(t)\top} W_{K+}^{(t)} c_2 \left( w_+^{(t)\top} c_2 - w_+^{(t)\top} X_{2,-,+} p_{q\leftarrow v_1}^{(+,t)} \right) p_{q\leftarrow v_1, k\leftarrow c_2}^{(+,t)}$$

$$- v_1^\top W_{K+}^{(t)\top} W_{K+}^{(t)} v_1 \left( w_+^{(t)\top} v_1 - w_+^{(t)\top} X_{2,-,+} p_{q\leftarrow v_1}^{(+,t)} \right) p_{q\leftarrow v_1, k\leftarrow v_1}^{(+,t)} \Bigg)$$

$$= g^{(t)} \Bigg( v_1^\top W_{K+}^{(t)\top} W_{K+}^{(t)} (-v_1) \left( \left( -w_+^{(t)\top} X_{1,-,+} p_{q\leftarrow -v_1}^{(+,t)} \right) p_{q\leftarrow -v_1, k\leftarrow -v_1}^{(+,t)} + \left( -w_+^{(t)\top} X_{2,-,+} p_{q\leftarrow v_1}^{(+,t)} \right) p_{q\leftarrow v_1, k\leftarrow v_1}^{(+,t)} \right)$$

$$+ v_1^\top W_{K+}^{(t)\top} W_{K+}^{(t)} v_1 w_+^{(t)\top} v_1 \left( p_{q\leftarrow -v_1, k\leftarrow -v_1}^{(+,t)}(X_{1,-,+}) - p_{q\leftarrow v_1, k\leftarrow v_1}^{(+,t)}(X_{2,-,+}) \right)$$

$$+ v_1^\top W_{K+}^{(t)\top} W_{K+}^{(t)} v_2 \left( \left( -w_+^{(t)\top} X_{1,-,+} p_{q\leftarrow -v_1}^{(+,t)} \right) p_{q\leftarrow -v_1, k\leftarrow v_2}^{(+,t)} + \left( -w_+^{(t)\top} X_{2,-,+} p_{q\leftarrow v_1}^{(+,t)} \right) p_{q\leftarrow v_1, k\leftarrow -v_2}^{(+,t)} \right)$$

$$+ v_1^\top W_{K+}^{(t)\top} W_{K+}^{(t)} v_2 w_+^{(t)\top} v_2 \left( p_{q\leftarrow -v_1, k\leftarrow v_2}^{(+,t)}(X_{1,-,+}) - p_{q\leftarrow v_1, k\leftarrow -v_2}^{(+,t)}(X_{2,-,+}) \right)$$

$$+ v_1^\top W_{K+}^{(t)\top} W_{K+}^{(t)} c_1 \left( w_+^{(t)\top} c_1 - w_+^{(t)\top} X_{1,-,+} p_{q\leftarrow -v_1}^{(+,t)} \right) p_{q\leftarrow -v_1, k\leftarrow c_1}^{(+,t)}$$

$$- v_1^\top W_{K+}^{(t)\top} W_{K+}^{(t)} c_2 \left( w_+^{(t)\top} c_2 - w_+^{(t)\top} X_{2,-,+} p_{q\leftarrow v_1}^{(+,t)} \right) p_{q\leftarrow v_1, k\leftarrow c_2}^{(+,t)} \Bigg)$$

$$\frac{d v_1^\top W_{K+}^{(t)\top} W_{Q+}^{(t)} c_1}{dt} = \mathbb{E}\left[ g^{(t)} y \dot{\sigma}_+^{(t)} \sum_{l=1}^L c_1^\top W_{Q+}^{(t)\top} q_{+l}^{(t)} p_l^{(+,t)\top} \text{diag}\left( w_+^{(t)\top} X - w_+^{(t)\top} X p_l^{(+,t)} \right) X^\top v_1 \,\Bigg|\, X_{k,s,-s} \right]$$

$$+ \mathbb{E}\left[ g^{(t)} y \dot{\sigma}_+^{(t)} \sum_{l=1}^L v_1^\top W_{K+}^{(t)\top} K_+^{(t)} \text{diag}\left( X^\top w_+ - w_+^\top X p_l^{(+,t)} \right) p_l^{(+,t)} x_l^\top c_1 \,\Bigg|\, X_{k,s,-s} \right]$$

$$\mathbb{E}\left[ g^{(t)} y \dot{\sigma}_+^{(t)} \sum_{l=1}^L c_1^\top W_{Q+}^{(t)\top} q_{+l}^{(t)} p_l^{(+,t)\top} \text{diag}\left( w_+^{(t)\top} X - w_+^{(t)\top} X p_l^{(+,t)} \right) X^\top v_1 \,\Bigg|\, X_{k,+,-} \right]$$

$$= g^{(t)} \Bigg( c_1^\top W_{Q+}^{(t)\top} W_{Q+}^{(t)} v_1 p_{q\leftarrow v_1, k\leftarrow v_1}^{(+,t)} \left( w_+^{(t)\top} v_1 - w_+^{(t)\top} X_{1,+,-} p_{q\leftarrow v_1}^{(+,t)} \right)$$

$$+ c_1^\top W_{Q+}^{(t)\top} W_{Q+}^{(t)} c_1 p_{q\leftarrow c_1, k\leftarrow v_1}^{(+,t)} \left( w_+^{(t)\top} v_1 - w_+^{(t)\top} X_{1,+,-} p_{q\leftarrow c_1}^{(+,t)} \right)$$

$$+ c_1^\top W_{Q+}^{(t)\top} W_{Q+}^{(t)} (-v_2) p_{q\leftarrow -v_2, k\leftarrow v_1}^{(+,t)} \left( w_+^{(t)\top} v_1 - w_+^{(t)\top} X_{1,+,-} p_{q\leftarrow -v_2}^{(+,t)} \right)$$

$$- c_1^\top W_{Q+}^{(t)\top} W_{Q+}^{(t)} v_2 p_{q\leftarrow v_2, k\leftarrow -v_1}^{(+,t)} \left( w_+^{(t)\top} (-v_1) - w_+^{(t)\top} X_{2,+,-} p_{q\leftarrow v_2}^{(+,t)} \right)$$

$$- c_1^\top W_{Q+}^{(t)\top} W_{Q+}^{(t)} c_2 p_{q\leftarrow c_2, k\leftarrow -v_1}^{(+,t)} \left( w_+^{(t)\top} (-v_1) - w_+^{(t)\top} X_{2,+,-} p_{q\leftarrow c_2}^{(+,t)} \right)$$

$$- c_1^\top W_{Q+}^{(t)\top} W_{Q+}^{(t)} (-v_1) p_{q\leftarrow -v_1, k\leftarrow -v_1}^{(+,t)} \left( w_+^{(t)\top} (-v_1) - w_+^{(t)\top} X_{2,+,-} p_{q\leftarrow -v_1}^{(+,t)} \right) \Bigg)$$

$$= g^{(t)} \Bigg( c_1^\top W_{Q+}^{(t)\top} W_{Q+}^{(t)} v_1 \left( p_{q\leftarrow v_1, k\leftarrow v_1}^{(+,t)} \left( -w_+^{(t)\top} X_{1,+,-} p_{q\leftarrow v_1}^{(+,t)} \right) + p_{q\leftarrow -v_1, k\leftarrow -v_1}^{(+,t)} \left( -w_+^{(t)\top} X_{2,+,-} p_{q\leftarrow -v_1}^{(+,t)} \right) \right)$$

$$+ c_1^\top W_{Q+}^{(t)\top} W_{Q+}^{(t)} v_1 w_+^{(t)\top} v_1 \left( p_{q\leftarrow v_1, k\leftarrow v_1}^{(+,t)}(X_{1,+,-}) - p_{q\leftarrow -v_1, k\leftarrow -v_1}^{(+,t)}(X_{2,+,-}) \right)$$

$$+ c_1^\top W_{Q+}^{(t)\top} W_{Q+}^{(t)} c_1 p_{q\leftarrow c_1, k\leftarrow v_1}^{(+,t)} \left( -w_+^{(t)\top} X_{1,+,-} p_{q\leftarrow c_1}^{(+,t)} \right) - c_1^\top W_{Q+}^{(t)\top} W_{Q+}^{(t)} c_2 p_{q\leftarrow c_2, k\leftarrow -v_1}^{(+,t)} \left( -w_+^{(t)\top} X_{2,+,-} p_{q\leftarrow c_2}^{(+,t)} \right)$$

$$+ c_1^\top W_{Q+}^{(t)\top} W_{Q+}^{(t)} (-v_2) \left( p_{q\leftarrow -v_2, k\leftarrow v_1}^{(+,t)} \left( -w_+^{(t)\top} X_{1,+,-} p_{q\leftarrow -v_2}^{(+,t)} \right) + p_{q\leftarrow v_2, k\leftarrow -v_1}^{(+,t)} \left( -w_+^{(t)\top} X_{2,+,-} p_{q\leftarrow v_2}^{(+,t)} \right) \right)$$

$$+ c_1^\top W_{Q+}^{(t)\top} W_{Q+}^{(t)} v_2 w_+^{(t)\top} v_1 \left( -p_{q\leftarrow -v_2, k\leftarrow v_1}^{(+,t)}(X_{1,+,-}) + p_{q\leftarrow v_2, k\leftarrow -v_1}^{(+,t)}(X_{2,+,-}) \right)$$

$$
\left. + c_1^\top W_{Q+}^{(t)\top} W_{Q+}^{(t)} c_1 p_{q\leftarrow c_1, k\leftarrow v_1}^{(+,t)}(X_{1,+,-}) w_+^{(t)\top} v_1 + c_1^\top W_{Q+}^{(t)\top} W_{Q+}^{(t)} c_2 p_{q\leftarrow c_2, k\leftarrow -v_1}^{(+,t)}(X_{2,+,-}) w_+^{(t)\top} v_1 \right)
$$

$$
\mathbb{E}\left[ g^{(t)} y \dot\sigma_+^{(t)} \sum_{l=1}^L c_1^\top W_{Q+}^{(t)\top} q_{+l}^{(t)} p_l^{(+,t)\top} \mathrm{diag}\left( w_+^{(t)\top} X - w_+^{(t)\top} X p_l^{(+,t)} \right) X^\top v_1 \,\middle|\, X_{k,-,+} \right]
$$

$$
= g^{(t)} \left( c_1^\top W_{Q+}^{(t)\top} W_{Q+}^{(t)} (-v_1) p_{q\leftarrow -v_1, k\leftarrow -v_1}^{(+,t)} \left( w_+^{(t)\top}(-v_1) - w_+^{(t)\top} X_{1,-,+} p_{q\leftarrow -v_1}^{(+,t)} \right) \right.
$$

$$
+ c_1^\top W_{Q+}^{(t)\top} W_{Q+}^{(t)} c_1 p_{q\leftarrow c_1, k\leftarrow -v_1}^{(+,t)} \left( w_+^{(t)\top}(-v_1) - w_+^{(t)\top} X_{1,-,+} p_{q\leftarrow c_1}^{(+,t)} \right)
$$

$$
+ c_1^\top W_{Q+}^{(t)\top} W_{Q+}^{(t)} v_2 p_{q\leftarrow v_2, k\leftarrow -v_1}^{(+,t)} \left( w_+^{(t)\top}(-v_1) - w_+^{(t)\top} X_{1,-,+} p_{q\leftarrow v_2}^{(+,t)} \right)
$$

$$
- c_1^\top W_{Q+}^{(t)\top} W_{Q+}^{(t)} (-v_2) p_{q\leftarrow -v_2, k\leftarrow v_1}^{(+,t)} \left( w_+^{(t)\top} v_1 - w_+^{(t)\top} X_{2,-,+} p_{q\leftarrow -v_2}^{(+,t)} \right)
$$

$$
- c_1^\top W_{Q+}^{(t)\top} W_{Q+}^{(t)} c_2 p_{q\leftarrow c_2, k\leftarrow v_1}^{(+,t)} \left( w_+^{(t)\top} v_1 - w_+^{(t)\top} X_{2,-,+} p_{q\leftarrow c_2}^{(+,t)} \right)
$$

$$
\left. - c_1^\top W_{Q+}^{(t)\top} W_{Q+}^{(t)} v_1 p_{q\leftarrow v_1, k\leftarrow v_1}^{(+,t)} \left( w_+^{(t)\top} v_1 - w_+^{(t)\top} X_{2,-,+} p_{q\leftarrow v_1}^{(+,t)} \right) \right)
$$

$$
= g^{(t)} \left( c_1^\top W_{Q+}^{(t)\top} W_{Q+}^{(t)} (-v_1) \left( p_{q\leftarrow -v_1, k\leftarrow -v_1}^{(+,t)} \left( -w_+^{(t)\top} X_{1,-,+} p_{q\leftarrow -v_1}^{(+,t)} \right) + p_{q\leftarrow v_1, k\leftarrow v_1}^{(+,t)} \left( -w_+^{(t)\top} X_{2,-,+} p_{q\leftarrow v_1}^{(+,t)} \right) \right) \right.
$$

$$
+ c_1^\top W_{Q+}^{(t)\top} W_{Q+}^{(t)} (-v_1) w_+^{(t)\top}(-v_1) \left( p_{q\leftarrow -v_1, k\leftarrow -v_1}^{(+,t)}(X_{1,-,+}) - p_{q\leftarrow v_1, k\leftarrow v_1}^{(+,t)}(X_{2,-,+}) \right)
$$

$$
+ c_1^\top W_{Q+}^{(t)\top} W_{Q+}^{(t)} c_1 p_{q\leftarrow c_1, k\leftarrow -v_1}^{(+,t)} \left( -w_+^{(t)\top} X_{1,-,+} p_{q\leftarrow c_1}^{(+,t)} \right) - c_1^\top W_{Q+}^{(t)\top} W_{Q+}^{(t)} c_2 p_{q\leftarrow c_2, k\leftarrow v_1}^{(+,t)} \left( -w_+^{(t)\top} X_{2,-,+} p_{q\leftarrow c_2}^{(+,t)} \right)
$$

$$
+ c_1^\top W_{Q+}^{(t)\top} W_{Q+}^{(t)} v_2 \left( p_{q\leftarrow v_2, k\leftarrow -v_1}^{(+,t)} \left( -w_+^{(t)\top} X_{1,-,+} p_{q\leftarrow v_2}^{(+,t)} \right) + p_{q\leftarrow -v_2, k\leftarrow -v_1}^{(+,t)} \left( -w_+^{(t)\top} X_{2,-,+} p_{q\leftarrow -v_2}^{(+,t)} \right) \right)
$$

$$
+ c_1^\top W_{Q+}^{(t)\top} W_{Q+}^{(t)} v_2 \cdot w_+^{(t)\top}(-v_1) \left( p_{q\leftarrow v_2, k\leftarrow -v_1}^{(+,t)}(X_{1,-,+}) - p_{q\leftarrow -v_2, k\leftarrow v_1}^{(+,t)}(X_{2,-,+}) \right)
$$

$$
\left. + c_1^\top W_{Q+}^{(t)\top} W_{Q+}^{(t)} c_1 p_{q\leftarrow c_1, k\leftarrow -v_1}^{(+,t)}(X_{1,-,+}) w_+^{(t)\top}(-v_1) - c_1^\top W_{Q+}^{(t)\top} W_{Q+}^{(t)} c_2 p_{q\leftarrow c_2, k\leftarrow v_1}^{(+,t)}(X_{2,-,+}) w_+^{(t)\top} v_1 \right)
$$

$$
\mathbb{E}\left[ g^{(t)} y \dot\sigma_+^{(t)} \sum_{l=1}^L v_1^\top W_{K+}^{(t)\top} K_+^{(t)} \mathrm{diag}\left( X^\top w_+ - w_+^\top X p_l^{(+,t)} \right) p_l^{(+,t)} x_l^\top c_1 \,\middle|\, X_{k,s,-s} \right]
$$

$$
= g^{(t)} \left( v_1^\top W_{K+}^{(t)\top} W_{K+}^{(t)} v_1 \left( w_+^{(t)\top} v_1 - w_+^{(t)} X_{1,+,-} p_{q\leftarrow c_1}^{(+,t)} \right) p_{q\leftarrow c_1, k\leftarrow v_1}^{(+,t)} \right.
$$

$$
+ v_1^\top W_{K+}^{(t)\top} W_{K+}^{(t)} c_1 \left( w_+^{(t)\top} c_1 - w_+^{(t)} X_{1,+,-} p_{q\leftarrow c_1}^{(+,t)} \right) p_{q\leftarrow c_1, k\leftarrow c_1}^{(+,t)}
$$

$$
+ v_1^\top W_{K+}^{(t)\top} W_{K+}^{(t)} (-v_2) \left( w_+^{(t)\top}(-v_2) - w_+^{(t)} X_{1,+,-} p_{q\leftarrow c_1}^{(+,t)} \right) p_{q\leftarrow c_1, k\leftarrow -v_2}^{(+,t)}
$$

$$
- v_1^\top W_{K+}^{(t)\top} W_{K+}^{(t)} (-v_1) \left( w_+^{(t)\top}(-v_1) - w_+^{(t)} X_{1,-,+} p_{q\leftarrow c_1}^{(+,t)} \right) p_{q\leftarrow c_1, k\leftarrow -v_1}^{(+,t)}
$$

$$
- v_1^\top W_{K+}^{(t)\top} W_{K+}^{(t)} c_1 \left( w_+^{(t)\top} c_1 - w_+^{(t)} X_{1,-,+} p_{q\leftarrow c_1}^{(+,t)} \right) p_{q\leftarrow c_1, k\leftarrow c_1}^{(+,t)}
$$

$$
\left. - v_1^\top W_{K+}^{(t)\top} W_{K+}^{(t)} v_2 \left( w_+^{(t)\top} v_2 - w_+^{(t)} X_{1,-,+} p_{q\leftarrow c_1}^{(+,t)} \right) p_{q\leftarrow c_1, k\leftarrow v_2}^{(+,t)} \right)
$$

$$
= g^{(t)} \left( v_1^\top W_{K+}^{(t)\top} W_{K+}^{(t)} v_1 \left( \left( -w_+^{(t)} X_{1,+,-} p_{q\leftarrow c_1}^{(+,t)} \right) p_{q\leftarrow c_1, k\leftarrow v_1}^{(+,t)} + \left( -w_+^{(t)} X_{1,-,+} p_{q\leftarrow c_1}^{(+,t)} \right) p_{q\leftarrow c_1, k\leftarrow -v_1}^{(+,t)} \right) \right.
$$

$$
\left. + v_1^\top W_{K+}^{(t)\top} W_{K+}^{(t)} v_1 w_+^{(t)\top} v_1 \left( p_{q\leftarrow c_1, k\leftarrow v_1}^{(+,t)}(X_{1,+,-}) - p_{q\leftarrow c_1, k\leftarrow -v_1}^{(+,t)}(X_{1,-,+}) \right) \right)
$$

$$+ v_1^\top W_{K+}^{(t)\top} W_{K+}^{(t)} (-v_2) \left( \left( -w_+^{(t)} X_{1,+,-} p_{q\leftarrow c_1}^{(+,t)} \right) p_{q\leftarrow c_1, k\leftarrow -v_2}^{(+,t)} + \left( -w_+^{(t)} X_{1,-,+} p_{q\leftarrow c_1}^{(+,t)} \right) p_{q\leftarrow c_1, k\leftarrow v_2}^{(+,t)} \right)$$

$$+ v_1^\top W_{K+}^{(t)\top} W_{K+}^{(t)} v_2 w_+^{(t)\top} v_2 \left( p_{q\leftarrow c_1, k\leftarrow -v_2}^{(+,t)} (X_{1,+,-}) - p_{q\leftarrow c_1, k\leftarrow v_2}^{(+,t)} (X_{1,-,+}) \right)$$

$$+ v_1^\top W_{K+}^{(t)\top} W_{K+}^{(t)} c_1 \left( w_+^{(t)\top} c_1 - w_+^{(t)} X_{1,+,-} p_{q\leftarrow c_1}^{(+,t)} \right) p_{q\leftarrow c_1, k\leftarrow c_1}^{(+,t)} (X_{1,+,-})$$

$$- v_1^\top W_{K+}^{(t)\top} W_{K+}^{(t)} c_1 \left( w_+^{(t)\top} c_1 - w_+^{(t)} X_{1,-,+} p_{q\leftarrow c_1}^{(+,t)} \right) p_{q\leftarrow c_1, k\leftarrow c_1}^{(+,t)} (X_{1,-,+}) \Big)$$

## F  STAGE 3

In stage 3, we show that training the neuron weights can drive the training loss to be arbitrarily small. This is possible since the attention features now are linearly separable with at least a constant margin. This can be seen by considering the neuron weight $w_+ = c_1 + c_2 + v_1 + v_2$ and $w_- = c_1 + c_2 - v_1 - v_2$. Simple calculation can show that this will make $yf(X) \geq \Omega(1)$ for all $(X, y) \in \mathcal{D}$.

**Lemma F.1** (Growth of classification token alignment). *Let* $g_+^{(t)} := g^{(t)}(X_{1,+,+})$ *and* $g_-^{(t)} := g^{(t)}(X_{1,+,-})$. *For* $t \in [T_2, T_3]$, $k \in [2]$,

$$\frac{\mathrm{d} w_+^{(t)\top} v_k}{\mathrm{d} t} = \Theta(g_+^{(t)}) + \widetilde{O}\left( \omega g_-^{(t)} \right).$$

*Proof.* First of all, by symmetry, we have $g_+^{(t)} = g^{(t)}(X_{2,+,+}) = g^{(t)}(X_{1,-,-}) = g^{(t)}(X_{2,-,-})$ and $g_-^{(t)} = g^{(t)}(X_{2,+,-}) = g^{(t)}(X_{1,-,+}) = g^{(t)}(X_{2,-,+})$. In stage 3, we have

$$\frac{\mathrm{d} w_+^{(t)\top} v_1}{\mathrm{d} t}$$

$$= \mathbb{E}\left[ g^{(t)} y \dot{\sigma}_+^{(t)} \sum_{l=1}^{L} \left( X p_l^{(+,t)} \right)^\top \right] v_1$$

$$= \frac{1}{8} g_+^{(t)} \left( p_{q\leftarrow c_1, k\leftarrow v_1}^{(+,t)}(X_{1,+,+}) + p_{q\leftarrow v_1, k\leftarrow v_1}^{(+,t)}(X_{1,+,+}) + p_{q\leftarrow v_2, k\leftarrow v_1}^{(+,t)}(X_{1,+,+}) \right.$$

$$+ \dot{\sigma}_+^{(t)}(X_{1,-,-}) \left( p_{q\leftarrow c_1, k\leftarrow -v_1}^{(+,t)}(X_{1,-,-}) + p_{q\leftarrow -v_1, k\leftarrow -v_1}^{(+,t)}(X_{1,-,-}) + p_{q\leftarrow -v_2, k\leftarrow -v_1}^{(+,t)}(X_{1,-,-}) \right)$$

$$+ p_{q\leftarrow c_2, k\leftarrow v_1}^{(+,t)}(X_{2,+,+}) + p_{q\leftarrow v_1, k\leftarrow v_1}^{(+,t)}(X_{2,+,+}) + p_{q\leftarrow v_2, k\leftarrow v_1}^{(+,t)}(X_{2,+,+})$$

$$+ \dot{\sigma}_+^{(t)}(X_{2,-,-}) \left. \left( p_{q\leftarrow c_2, k\leftarrow -v_1}^{(+,t)}(X_{2,-,-}) + p_{q\leftarrow -v_1, k\leftarrow -v_1}^{(+,t)}(X_{2,-,-}) + p_{q\leftarrow -v_2, k\leftarrow -v_1}^{(+,t)}(X_{2,-,-}) \right) \right)$$

$$+ \frac{1}{8} g_-^{(t)} \left( p_{q\leftarrow c_1, k\leftarrow v_1}^{(+,t)}(X_{1,+,-}) + p_{q\leftarrow v_1, k\leftarrow v_1}^{(+,t)}(X_{1,+,-}) + p_{q\leftarrow -v_2, k\leftarrow v_1}^{(+,t)}(X_{1,+,-}) \right.$$

$$+ p_{q\leftarrow c_1, k\leftarrow -v_1}^{(+,t)}(X_{1,-,+}) + p_{q\leftarrow -v_1, k\leftarrow -v_1}^{(+,t)}(X_{1,-,+}) + p_{q\leftarrow v_2, k\leftarrow -v_1}^{(+,t)}(X_{1,-,+})$$

$$- p_{q\leftarrow c_2, k\leftarrow -v_1}^{(+,t)}(X_{2,+,-}) - p_{q\leftarrow v_1, k\leftarrow -v_1}^{(+,t)}(X_{2,+,-}) - p_{q\leftarrow v_2, k\leftarrow -v_1}^{(+,t)}(X_{2,+,-})$$

$$- p_{q\leftarrow c_2, k\leftarrow v_1}^{(+,t)}(X_{2,-,+}) - p_{q\leftarrow v_1, k\leftarrow v_1}^{(+,t)}(X_{2,-,+}) - p_{q\leftarrow -v_2, k\leftarrow v_1}^{(+,t)}(X_{2,-,+}) \right)$$

By Theorem E.7, we have

$$p_{q\leftarrow c_1, k\leftarrow v_1}^{(+,t)}(X_{1,+,+}) + p_{q\leftarrow v_1, k\leftarrow v_1}^{(+,t)}(X_{1,+,+}) + p_{q\leftarrow v_2, k\leftarrow v_1}^{(+,t)}(X_{1,+,+})$$

$$+ \dot{\sigma}_+^{(t)}(X_{1,-,-}) \left( p_{q\leftarrow c_1, k\leftarrow -v_1}^{(+,t)}(X_{1,-,-}) + p_{q\leftarrow -v_1, k\leftarrow -v_1}^{(+,t)}(X_{1,-,-}) + p_{q\leftarrow -v_2, k\leftarrow -v_1}^{(+,t)}(X_{1,-,-}) \right)$$

$$+ p_{q\leftarrow c_2, k\leftarrow v_1}^{(+,t)}(X_{2,+,+}) + p_{q\leftarrow v_1, k\leftarrow v_1}^{(+,t)}(X_{2,+,+}) + p_{q\leftarrow v_2, k\leftarrow v_1}^{(+,t)}(X_{2,+,+})$$

$$+ \dot{\sigma}_+^{(t)}(X_{2,-,-}) \left( p_{q\leftarrow c_2, k\leftarrow -v_1}^{(+,t)}(X_{2,-,-}) + p_{q\leftarrow -v_1, k\leftarrow -v_1}^{(+,t)}(X_{2,-,-}) + p_{q\leftarrow -v_2, k\leftarrow -v_1}^{(+,t)}(X_{2,-,-}) \right)$$

$$= \Theta(1)$$

and by symmetry between two groups Lemma C.2

$$
\begin{aligned}
& p^{(+,t)}_{q\leftarrow c_1, k\leftarrow v_1}(X_{1,+,-}) + p^{(+,t)}_{q\leftarrow v_1, k\leftarrow v_1}(X_{1,+,-}) + p^{(+,t)}_{q\leftarrow -v_2, k\leftarrow v_1}(X_{1,+,-}) \\
& \quad + p^{(+,t)}_{q\leftarrow c_1, k\leftarrow -v_1}(X_{1,-,+}) + p^{(+,t)}_{q\leftarrow -v_1, k\leftarrow -v_1}(X_{1,-,+}) + p^{(+,t)}_{q\leftarrow v_2, k\leftarrow -v_1}(X_{1,-,+}) \\
& \quad - p^{(+,t)}_{q\leftarrow c_2, k\leftarrow -v_1}(X_{2,+,-}) - p^{(+,t)}_{q\leftarrow -v_1, k\leftarrow -v_1}(X_{2,+,-}) - p^{(+,t)}_{q\leftarrow v_2, k\leftarrow -v_1}(X_{2,+,-}) \\
& \quad - p^{(+,t)}_{q\leftarrow c_2, k\leftarrow v_1}(X_{2,-,+}) - p^{(+,t)}_{q\leftarrow v_1, k\leftarrow v_1}(X_{2,-,+}) - p^{(+,t)}_{q\leftarrow -v_2, k\leftarrow v_1}(X_{2,-,+}) \\
& = \widetilde{O}(\omega)
\end{aligned}
$$

Thus,

$$
\frac{\mathrm{d}w_+^{(t)\top} v_1}{\mathrm{d}t} = \Theta(g_+^{(t)}) + \widetilde{O}\left(\omega g_-^{(t)}\right).
$$

$\square$

**Lemma F.2** (Growth of group token alignment). *Let $g_+^{(t)} := g^{(t)}(X_{1,+,+})$ and $g_-^{(t)} := g^{(t)}(X_{1,+,-})$. For $t \in [T_2, T_3]$, $k \in [2]$,*

$$
\frac{\mathrm{d}w_+^{(t)\top} c_k}{\mathrm{d}t} \geq \Omega(g_-^{(t)}) + \widetilde{O}\left(\omega g_+^{(t)}\right)
$$

*Proof.* In stage 3, by Theorem E.7, we have

$$
\begin{aligned}
& \frac{\mathrm{d}w_+^{(t)\top} c_1}{\mathrm{d}t} \\
& = \mathbb{E}\left[g^{(t)} y \dot{\sigma}_+^{(t)} \sum_{l=1}^L \left(X p_l^{(+,t)}\right)^\top\right] c_1 \\
& = \frac{1}{8} g_+^{(t)} \left( p^{(+,t)}_{q\leftarrow c_1, k\leftarrow c_1}(X_{1,+,+}) + p^{(+,t)}_{q\leftarrow v_1, k\leftarrow c_1}(X_{1,+,+}) + p^{(+,t)}_{q\leftarrow v_2, k\leftarrow c_1}(X_{1,+,+}) \right. \\
& \qquad \left. - \dot{\sigma}_+^{(t)}(X_{1,-,-}) \left( p^{(+,t)}_{q\leftarrow c_1, k\leftarrow c_1}(X_{1,-,-}) + p^{(+,t)}_{q\leftarrow -v_1, k\leftarrow c_1}(X_{1,-,-}) + p^{(+,t)}_{q\leftarrow -v_2, k\leftarrow c_1}(X_{1,-,-}) \right) \right) \\
& \quad + \frac{1}{8} g_-^{(t)} \left( p^{(+,t)}_{q\leftarrow c_1, k\leftarrow c_1}(X_{1,+,-}) + p^{(+,t)}_{q\leftarrow v_1, k\leftarrow c_1}(X_{1,+,-}) + p^{(+,t)}_{q\leftarrow -v_2, k\leftarrow c_1}(X_{1,+,-}) \right. \\
& \qquad \left. - p^{(+,t)}_{q\leftarrow c_1, k\leftarrow c_1}(X_{1,-,+}) - p^{(+,t)}_{q\leftarrow -v_1, k\leftarrow c_1}(X_{1,-,+}) - p^{(+,t)}_{q\leftarrow v_2, k\leftarrow c_1}(X_{1,-,+}) \right) \\
& \geq \Theta(g_-^{(t)}) + \widetilde{O}\left(\omega g_+^{(t)}\right)
\end{aligned}
$$

$\square$

**Theorem F.3** (Convergence). *There exists $T_3$ such that $T_3 - T_2 = O(1/\epsilon)$ and $\mathbb{E}[\ell(y f^{(t)}(X))] \leq \epsilon$.*

*Proof.* Let $g_+^{(t)} := g^{(t)}(X_{1,+,+})$ and $g_-^{(t)} := g^{(t)}(X_{1,+,-})$. For consistent samples $X_{k,s,s}$, we have

$$
\begin{aligned}
\frac{\mathrm{d}f^{(t)}(X_{1,+,+})}{\mathrm{d}t} &= \frac{\mathrm{d}w_+^{(t)\top} v_1}{\mathrm{d}t} \left( p^{(+,t)}_{q\leftarrow c_1, k\leftarrow v_1} + p^{(+,t)}_{q\leftarrow v_1, k\leftarrow v_1} + p^{(+,t)}_{q\leftarrow v_2, k\leftarrow v_1} \right) \\
&\quad + \frac{\mathrm{d}w_+^{(t)\top} v_2}{\mathrm{d}t} \left( p^{(+,t)}_{q\leftarrow c_1, k\leftarrow v_2} + p^{(+,t)}_{q\leftarrow v_1, k\leftarrow v_2} + p^{(+,t)}_{q\leftarrow v_2, k\leftarrow v_2} \right) \\
&\quad + \frac{\mathrm{d}w_+^{(t)\top} c_1}{\mathrm{d}t} \left( p^{(+,t)}_{q\leftarrow c_1, k\leftarrow c_1} + p^{(+,t)}_{q\leftarrow v_1, k\leftarrow c_1} + p^{(+,t)}_{q\leftarrow v_2, k\leftarrow c_1} \right) \\
&\quad - \frac{\mathrm{d}w_-^{(t)\top} v_1}{\mathrm{d}t} \dot{\sigma}_-^{(t)} \left( p^{(-,t)}_{q\leftarrow c_1, k\leftarrow v_1} + p^{(-,t)}_{q\leftarrow v_1, k\leftarrow v_1} + p^{(-,t)}_{q\leftarrow v_2, k\leftarrow v_1} \right)
\end{aligned}
$$

$$- \frac{\mathrm{d}w_-^{(t)\top} v_2}{\mathrm{d}t} \dot{\sigma}_-^{(t)} \left( p_{q \leftarrow c_1, k \leftarrow v_2}^{(-,t)} + p_{q \leftarrow v_1, k \leftarrow v_2}^{(-,t)} + p_{q \leftarrow v_2, k \leftarrow v_2}^{(-,t)} \right)$$

$$- \frac{\mathrm{d}w_-^{(t)\top} c_1}{\mathrm{d}t} \dot{\sigma}_-^{(t)} \left( p_{q \leftarrow c_1, k \leftarrow c_1}^{(-,t)} + p_{q \leftarrow v_1, k \leftarrow c_1}^{(-,t)} + p_{q \leftarrow v_2, k \leftarrow c_1}^{(-,t)} \right)$$

By symmetry, we have

$$- \frac{\mathrm{d}w_-^{(t)\top} v_1}{\mathrm{d}t} \left( p_{q \leftarrow c_1, k \leftarrow v_1}^{(-,t)}(X_{1,+,+}) + p_{q \leftarrow v_1, k \leftarrow v_1}^{(-,t)}(X_{1,+,+}) + p_{q \leftarrow v_2, k \leftarrow v_1}^{(-,t)}(X_{1,+,+}) \right)$$

$$- \frac{\mathrm{d}w_-^{(t)\top} v_2}{\mathrm{d}t} \left( p_{q \leftarrow c_1, k \leftarrow v_2}^{(-,t)}(X_{1,+,+}) + p_{q \leftarrow v_1, k \leftarrow v_2}^{(-,t)}(X_{1,+,+}) + p_{q \leftarrow v_2, k \leftarrow v_2}^{(-,t)}(X_{1,+,+}) \right)$$

$$- \frac{\mathrm{d}w_-^{(t)\top} c_1}{\mathrm{d}t} \left( p_{q \leftarrow c_1, k \leftarrow c_1}^{(-,t)}(X_{1,+,+}) + p_{q \leftarrow v_1, k \leftarrow c_1}^{(-,t)}(X_{1,+,+}) + p_{q \leftarrow v_2, k \leftarrow c_1}^{(-,t)}(X_{1,+,+}) \right)$$

$$= \frac{\mathrm{d}w_+^{(t)\top} v_1}{\mathrm{d}t} \left( p_{q \leftarrow c_1, k \leftarrow -v_1}^{(+,t)}(X_{1,-,-}) + p_{q \leftarrow -v_1, k \leftarrow -v_1}^{(+,t)}(X_{1,-,-}) + p_{q \leftarrow -v_2, k \leftarrow -v_1}^{(+,t)}(X_{1,-,-}) \right)$$

$$+ \frac{\mathrm{d}w_+^{(t)\top} v_2}{\mathrm{d}t} \left( p_{q \leftarrow c_1, k \leftarrow -v_2}^{(+,t)}(X_{1,-,-}) + p_{q \leftarrow -v_1, k \leftarrow -v_2}^{(+,t)}(X_{1,-,-}) + p_{q \leftarrow -v_2, k \leftarrow -v_2}^{(+,t)}(X_{1,-,-}) \right)$$

$$- \frac{\mathrm{d}w_+^{(t)\top} c_1}{\mathrm{d}t} \left( p_{q \leftarrow c_1, k \leftarrow c_1}^{(+,t)}(X_{1,-,-}) + p_{q \leftarrow -v_1, k \leftarrow c_1}^{(+,t)}(X_{1,-,-}) + p_{q \leftarrow -v_2, k \leftarrow c_1}^{(+,t)}(X_{1,-,-}) \right)$$

Thus,

$$\frac{\mathrm{d}f^{(t)}(X_{1,+,+})}{\mathrm{d}t} = \Omega \left( \frac{\mathrm{d}w_+^{(t)\top} v_1}{\mathrm{d}t} + \frac{\mathrm{d}w_+^{(t)\top} v_2}{\mathrm{d}t} \right) + \widetilde{O} \left( \omega \frac{\mathrm{d}w_-^{(t)\top} c_1}{\mathrm{d}t} \right)$$

$$= \Omega(g_+^{(t)}) + \widetilde{O} \left( \omega g_-^{(t)} \right) \qquad \text{(by Lemma F.1)}$$

For conflicting samples $X_{k,s,-s}$, we have

$$\frac{\mathrm{d}f^{(t)}(X_{1,+,-})}{\mathrm{d}t} = \frac{\mathrm{d}w_+^{(t)\top} v_1}{\mathrm{d}t} \left( p_{q \leftarrow c_1, k \leftarrow v_1}^{(+,t)} + p_{q \leftarrow v_1, k \leftarrow v_1}^{(+,t)} + p_{q \leftarrow -v_2, k \leftarrow v_1}^{(+,t)} \right)$$

$$- \frac{\mathrm{d}w_+^{(t)\top} v_2}{\mathrm{d}t} \left( p_{q \leftarrow c_1, k \leftarrow -v_2}^{(+,t)} + p_{q \leftarrow v_1, k \leftarrow -v_2}^{(+,t)} + p_{q \leftarrow -v_2, k \leftarrow -v_2}^{(+,t)} \right)$$

$$+ \frac{\mathrm{d}w_+^{(t)\top} c_1}{\mathrm{d}t} \left( p_{q \leftarrow c_1, k \leftarrow c_1}^{(+,t)} + p_{q \leftarrow v_1, k \leftarrow c_1}^{(+,t)} + p_{q \leftarrow -v_2, k \leftarrow c_1}^{(+,t)} \right)$$

$$+ \frac{\mathrm{d}w_-^{(t)\top} v_1}{\mathrm{d}t} \dot{\sigma}_-^{(t)} \left( p_{q \leftarrow c_1, k \leftarrow v_1}^{(-,t)} + p_{q \leftarrow v_1, k \leftarrow v_1}^{(-,t)} + p_{q \leftarrow -v_2, k \leftarrow v_1}^{(-,t)} \right)$$

$$- \frac{\mathrm{d}w_-^{(t)\top} v_2}{\mathrm{d}t} \dot{\sigma}_-^{(t)} \left( p_{q \leftarrow c_1, k \leftarrow -v_2}^{(-,t)} + p_{q \leftarrow v_1, k \leftarrow -v_2}^{(-,t)} + p_{q \leftarrow -v_2, k \leftarrow -v_2}^{(-,t)} \right)$$

$$- \frac{\mathrm{d}w_-^{(t)\top} c_1}{\mathrm{d}t} \dot{\sigma}_-^{(t)} \left( p_{q \leftarrow c_1, k \leftarrow c_1}^{(-,t)} + p_{q \leftarrow v_1, k \leftarrow c_1}^{(-,t)} + p_{q \leftarrow -v_2, k \leftarrow c_1}^{(-,t)} \right)$$

$$= \Omega \left( \frac{\mathrm{d}w_-^{(t)\top} c_1}{\mathrm{d}t} \right) + \widetilde{O} \left( \omega \frac{\mathrm{d}w_+^{(t)\top} v_1}{\mathrm{d}t} \right)$$

$$= \Omega(g_-^{(t)}) + \widetilde{O} \left( \omega g_+^{(t)} \right) \qquad \text{(by Lemma F.2)}$$

Thus,

$$\frac{\mathrm{d}\mathbb{E}[\ell(yf^{(t)}(X))]}{\mathrm{d}t} = \frac{1}{2} \frac{\mathrm{d}\ell(f^{(t)}(X_{1,+,+}))}{\mathrm{d}t} + \frac{1}{2} \frac{\mathrm{d}\ell(f^{(t)}(X_{1,+,-}))}{\mathrm{d}t}$$

$$= \frac{1}{2} \ell'(f^{(t)}(X_{1,+,+})) \frac{\mathrm{d}f^{(t)}(X_{1,+,+})}{\mathrm{d}t} + \frac{1}{2} \ell'(f^{(t)}(X_{1,+,-})) \frac{\mathrm{d}f^{(t)}(X_{1,+,-})}{\mathrm{d}t}$$

$$= -\Omega\left((g_+^{(t)})^2 + (g_-^{(t)})^2\right)$$

$$= -\Omega\left(\ell^2(f^{(t)}(X_{1,+,+})) + \ell^2(f^{(t)}(X_{1,+,-}))\right)$$

which implies that

$$\mathbb{E}[\ell(yf^{(t)}(X))] \leq O\left(\frac{1}{t - T_2 + 1}\right).$$

$\square$

## G    HANDLING THE FINITE-WIDTH DISCRETIZATION

In this section, we describe how to incorporate the finite-width discretization. As we mentioned in the beginning of Section 5, the gradient flow for the finite width transformer can be seen as the gradient flow for infinite-width transformer model with perturbation. Thus, we need to calculate the deviation of the finite-width gradient flow from the infinite-width gradient flow. Since we only train the neuron weights in stage 1 and stage 3, we only need to consider how the finite-width discretization will affect stage 2.

First, at initialization, by the standard concentration bound, we have the following:

**Lemma G.1.** *With probability at least $1 - \delta$ over the randomness of the initialization of $W_K, W_Q$, for any $\mu, \nu \in \mathcal{C} \cup \mathcal{V}$,*

$$\left|\left\langle W_K^{(0)}\mu, W_Q^{(0)}\nu\right\rangle\right| \leq \omega^2\left(\sqrt{\frac{4}{m}\log\frac{8}{\delta}} + \frac{4}{m}\log\frac{8}{\delta}\right)$$

$$\left|\left\langle W_K^{(0)}\mu, W_K^{(0)}\nu\right\rangle\right| \leq \omega^2\left(\sqrt{\frac{4}{m}\log\frac{8}{\delta}} + \frac{4}{m}\log\frac{8}{\delta}\right),$$

$$\left|\left\langle W_Q^{(0)}\mu, W_Q^{(0)}\nu\right\rangle\right| \leq \omega^2\left(\sqrt{\frac{4}{m}\log\frac{8}{\delta}} + \frac{4}{m}\log\frac{8}{\delta}\right),$$

*and for any $\mu \in \mathcal{C} \cup \mathcal{V}$,*

$$\|W_K^{(0)}\mu\|_2^2 = \omega^2\left(1 \pm \left(\sqrt{\frac{4}{m}\log\frac{8}{\delta}} + \frac{4}{m}\log\frac{8}{\delta}\right)\right)$$

$$\|W_Q^{(0)}\mu\|_2^2 = \omega^2\left(1 \pm \left(\sqrt{\frac{4}{m}\log\frac{8}{\delta}} + \frac{4}{m}\log\frac{8}{\delta}\right)\right)$$

*Proof.* Since $W_K^{(0)}\mu, W_Q^{(0)}\nu \sim \mathcal{N}(0, \omega^2 I)$, we apply Lemma H.2 to get the corresponding bound. $\square$

Next, recall that from Section 5.2 the score and self-score variables evolve according to the dynamical system

$$\frac{\mathrm{d}}{\mathrm{d}t}\begin{bmatrix}\mathrm{vec}(\{\mu^\top W_{K+}^{(t)\top} W_{Q+}^{(t)}\nu\}_{\mu,\nu\in\mathcal{C}\cup\mathcal{V}}) \\ \mathrm{vec}(\{\mu^\top W_{K+}^{(t)\top} W_{K+}^{(t)}\nu\}_{\mu,\nu\in\mathcal{C}\cup\mathcal{V}}) \\ \mathrm{vec}(\{\mu^\top W_{Q+}^{(t)\top} W_{Q+}^{(t)}\nu\}_{\mu,\nu\in\mathcal{C}\cup\mathcal{V}})\end{bmatrix} = A^{(t)}\begin{bmatrix}\mathrm{vec}(\{\mu^\top W_{K+}^{(t)\top} W_{Q+}^{(t)}\nu\}_{\mu,\nu\in\mathcal{C}\cup\mathcal{V}}) \\ \mathrm{vec}(\{\mu^\top W_{K+}^{(t)\top} W_{K+}^{(t)}\nu\}_{\mu,\nu\in\mathcal{C}\cup\mathcal{V}}) \\ \mathrm{vec}(\{\mu^\top W_{Q+}^{(t)\top} W_{Q+}^{(t)}\nu\}_{\mu,\nu\in\mathcal{C}\cup\mathcal{V}})\end{bmatrix}$$

for some matrix $A^{(t)}$ with $\|A^{(t)}\|_\infty = O(1)$. By Lemma E.3, stage 2 lasts $O(\log(1/\omega))$ time. Then, we can bound the effect of the perturbation to the dynamical system as follows. Consider the dynamical system

$$\frac{dx(t)}{dt} = A(t)x(t), \quad x(0) = x_0$$

and the perturbed dynamical system

$$\frac{dx_\epsilon(t)}{dt} = (A(t) + \Delta A(t))x_\epsilon(t), \quad x_\epsilon(0) = x_0 + \epsilon$$

where $\epsilon$ and $\Delta A(t)$ are the perturbation introduced by the finite-width discretization. Notice that we have $||\Delta A(s)|| \le c||\delta x(s)||$ for some constant $c$ for all $s \in [T_1, T_2]$. Define $\delta x(t) = x_\epsilon(t) - x(t)$. We have

$$\frac{d\delta x(t)}{dt} = A(t)\delta x(t) + \Delta A(t)x_\epsilon(t)$$

This implies that

$$||\delta x(t)|| \le \int_0^t ||A(s)\delta x(s) + \Delta A(s)x_\epsilon(s)||ds$$

$$\le \int_0^t ||A(s)||||\delta x(s)|| + ||\Delta A(s)||||x_\epsilon(s)||ds$$

$$\le \int_0^t ||\delta x(s)||(||A(s)|| + c||x_\epsilon(s)||)ds$$

Applying Grönwall's inequality, we have

$$||\delta x(t)|| \le ||\delta x(0)|| \exp\left(\int_0^t ||A(s)|| + c||x_\epsilon(s)||\right) = \text{poly}(1/\omega)/\sqrt{m}$$

since $||A(t)|| \le O(1)$ and $||x_\epsilon(s)|| \le O(1)$ and $t \le O(\log(1/\omega))$. Thus, picking $m \ge \text{poly}(1/\omega)$ can make the perturbation to the gradient flow system at most $O(\omega)$.

**Proposition G.2.** *For $t \le T_2$,*

$$||x_\epsilon(t)|| \le O(1).$$

*Proof.* We are going to show that in the infinite-width case, $||x(t)|| = O(1)$ for $t \le T_2$. Later, we are going to show that $||\delta x(t)|| < 1$ and by triangle inequality, we have $||x_\epsilon(t)|| \le O(1)$. The key observation is to use Lemma E.1 which provides an upper bound on the score variables after stage 2. However, there is an issue: $x$ will also contain self-score variables whereas Lemma E.1 only provides an upper bound on the score variables.

To solve this issue, we refine our analysis for the infinite-width case. We are going to show that $v_1^\top W_{Q+}^{(t)\top} W_{Q+}^{(t)} v_1$ and $c_1^\top W_{K+}^{(t)\top} W_{K+}^{(t)} c_1$ are within (multiplicative) constant factor of $c_1^\top W_{K+}^{(t)\top} W_{Q+}^{(t)} v_1$. First of all, a direct consequence of Lemma E.1 is that it also provides an upper bound of the growth of the 3 dominating variables and thus Lemma E.2 can be modified as

$$\frac{d}{dt} \begin{bmatrix} c_1^\top W_{K+}^{(t)\top} W_{Q+}^{(t)} v_1 \\ c_1^\top W_{K+}^{(t)\top} W_{K+}^{(t)} c_1 \\ v_1^\top W_{Q+}^{(t)\top} W_{Q+}^{(t)} v_1 \end{bmatrix} = \Theta(1) \begin{bmatrix} 0 & 1 & 1 \\ 1 & 0 & 0 \\ 1 & 0 & 0 \end{bmatrix} \begin{bmatrix} c_1^\top W_{K+}^{(t)\top} W_{Q+}^{(t)} v_1 \\ c_1^\top W_{K+}^{(t)\top} W_{K+}^{(t)} c_1 \\ v_1^\top W_{Q+}^{(t)\top} W_{Q+}^{(t)} v_1 \end{bmatrix}$$

Notice that there is a self-balancing property of this dynamical system: there exists a constant $C > 0$ such that if $c_1^\top W_{K+}^{(t)\top} W_{K+}^{(t)} c_1 > C \cdot c_1^\top W_{K+}^{(t)\top} W_{Q+}^{(t)} v_1$, then $\frac{d}{dt} c_1^\top W_{K+}^{(t)\top} W_{Q+}^{(t)} v_1 > \frac{d}{dt} c_1^\top W_{K+}^{(t)\top} W_{K+}^{(t)} c_1$. Thus, if those variables are already within constant factor of each other, they will maintain this relationship.

Then, the base case can be established by the existence of a time $T_{1.5}$ such that $T_{1.5} - T_1 = \Theta(1)$ and $c_1^\top W_{K+}^{(T_{1.5})\top} W_{Q+}^{(T_{1.5})} v_1, c_1^\top W_{K+}^{(T_{1.5})\top} W_{K+}^{(T_{1.5})} c_1, v_1^\top W_{Q+}^{(T_{1.5})\top} W_{Q+}^{(T_{1.5})} v_1 = \Theta(\omega^2)$. This finishes the proof of $v_1^\top W_{Q+}^{(t)\top} W_{Q+}^{(t)} v_1$ and $c_1^\top W_{K+}^{(t)\top} W_{K+}^{(t)} c_1$ are within constant factor of $c_1^\top W_{K+}^{(t)\top} W_{Q+}^{(t)} v_1$ during stage 2.

Finally, for the finite-width case, as long as $||\delta x(t)|| \le 1$ for $t \le T_2$, we can apply Grönwall's inequality as before and get $||\delta x(t)|| \le \text{poly}(1/\omega)/\sqrt{m} < 1$ as long as $m \ge \text{poly}(1/\omega)$. This argument is proved in a way similar to Section 1.3 in Tao (2006). $\qquad\square$

# H  PROBABILITY

**Lemma H.1** (Bernstein's inequality for bounded random variables)**.** *Assume $Z_1, \ldots, Z_n$ are $n$ i.i.d. random variables with $\mathbb{E}[Z_i] = 0$ and $|Z_i| \leq M$ for all $i \in [n]$ almost surely. Let $Z = \sum_{i=1}^{n} Z_i$. Then, for all $t > 0$,*

$$\mathbb{P}[Z > t] \leq \exp\left(-\frac{t^2/2}{\sum_{j=1}^{n} \mathbb{E}[Z_j^2] + Mt/3}\right) \leq \exp\left(-\min\left\{\frac{t^2}{2\sum_{j=1}^{n} \mathbb{E}[Z_j^2]}, \frac{t}{2M}\right\}\right)$$

*which implies with probability at least $1 - \delta$,*

$$Z \leq \sqrt{2\sum_{j=1}^{n} \mathbb{E}[Z_j^2] \log\frac{1}{\delta}} + 2M \log\frac{1}{\delta}.$$

**Lemma H.2.** *For $w_1, w_2 \in \mathbb{R}^m$ with $w_1, w_2 \overset{i.i.d.}{\sim} \mathcal{N}(0, I_m/m)$,*

$$\mathbb{P}\left[\left|\|w_1\|_2^2 - 1\right| \geq \sqrt{\frac{4}{m}\log\frac{2}{\delta}} + \frac{4}{m}\log\frac{2}{\delta}\right] \leq \delta$$

$$\mathbb{P}\left[|\langle w_1, w_2\rangle| \geq \sqrt{\frac{4}{m}\log\frac{2}{\delta}} + \frac{4}{m}\log\frac{2}{\delta}\right] \leq \delta$$

*Proof.* We have

$$\mathbb{E}\left[\|w_1\|_2^2\right] = \mathbb{E}\left[\sum_{i=1}^{m} w_{1,i}^2\right] = 1$$

Notice that $w_{1,i}^2$ is a sub-Gamma random variable with parameters $(\frac{4}{m^2}, \frac{4}{m})$. Thus, by Bernstein's inequality,

$$\mathbb{P}\left[\left|\|w_1\|_2^2 - \mathbb{E}\left[\|w_1\|_2^2\right]\right| \geq \sqrt{\frac{4}{m}\log\frac{2}{\delta}} + \frac{4}{m}\log\frac{2}{\delta}\right] \leq \delta$$

Next,

$$\mathbb{E}[\langle w_1, w_2\rangle] = \mathbb{E}\left[\sum_{i=1}^{m} w_{1,i} w_{2,i}\right] = 0$$

By Bernstein's inequality,

$$\mathbb{P}\left[|\langle w_1, w_2\rangle| \geq \sqrt{\frac{4}{m}\log\frac{2}{\delta}} + \frac{4}{m}\log\frac{2}{\delta}\right] \leq \delta$$

$\square$

