# OpenReview forum: "Transformers Provably Learn Two-Mixture of Linear Classification via Gradient Flow"
_ICLR.cc/2025/Conference — ICLR 2025 Poster_

### Official Review · Reviewer_7eXW · 2024-11-01

**Soundness:** 3
**Presentation:** 3
**Contribution:** 3
**Rating:** 8
**Confidence:** 4

**Summary:**

The paper analyzes training dynamics of a single attention layer two-headed transformer on a specific task. The task -- a two-mixture of linear classification -- possesses a hidden correspondence structure between the feature vectors in the sequence. Each sample contains an indicator vector that dictates, sign of which of the other feature vectors in the sequence is important. Thus, to efficiently solve the task, a communication between the class indicator and the rest of the feature vectors is necessary. The authors first observe the training dynamics empirically, specifying three distinct training phases. Based on these observation they define a training algorithm similar to standard full-batch gradient flow, for which they prove that the transformer not only learns the task with perfect accuracy, but, more importantly, learns to effectively utilize the hidden structure in a human interpretable way. Finally, the authors discuss the extension to more than just two-mixture of linear classification, constructing a transformer architecture that solves the task, but showing that a gradient-based analysis would need to utilize more refined proof techniques.

**Strengths:**

What I particularly like about the paper is the very well chosen combination of data distribution, architecture and proof assumptions. This combination is perhaps a minimal example on which it is possible to demonstrate this phenomenon of transformers learning to utilize the hidden communication between tokens at this level of complexity. On the other hand, this setting is perhaps a maximal example under which the proof still remains practically doable and comprehensible. Therefore I find the setting of the paper very suitable.

Unlike many other papers studying transformers, the advantage of this work is that the studied task and the provided results are simple enough that they provide human-interpretable understanding of the transformers' behavior, that agrees with our expectations -- the attention layers are crucial to provide the necessary communication between tokens and they can learn to do so. Further, the importance of the use of multiple heads for certain tasks is instructively demonstrated.

The authors also do a very good job extracting the essence of otherwise long and tedious proof into the main body and make it understandable for a wide audience. I think the presentation improves reader's own understanding of the inner workings of transformers and allows the reader to re-use some of the techniques in their own transformer research. The presented analysis and the core proof idea makes sense and nicely captures the important aspects of the optimization.

**Weaknesses:**

- W1: While I appreciate the setting of the paper in general, there are a few questionable decisions when it comes to architecture. In particular, in the main part of the paper, it is a bit unusual that the value matrix basically maps to 1D outputs and the representations are averaged even before applying ReLU. I would expect averaging only after the application of the ReLU and also like to see what would happen if more than 1D value matrices would be used. However, bigger issue comes with Lemma 6.1. In particular, the order in which the operations are made in the transformer defined in the proof is so unusual I am not sure it can be interpreted as transformer anymore. The summation over $l_2$ should definitely come as the inner-most, otherwise we cannot even talk about attention. Then should come the summation over $k$, only after that should come the application of $\sigma$ and the summation over $l_1$ should be the outer-most. I would ask the authors to redo the statement and proof of Lemma 6.1 so that the summation over $l_2$ is the inner-most or otherwise justify their architectural choice.

- W2: Although the presented phenomenon is intuitive, it is unclear *if* and *how* does it generalize to more general settings. In particular to multi-mixture of linear regression, where I would expect (almost) the same thing happen. It is unclear why the symmetry is broken and therefore it is also unclear whether the described phenomenon even holds the same way in multi-class setting. The same question holds for another natural extensions of the considered setting -- multi-class classification, presence of noise, non-orthogonality of features or class imbalance and more natural weight initialization. I wouldn't mind that the paper only studies the simplest setting if there was evidence that the knowledge obtained can be transferred to the more general settings but in this case it is not clear whether the same mechanism even works in extended settings.

- W3: Some aspects of the presentation should be improved. To provide a couple of examples:

- *It could be clearer if authors re-named classes to something else as they are confused with the classification classes.*
- *The assumption of averaged initial value model is not discussed enough. In particular it is not clear where any asymmetry at all comes from if we have this assumption.*
- *The proof in appendix should be more verbose and make some more explicit statements as intermediate steps. For instance, the separability of the consistent samples after phase 2 is only proved implicitly and is never explicitly stated in the proof. This makes it hard to make look-ups of specific proof parts mentioned in the main body, because the proof itself is an ocean of infinite formulas.*
- *The use of $g$ is somewhat peculiar. I suspect the authors to either make a few typos when using $g$ or not to define it properly.*

**Typos:**

- The property 4 in Definition 2.1 is inconsistent with the verbal description in the rest of the paper as this way, you would leave most of the positions actually empty (zeros).

- line 170: You should write $\nu$ instead of $X^{(i)}$.

- In lemma 5.2, the use of $g$ is somewhat weird. Please double-check what you meant there. The same holds for lines 427, 430

- Line 408: based on my counting the numbers should be 16, 10, 10. Correct me if I am wrong.

- In line 60 you talk about class identifier getting all the signal from classification features, but later in your theory you consider the class identifier's role as the key to be crucial. I think it better to rewrite this sentence so that the roles are exchanged.

**Questions:**

- Q1: Have you tried to measure this phenomenon and the training dynamics for any of the setting extensions I mentioned in W2?

- Q2: In Figure 1, we see that the neuron alignment grows with $c_k$ immediately from the beginning. This seems to be in contradiction with line 367, especially since I believe the weight initialization in experiments was the same as in the theory. What makes this difference?

- Q3: In figure 1c, how would the curves for $c_1$ being query look like? Can you provide a figure with these included?

- Q4: Where do the slight asymmetries in Theorem 4.1 come from when you consider the averaged initial value model?

**Summary:** I consider this paper to be a very instructive, novel, theoretical contribution. It allows us to understand transformer training dynamics in a human-interpretable way that aligns with our natural understanding of what attention should be learning to do. This highly precise description of the role of attention in simple transformers makes the paper a strong contribution. Therefore, I recommend accepting the paper. However, my high score is conditioned on the authors addressing my concerns in the discussion period.

---

> ### Author Response · Authors · 2024-11-22
> **Response (1/)**
>
> We thank the reviewer for providing the highly inspiring feedback. In the revised paper, we have made changes based on your review comments and have highlighted all revisions in blue fonts.
>
>
> **W1** While I appreciate the setting of the paper in general, there are a few questionable decisions when it comes to architecture. In particular, in the main part of the paper, it is a bit unusual that the value matrix basically maps to 1D outputs and the representations are averaged even before applying ReLU. I would expect averaging only after the application of the ReLU and also like to see what would happen if more than 1D value matrices would be used. However, bigger issue comes with Lemma 6.1. In particular, the order in which the operations are made in the transformer defined in the proof is so unusual I am not sure it can be interpreted as transformer anymore. The summation over $l_2$ should definitely come as the inner-most, otherwise we cannot even talk about attention. Then should come the summation over $k$, only after that should come the application of $\sigma$ and the summation over $l_1$ should be the outer-most. I would ask the authors to redo the statement and proof of Lemma 6.1 so that the summation over $l_2$ is the inner-most or otherwise justify their architectural choice.
>
> **A** Thank you for this suggestion!
> We made a new construction in the revision in Lemma 6.1 and we highlight the new construction in blue.
>
> **W2** Although the presented phenomenon is intuitive, it is unclear if and how does it generalize to more general settings. In particular to multi-mixture of linear regression, where I would expect (almost) the same thing happen. It is unclear why the symmetry is broken and therefore it is also unclear whether the described phenomenon even holds the same way in multi-class setting. The same question holds for another natural extensions of the considered setting -- multi-class classification, presence of noise, non-orthogonality of features or class imbalance and more natural weight initialization. I wouldn't mind that the paper only studies the simplest setting if there was evidence that the knowledge obtained can be transferred to the more general settings but in this case it is not clear whether the same mechanism even works in extended settings.
>
> **A** Great question! We first point out that the technique we used in our analysis can be applied to the settings the reviewer mentioned. We next discuss the extensions that the reviewer points out.
>
> **Multi-mixture of linear regression:** We further investigated why the asymmetries between groups arise in the scores for 3 groups. We found out that the asymmetry is due to the randomness in the scores at initialization. We tried to increase the embedding dimension of $W_K, W_Q$ and the asymmetries between groups start to go away, see Figure 5(c) and Figure 5(d). (In the infinite-width transformer in our revision, the randomness will be reduced if we increase the width.) We further investigated the multi-group case and we found that with more groups, the magnitudes of between-group scores like $c_2^\top W_{K+}^{(t)\top} W_{Q+}^{(t)} v_1$ will become smaller and smaller with more groups and the magnitudes of same-group scores such as $ c_1^\top W_{K+}^{(t)\top} W_{Q+}^{(t)} v_1$ remain big, see Figure 5(d), 6(b), 7(b) in Appendix A.4.
> On the other hand, our techniques are generalizable to study this setting.
>
> **Multi-class classification setting:**
> This is indeed an interesting extension and our proof technique can be adapted to study this setting.
> In this case, the transformer model needs to be modified to output a vector instead of a single scalar in the binary case.
> We expect that the layer-wise learning behavior (i.e., first neurons learn and then the attention learns) will still occur in the multi-class classification setting.
> On the other hand, how the softmax attention will behave in this setting needs further investigation.
>
> **Presence of noise, non-orthogonality of features or class imbalance and more natural weight
> initialization:** We expect that the training dynamics is similar when the data contains appropriate amount of Gaussian noise or we use Gaussian initialization with sufficiently small variance to initialize the neuron weight. On the other hand, non-orthogonality of the signals or class imbalance will definitely break the symmetry and the training dynamics will be certainly different.
> For example, (Huang, Cheng \& Liang, 2024) studied the class imbalanced case in in-context learning and they found that class imbalance will introduce more phases in training dynamics. We also expect that more training phases will arise due to data dominance/under-representation.
>
> Huang, Yu, Yuan Cheng, and Yingbin Liang. "In-context Convergence of Transformers." Forty-first International Conference on Machine Learning, 2024.

---

> ### Author Response · Authors · 2024-11-22
> **Response (2/)**
>
> **W3** Some aspects of the presentation should be improved. To provide a couple of examples:
>
> - **Q** It could be clearer if authors re-named classes to something else as they are confused with the classification classes.
> **A** Thank you for your suggestion. We have changed \``class" to \``group".
>
> - **Q** The assumption of averaged initial value model is not discussed enough. In particular it is not clear where any asymmetry at all comes from if we have this assumption.
> **A** Thank you for pointing this out. We also realized that the averaged initial value model is an unnatural assumption.
> In our revision, we were able to remove this assumption via mean-field technique by considering an infinite-width transformer, which is considered to be a more natural and easily justifiable technique in theoretical literature.
> With infinite-width transformer, the averaged initial value model assumption can be exactly satisfied at initialization, see Lemma C.1 in Appendix C.
> We have checked that replacing the original averaged initial value model by infinite-width transformer doesn't introduce much changes in the proof. The discretization to finite-width case can be handled easily in Appendix G.
> Furthermore, by adapting to the mean-field technique, we remove the $1/\sqrt{m}$ factor in the softmax attention and we initialize the attention weights by $\mathcal{N}(0, \omega^2 \frac{1}{m})$.
> One of the consequences of removing the $1/\sqrt{m}$ factor in the softmax attention is that now stage 2 of the training becomes shorter and only runs in $O(\log (1/\omega))$ time.
> We also updated the experiment plots in Section 3.1 accordingly to reflect this change.
>
> - **Q** The proof in appendix should be more verbose and make some more explicit statements as intermediate steps. For instance, the separability of the consistent samples after phase 2 is only proved implicitly and is never explicitly stated in the proof. This makes it hard to make look-ups of specific proof parts mentioned in the main body, because the proof itself is an ocean of infinite formulas.
> **A** Thank you for your suggestions.
> First of all, the separability of the data can be seen by considering the neuron weight $w_+ = c_1 + c_2 + v_1 + v_2$ and $w_- = c_1 + c_2 - v_1 - v_2$ which can create a constant margin. We have made this statement more precise in the proof.
> In the meantime, we have added more explanation in various places in our proof.
> We will continue to polish our proof in the Appendix.
>
> - **Q** The use of $g$ is somewhat peculiar. I suspect the authors to either make a few typos when using $g$ or not to define it properly.
> **A** The function $g$ is defined on line 175 as the negative gradient of the cross entropy loss.
>
> **Typo 1** The property 4 in Definition 2.1 is inconsistent with the verbal description in the rest of the paper as this way, you would leave most of the positions actually empty (zeros).
>
> **A** Thank you for pointing it out!
> You are right.
> What we should write is for each $l_2 \in [L] \setminus \{l_0, l_1\}$, sample $k' \sim \textnormal{Uniform}([K] \setminus \{k\})$ and set $X_{l_2} = \epsilon v_{k'}$, where $\epsilon \stackrel{i.i.d.}{\sim} \textnormal{Uniform}(\{\pm 1\})$.
>
> **Typo 2** line 170: You should write $\nu$ instead of $X^{(i)}$.
>
> **A** The input to the softmax function should be a vector and thus it should be $X$ (notice that $X^\top W_K^\top W_Q \mu$ is a vector and $l(\nu)$ will index the corresponding entry to $\nu$).
>
> **Typo 3** In lemma 5.2, the use of $g$ is somewhat weird. Please double-check what you meant there. The same holds for lines 427, 430.
>
> **A** In Lemma 5.2, we have $g(X_{k,s,s'}) = -\ell'(f(X_{k,s,s'}))$.
> On line 427 and 430, we use $g^{(t)}$ to denote $g(f^{(t)}(X_{k,s,-s}))$ (by symmetry in the averaged initial value model (old version) and infinite-width transformer (revised version), $f^{(t)}(X_{k,s,-s})$ are the same for $k \in [2], s \in \{\pm 1\}$).
>
> **Typo 4** Line 408: based on my counting the numbers should be 16, 10, 10. Correct me if I am wrong.
>
> **A** Yes! You are right about the counting. Thanks for pointing it out!
>
> **Typo 5** In line 60 you talk about class identifier getting all the signal from classification features, but later in your theory you consider the class identifier's role as the key to be crucial. I think it better to rewrite this sentence so that the roles are exchanged.
>
> **A** Thank you for the suggestion!
> We have modified our introduction accordingly.

---

> ### Author Response · Authors · 2024-11-22
> **Response (3/)**
>
> **Q1** Have you tried to measure this phenomenon and the training dynamics for any of the setting extensions I mentioned in W2?
>
> **A** Please see our response in W2.
> We have conducted experiments on the multi-mixture setting, and the results can be found in Appendix A.4.
>
>
> **Q2** In Figure 1, we see that the neuron alignment grows with $c_k$ immediately from the beginning. This seems to be in contradiction with line 367, especially since I believe the weight initialization in experiments was the same as in the theory. What makes this difference?
>
>
> **A** The neuron alignment for $c_k$ indeed stays flat at the beginning.
> Because the duration is not very long, it is not very obvious on the plot.
>
>
> **Q3** In figure 1c, how would the curves for $c_1$ being query look like? Can you provide a figure with these included?
>
> **A** The shape of the curve for $c_1$ being query is similar to the curve for $v_1$ being query.
> The difference is that the magnitude of the scores for $c_1$ being query is a lot smaller than the scores for $v_1$ being the query.
> We have included a plot with $c_1$ being query in Figure 4 in Appendix section A.3 in the revision.
>
> **Q4** Where do the slight asymmetries in Theorem 4.1 come from when you consider the averaged initial value model?
>
> **A** The asymmetries are defined in Lemma E.4 in the Appendix.
> In Lemma E.4, besides the scores, we also need symmetries between the self-score variables such as $v_1^\top W_{Q+}^{(T_2) \top} W_{Q+}^{(T_2)} v_1 \approx - v_1^\top W_{Q+}^{(T_2) \top} W_{Q+}^{(T_2)} v_2$.
> The slight asymmetries come from the initialization in both the averaged initial value model (original version) and the infinite-width transformer model (revised version), since $ v_1^\top W_{Q+}^{(0) \top} W_{Q+}^{(0)} v_1 > 0$ whereas $ v_1^\top W_{Q+}^{(0) \top} W_{Q+}^{(0)} v_2 = 0$.
> Thus, the *exact* symmetry cannot hold and the asymmetries in the self-score variables will propagate to the score variables during training.
> Fortunately, we are able to show that these asymmetries will not become very big after the training.

---

> > ### Comment · Reviewer_7eXW · 2024-11-25
> > **Thank you for the answers**
> >
> > Thank you for the detailed answers. I am convinced by most of the points. I will just follow-up on a few things:
> >
> > - The theoretical changes you made are very profound. To my taste it is a bit too much for the rebuttal. It also raises the questions of the width under which the mean-field approximation becomes close to the discrete dynamics. On the other hand it is an improvement compared with the averaged initial value model.
> >
> > - Regarding your answer to my Q2, how many steps are we talking about here? I zoomed in as much as the PDF allowed me to but still couldn't see the plateau.
> >
> > - In your new theorem 4.1, you still have one appearance of $m.$ What does this value mean in the infinite-width context?

---

> > > ### Author Response · Authors · 2024-11-26
> > > **Thank you for your follow-up**
> > >
> > > Thank you for your response.
> > >
> > > **Q** It also raises the questions of the width under which the mean-field approximation becomes close to the discrete dynamics.
> > >
> > > **A** The finite-width discretization will introduce some perturbation to the dynamical system at the initialization.
> > > The hardest part is to bound how this perturbation will affect the gradient flow dynamics in stage 2.
> > > First of all, at initialization, those perturbation can be characterized by standard concentration results in Lemma G.1, which is at most $O(\omega^2 /\sqrt{m})$.
> > > Then, we can bound the effect of the perturbation to the dynamical system as follows.
> > > Consider the dynamical system
> > > \begin{align*}
> > >     \frac{d x(t)}{dt} = A(t) x(t), \quad x(0) = x_0
> > > \end{align*}
> > > and the perturbed dynamical system
> > > \begin{align*}
> > >     \frac{d x_\epsilon (t)}{dt} = (A(t) + \Delta A(t)) x_\epsilon(t), \quad x_\epsilon(0) = x_0 + \epsilon
> > > \end{align*}
> > > where $\epsilon$ and $\Delta A(t)$ are the perturbation introduced by the finite-width discretization.
> > > Notice that we have $\lvert| \Delta A(s) \rvert| \leq c \lvert| \delta x(s) \rvert|$ for some constant $c$.
> > > Define $\delta x(t) = x_\epsilon(t) - x(t)$. We have
> > > \begin{align*}
> > >     \frac{d \delta x(t)}{dt} = A(t) \delta x(t) + \Delta A(t) x_\epsilon(t)
> > > \end{align*}
> > > This implies that
> > > \begin{align*}
> > >     \lvert| \delta x(t) \rvert| &\leq \int_0^t \lvert| A(s) \delta x(s) + \Delta A(s) x_\epsilon(s) \rvert| ds \\\\
> > >     &\leq \int_0^t \lvert| A(s) \rvert| \lvert| \delta x(s) \rvert| + \lvert| \Delta A(s) \rvert| \lvert| x_\epsilon(s) \rvert| ds \\\\
> > >     &\leq \int_0^t \lvert| \delta x(s) \rvert| (\lvert| A(s) \rvert| + c\lvert| x_\epsilon(s) \rvert|) ds
> > > \end{align*}
> > > Applying Gronwall's inequality, we have
> > > \begin{align*}
> > >     \lvert| \delta x(t) \rvert| \leq \lvert| \delta x(0) \rvert| \exp\left( \int_0^t \lvert| A(s) \rvert| + c\lvert| x_\epsilon(s) \rvert|  ds\right) = \textnormal{poly}(1/\omega) / \sqrt{m}
> > > \end{align*}
> > > since $\lvert| A(t) \rvert| \leq O(1)$ and $\lvert| x_\epsilon(s) \rvert| \leq O(1)$ and $t \leq O(\log(1/\omega))$.
> > > Thus, picking $m \geq \textnormal{poly}(1/\omega)$ makes the perturbation to the gradient flow system at most $O(\omega)$, which is desirable in the context of our proof.
> > >
> > > This analysis is also included in Appendix G.
> > >
> > > **Q** Regarding your answer to my Q2, how many steps are we talking about here? I zoomed in as much as the PDF allowed me to but still couldn't see the plateau.
> > >
> > > **A** We have a plot about the early neuron alignment with the signals.
> > > The plateau is only about 20 steps and it is indeed hard to see on the full plot.
> > >
> > > **Q** In your new theorem 4.1, you still have one appearance of $m.$ What does this value mean in the infinite-width context?
> > >
> > > **A** Thank you for pointing it out. It is a typo. We have corrected the form in the revision.

---

> > > > ### Comment · Reviewer_7eXW · 2024-11-26
> > > >
> > > > Thank you for the detailed explanation! Just to double-check - how do we know $||x_\epsilon(s)||\le O(1)$?

---

> > > > > ### Author Response · Authors · 2024-11-28
> > > > >
> > > > > Good catch! The condition $\lvert| x_\epsilon(t) \rvert| \leq O(1)$ in our previous response indeed cannot be inferred from the original analysis.
> > > > > In the worst case, we will have $\lvert| x(t) \rvert| = O(1/\omega)$ which leads to $\lvert| x_\epsilon(t) \rvert| = O(1/\omega)$.
> > > > >
> > > > > Fortunately, there is a work-around on this.
> > > > > We are going to show that in the infinite-width case, $\lvert| x(t) \rvert| = O(1)$ for $t \leq T_2$.
> > > > > Later, we are going to show that $\lvert| \delta x(t) \rvert| < 1$ and by triangle inequality, we have $\lvert| x_\epsilon(t) \rvert| \leq O(1)$.
> > > > > The key observation is to use Lemma E.1 which provides an upper bound on the score variables after stage 2.
> > > > > However, there is an issue: $x$ will also contain self-score variables whereas Lemma E.1 only provides an upper bound on the score variables.
> > > > >
> > > > > To solve this issue, we refine our analysis for the infinite-width case.
> > > > > We are going to show that $ v_1^\top W_{Q+}^{(t)\top} W_{Q+}^{(t)} v_1$ and $ c_1^\top W_{K+}^{(t)\top} W_{K+}^{(t)} c_1$ are within (multiplicative) constant factor of $c_1^\top W_{K+}^{(t)\top} W_{Q+}^{(t)} v_1$.
> > > > > First of all, a direct consequence of Lemma E.1 is that it also provides an upper bound of the growth of the 3 dominating variables and thus Lemma E.2 can be modified as
> > > > > \begin{align*}
> > > > >     \frac{d}{dt}
> > > > >     \begin{bmatrix}
> > > > >         c_1^\top W_{K+}^{(t)\top} W_{Q+}^{(t)} v_1 \\\\
> > > > >         c_1^\top W_{K+}^{(t)\top} W_{K+}^{(t)} c_1 \\\\
> > > > >         v_1^\top W_{Q+}^{(t)\top} W_{Q+}^{(t)} v_1 \\\\
> > > > >     \end{bmatrix}
> > > > >     = \Theta(1)
> > > > >     \begin{bmatrix}
> > > > >         0 & 1 & 1 \\\\
> > > > >         1 & 0 & 0 \\\\
> > > > >         1 & 0 & 0
> > > > >     \end{bmatrix}
> > > > >     \begin{bmatrix}
> > > > >         c_1^\top W_{K+}^{(t)\top} W_{Q+}^{(t)} v_1 \\\\
> > > > >         c_1^\top W_{K+}^{(t)\top} W_{K+}^{(t)} c_1 \\\\
> > > > >         v_1^\top W_{Q+}^{(t)\top} W_{Q+}^{(t)} v_1 \\\\
> > > > >     \end{bmatrix}
> > > > > \end{align*}
> > > > > Notice that there is a self-balancing property of this dynamical system: there exists a constant $C>0$ such that if $c_1^\top W_{K+}^{(t)\top} W_{K+}^{(t)} c_1 > C \cdot c_1^\top W_{K+}^{(t)\top} W_{Q+}^{(t)} v_1$, then $\frac{d}{dt} c_1^\top W_{K+}^{(t)\top} W_{Q+}^{(t)} v_1 > \frac{d}{dt} c_1^\top W_{K+}^{(t)\top} W_{K+}^{(t)} c_1 $.
> > > > > Thus, if those variables are already within constant factor of each other, they will maintain this relationship.
> > > > >
> > > > > Then, the base case can be established by the existence of a time $T_{1.5}$ such that $T_{1.5} - T_1 = \Theta(1)$ and $c_1^\top W_{K+}^{(T_{1.5})\top} W_{Q+}^{(T_{1.5})} v_1, c_1^\top W_{K+}^{(T_{1.5})\top} W_{K+}^{(T_{1.5})} c_1, v_1^\top W_{Q+}^{(T_{1.5})\top} W_{Q+}^{(T_{1.5})} v_1 = \Theta(\omega^2)$.
> > > > > This finishes the proof of $ v_1^\top W_{Q+}^{(t)\top} W_{Q+}^{(t)} v_1$ and $ c_1^\top W_{K+}^{(t)\top} W_{K+}^{(t)} c_1$ are within constant factor of $c_1^\top W_{K+}^{(t)\top} W_{Q+}^{(t)} v_1$ during stage 2.
> > > > >
> > > > > Finally, for the finite-width case, as long as $\lvert| \delta x(t) \rvert| \leq 1$ for $t \leq T_2$, we can apply Gr&ouml;nwall's inequality as before and get $\lvert| \delta x(t) \rvert| \leq \textnormal{poly}(1/\omega)/\sqrt{m} < 1$ as long as $m \geq \textnormal{poly}(1/\omega)$.

---

> ### Comment · Reviewer_7eXW · 2024-11-28
>
> Thank you. There are a few unclear steps in this answer.
>
> - As far as I understand Lemma E.1 does not prove an upper-bound on the score variables, instead it assumes they are bounded and proves some relative size bounds. How can we see the boundedness of the score variables from Lemma E.1?
>
> - Next, in the penultimate paragraph you say that the base case can be established by the existence of $T_{1.5}.$ How do you prove that such a time exists?
>
> - Finally, in the last paragraph you assume $||\delta x(t)||\le 1$ but that itself is waht you get in the end of the sentence and also what you need in your agrument in the first place. So it seems you are assuming what you want to prove.
>
> Please clarify these points.

---

> > ### Author Response · Authors · 2024-12-01
> >
> > Thank you very much for engaging in further discussion.
> >
> > **Q** As far as I understand Lemma E.1 does not prove an upper-bound on the score variables, instead it assumes they are bounded and proves some relative size bounds. How can we see the boundedness of the score variables from Lemma E.1?
> >
> > **A** In our analysis of stage 2, we pick a specific part of gradient flow training such that the score variables are within those bound. Then the analysis is based on such bounds.
> >
> > **Q** Next, in the penultimate paragraph you say that the base case can be established by the existence of $T_{1.5}.$ How do you prove that such a time exists?
> >
> > **A** Notice that $\left. \frac{d}{dt} c\_1^\top W\_{K+}^{(t)\top} W\_{Q+}^{(t)} v\_1 \right|\_{t=T_1} = \Theta(\omega^2)$.
> > From the $3\times 3$ dynamical system, as long as $ c_1^\top W_{K+}^{(t)\top} W_{Q+}^{(t)} v_1 = O(\omega^2)$, there exists a time $T^\star$ such that $T^\star - T_1 = \Theta(1)$ such that $||c_1^\top W_{K+}^{(t)\top} W_{K+}^{(t)} c_1||, ||v_1^\top W_{Q+}^{(t)\top} W_{Q+}^{(t)} v_1|| = \Theta(\omega^2)$ for $t \leq T^\star$ (recall that the two values at initialization are $\Theta(\omega^2)$ by concentration).
> > Take $T_{1.5} = (T^\star - T_1) / 2 + T_1$ finishes the proof since $ \frac{d}{dt} c_1^\top W_{K+}^{(t)\top} W_{Q+}^{(t)} v_1 = \Theta(\omega^2)$ for $T_1 \leq t \leq T_{1.5}$.
> >
> > **Q** Finally, in the last paragraph you assume $||\delta x(t)||\le 1$ but that itself is what you get in the end of the sentence and also what you need in your argument in the first place. So it seems you are assuming what you want to prove.
> >
> > **A** It is not a circular argument. We started with the assumption that $||\delta x(t)|| < 1$ (where $||\delta x(0)|| < 1$ can be proved by concentration at initialization). We then proved $||\delta x(t)|| < \textnormal{poly}(1/\omega)/\sqrt{m} \leq O(\omega)$ if $m \geq \textnormal{poly}(1/\omega)$, which is stronger than $||\delta x(t)|| <1$.
> > Such a way of proof is similar to Section 1.3 in (Tao, 2006).
> >
> > Tao, Terence. Nonlinear dispersive equations: local and global analysis. No. 106. American Mathematical Soc., 2006. URL: https://www.math.ucla.edu/~tao/preprints/chapter.pdf

---

> > > ### Comment · Reviewer_7eXW · 2024-12-02
> > >
> > > Thank you for the further explanations. Regarding the last question, you are right, I didn't think it through properly. I also agree with the answers of the first two questions, except that it was perhaps helpful to mention lemma E.3 in the appendix that guarantees the first point and provides similar argumentation as for the second point.
> > >
> > > With these questions settled I would like to thank the authors again for an in-depth discussion. I am keeping the score.

---

> > > > ### Author Response · Authors · 2024-12-02
> > > >
> > > > Thank you for the enlightening discussion. Your comments and suggestions are highly helpful to improve our work. We will make sure to include all your suggestions during rebuttal discussion into our final version.

---

### Official Review · Reviewer_qpZd · 2024-11-02

**Soundness:** 3
**Presentation:** 3
**Contribution:** 3
**Rating:** 6
**Confidence:** 3

**Summary:**

This paper analyzes training dynamics of transformer estimated via gradient flow under the mixture of linear classification setting. Based on the empirical findings, they construct their own three-stage training procedures, which essentially give the same training dynamics with actual gradient descent simultaneously train all the parameters in Transformers.

**Strengths:**

The paper provides rigorous theoretical analysis on the training dynamics of transformer which is a highly non-trivial task.

**Weaknesses:**

Please see the below questions:

**Questions:**

A.	I had trouble understanding the setting of mixture of linear classification. Authors only spend one paragraph in the 2nd page discussing about the model. But I think it should be elaborated more in details to motivate the readers.

B.	I might have missed the part while reading the paper, but I wonder how training dynamics discovered in the paper corresponds to the actual reading procedures of the human beings? For instance, using the example in the paper: “Bob is watching TV in the living room. Where is Bob?” How do people catch the correspondence between Where and Living room and how does this correspond to the training dynamics in the paper? I wonder if authors have any thoughts on this.

C.	I think this is important question: is the training dynamics in the paper also observable in the vanilla transformer case? The authors consider two-headed symmetric transformers, but in real practice vanilla transformer is commonly used. If this is the case, I think this paper’s contribution is significant.

D.	Do authors have any insights on the multiple layer cases?

**Details Of Ethics Concerns:**

This is a theoretical paper, therefore there is no ethical concerns regarding the contents of the paper.

---

> ### Author Response · Authors · 2024-11-22
>
> We thank Reviewer qpZd for the review.
> In the revised paper, we have made changes based on the reviewers' comments and have highlighted all revisions in blue fonts.
>
> **Q** I had trouble understanding the setting of mixture of linear classification. Authors only spend one paragraph in the 2nd page discussing about the model. But I think it should be elaborated more in details to motivate the readers.
>
> **A** Thank you for pointing this out. Below the formal definition of the mixture of linear classification task in Definition 2.1, we elaborate our explanation for the data model with an example. We also provide a high level description of the mixture of linear classification task in the introduction in our revision.
>
> **Q** I might have missed the part while reading the paper, but I wonder how training dynamics discovered in the paper corresponds to the actual reading procedures of the human beings? For instance, using the example in the paper: “Bob is watching TV in the living room. Where is Bob?” How do people catch the correspondence between Where and Living room and how does this correspond to the training dynamics in the paper? I wonder if authors have any thoughts on this.
>
> **A** Great question! For this example, the supervised loss function will be small if the answer to \``where" is \``living room". Our analysis of gradient flow suggests that the gradient of the loss with respect to the attention matrix will point to the direction to align the query token of \``where" with the key token of \``living" and \``room" during the training process, thus driving the attention matrix to align those two. This will ultimately derive the loss to be small and provide good answer. Such process is consistent with our theoretical analysis of the behavior of the attention matrix.
>
>
> We point out that the example given in the introduction serves as an *motivating* example to justify our choice of mixture of linear classification task for studying how transformers utilize connections between words to solve certain tasks. In this example, there is a connection between the word \``where" and \``living room".
>
>
> **Q** Is the training dynamics in the paper also observable in the vanilla transformer case? The authors consider two-headed symmetric transformers, but in real practice vanilla transformer is commonly used. If this is the case, I think this paper’s contribution is significant.
>
> **A** Thank you for the question.
> In our work we observed layer-wise learning behavior (in our case, neuron weights learn first and then the softmax attention modules learn) during transformer training. Similar behavior has been observed in previous work such as [1], where the authors discovered that there is a layer-wise learning behavior for multi-layer transformer. Further, our insight about the training dynamics of the self-attention modules, where the attention matrices are driven by gradient flow to align relevant tokens for classifications, will also carry over to multi-layer transformers. Such an observation has been made in [2].
>
> We would also like to point out that the purpose of our work is to understand how transformers especially the self-attention modules leverage the connections between tokens to solve certain task.
> At the same time, we aim to provide theoretical tools that can serve as building blocks and are generalizable to other settings when we analyze the training dynamics of transformers.
>
>
> **Q** Do authors have any insights on the multiple layer cases?
>
> **A** Our work currently focuses on a single layer transformer, and we discovered a layer-wise learning behavior in our setting.
> Similar layer-wise learning behavior has been observed previously in [1]. Further, our insight about the training dynamics of the self-attention modules, where the attention matrices are driven by gradient flow to align relevant tokens for classifications, will also carry over to multi-layer transformers. Such an observation has been made in [2].
> We also believe that our techniques and tools developed in this work can be generalized to study more complicated architectures like multi-layer transformers.
>
> [1] Nichani, Eshaan, Alex Damian, and Jason D. Lee. "How Transformers Learn Causal Structure with Gradient Descent." Forty-first International Conference on Machine Learning.
>
> [2] Yuandong Tian, Yiping Wang, Zhenyu Zhang, Beidi Chen, Simon Shaolei Du. JoMA: Demystifying Multilayer Transformers via Joint Dynamics of MLP and Attention, ICLR 2024.

---

### Official Review · Reviewer_Kkd8 · 2024-11-03

**Soundness:** 2
**Presentation:** 2
**Contribution:** 2
**Rating:** 6
**Confidence:** 3

**Summary:**

This paper studies the training dynamics of a two-headed transformer using a three-stage, layer-wise training algorithm. It further characterizes the feature learning process and captures the attention mechanism involved in solving the task.

**Strengths:**

This paper presents a rigorous theoretical framework for understanding the training dynamics of non-linear Transformer architectures.

The most interesting finding, in my view, is how the authors analyze the progressive emergence of connections across diverse data types at various stages, offering a comprehensive perspective on the layered learning process within Transformers. Specifically, the study demonstrates that weights in distinct components of the model are updated in different stages, indicating an organized, sequential approach to learning. This stage-wise learning strategy introduces a novel perspective on the interplay between weight adaptation and data distribution, highlighting how the architecture distinguishes and captures distinct data characteristics over time. Additionally, the paper provides a theoretical characterization to substantiate its findings, anchoring the analysis in formal proofs.

**Weaknesses:**

(1) The description of the experimental setup is missing, but I assume the results are all based on synthetic data. Therefore, justification with real data is necessary.


(2) The theoretical insights for guiding the training and deployment of Transformers are unclear in practical applications.

**Questions:**

N/A

---

> ### Author Response · Authors · 2024-11-22
>
> We thank Reviewer Kkd8 for the review.
> In the revised paper, we have made changes based on the reviewers' comments and have highlighted all revisions in blue fonts.
>
> **Q** The description of the experimental setup is missing, but I assume the results are all based on synthetic data. Therefore, justification with real data is necessary.
>
> **A** Yes, the experiment results are based on the data distribution we introduced in Definition 2.1. We are actively developing code base to test our Algorithm 1 on real-world dataset. If our experiment results would not come out before the rebuttal deadline due to the complexity of the real world experiments, we commit to include these results in the final version of the paper (if it is accepted) or in the future version of the paper.
>
> We also point out that the primary contribution of this paper lies in the theoretical analysis, which can help us understand and demonstrate the mechanism of neurons and attentions, and how they evolve during training.
>
> **Q** The theoretical insights for guiding the training and deployment of Transformers are unclear in practical applications.
>
> **A** Thank you for the comment. The main theoretical insight from our gradient flow analysis is that during training, the gradient flow (derived from the loss function) is able to effectively guide the transformer's weights toward learning the appropriate representations required to solve the underlying task. Specifically, the gradient of the attention weights actively steers the attention matrix to align with the correct feature directions for classification. This finding is particularly significant because, despite the highly nonconvex nature of the loss function, its gradient is strong enough to overcome the nonconvex landscape and consistently points toward the global minimum. Practically, this insight implies that to train transformers with specific desired properties in practice, we can design the training objective to shape the gradient accordingly, thereby directing the transformer's weights toward the desired solutions.
>
> Another insight from our theoretical analysis is that the training process of transformer can often occur in phases. In our case, the neuron weights of the transformer learn first and after that the softmax attention modules start to learn. This implies that in some cases, we can train transformers layer-wise to reduce the training cost in practice.
>
> In addition, we would like to point out to the reviewer that the purpose of our work is to understand how transformers especially the self-attention modules leverage the connections between tokens to solve certain task. At the same time, we aim to provide theoretical tools that can serve as building blocks and are generalizable to other settings when we analyze the training dynamics of transformers.

---

### Official Review · Reviewer_fKbv · 2024-11-04

**Soundness:** 3
**Presentation:** 2
**Contribution:** 2
**Rating:** 6
**Confidence:** 3

**Summary:**

This paper investigates the dynamics of transformers in learning linear classification tasks where input tokens have hidden correspondence structures. The authors examine a two-headed transformer with ReLU neurons on a specific task involving two classes with hidden linear relationships. They observe a three-stage learning process: (1) neuron alignment with classification signals (2) (2) attention feature learning where features evolve into a separable state and (3) training the neurons again to the final convergence. Baaed on the empirical observation, they propose a three-stage training algorithm and derive the main theorem based on the algorithm.
Their theoretical analysis, using gradient flow dynamics, shows how attention mechanisms enable the model to leverage hidden correspondences to solve classification tasks effectively. The study also considers settings with more than two mixtures, highlighting challenges in analysis yet demonstrating the model's empirical success.

**Strengths:**

**Theoretical Contribution**: The paper provides a rigorous theoretical analysis of transformers’ training dynamics on structured linear classification tasks, specifically using gradient flow dynamics. This theoretical foundation contributes to a deeper understanding of how attention mechanisms in transformers learn hidden relationships, a topic that is not yet fully understood in the community.

**Algorithmic Insights**: The proposed three-stage training algorithm mirrors the empirical learning dynamics, providing a potentially practical framework for initializing and training transformers on tasks with similar hidden structures. This approach could be valuable for applications requiring attention to specific feature-alignment steps.

**Novel Analysis of Attention Dynamics**: The paper effectively addresses the softmax attention mechanism’s behavior within the training dynamics. The authors introduce techniques to simplify and analyze the system, such as reducing the attention's complex nonlinear dynamics to a manageable subsystem, which may serve as a foundation for future analyses in related settings.

**Weaknesses:**

- This paper only consider learning two-mixture of binary classification, and the data distribution is of size $d\times 3$. The main theorem is based on Algorithm 1, an algorithm designed from empirical observations on a synthetic dataset but not tested on any real-world datasets. So, I question whether analyzing Algorithm 1 and this specific choice of dataset is really helpful. I suggest the authors to apply Algorithm 1 on some real-world dataset, and shows that it works. (do not need to be better than standard training).

- Although this paper's title is "two-mixture", there should be more experiments and discussions on $K\geq 3$ than current Appendix A.3.

**Questions:**

See weakness.

---

> ### Author Response · Authors · 2024-11-22
>
> We thank Reviewer fKbv for the review.
> In the revised paper, we have made changes based on the reviewers' comments and have highlighted all revisions in blue fonts.
>
> **Q** This paper only consider learning two-mixture of binary classification, and the data distribution is of size $d\times 3$. The main theorem is based on Algorithm 1, an algorithm designed from empirical observations on a synthetic dataset but not tested on any real-world datasets. So, I question whether analyzing Algorithm 1 and this specific choice of dataset is really helpful. I suggest the authors to apply Algorithm 1 on some real-world dataset, and shows that it works. (do not need to be better than standard training).
>
> **A** Thank you for your suggestions. We would like to point out that
> although the proof idea for our theorem is inspired by a data distribution of size $d\times 3$, the actual proof technique is developed for data distributions of inputs with arbitrary sizes. The mathematical rigor of the proof ensures that the theorem is valid and applicable to more general distributions.
>
> We are actively developing code base to test our Algorithm 1 on real-world dataset. If our experiment results would not come out before the rebuttal deadline due to the complexity of the real world experiments, we commit to include these results in the final version of the paper (if it is accepted) or in the future version of the paper.
>
> We also point out that the primary contribution of this paper lies in the theoretical analysis of Algorithm 1, which can help us understand and demonstrate the mechanism of neurons and attentions, while serving as an accurate model to reflect how the weights in transformer evolve.
>
> **Q** Although this paper's title is "two-mixture", there should be more experiments and discussions on $K\geq 3$ than current Appendix A.3.
>
> **A** Thank you for your suggestions. We have included more experiment results and discussion for the case with $K\geq 3$ in Appendix A.4 in our revision. Our experiment results indicate that the training dynamics of 3-mixture, 4-mixture and 8-mixture models are all showing some similarities as that for 2-mixture model, demonstrating the broad applicability of our developed theoretical results.

---

> > ### Comment · Reviewer_fKbv · 2024-11-23
> >
> > Thanks for your comments. I will keep my score.

---

### Author Response · Authors · 2024-11-22
**Summary of the revision**

We thank all the reviewers for the comments and suggestions.
Since the deadline is approaching, we summarize some most noticeable changes we made here. In our revision, we highlight the changes in blue fonts.

- In our original submission, our analysis is based on the averaged initial value model in Definition 5.1. In our revision, we were able to remove this assumption via mean-field technique by considering an infinite-width transformer, which is considered to be a more natural and easily justifiable technique in theoretical literature.
With infinite-width transformer, the averaged initial value model assumption can be exactly satisfied at initialization. We show this in Lemma C.1 in Appendix C.
We have checked that replacing the original averaged initial value model by infinite-width transformer doesn't introduce much changes in the proof. The discretization to finite-width case can be handled easily in Appendix G.
Furthermore, by adapting to the mean-field technique, we remove the $1/\sqrt{m}$ factor in the softmax attention (section 2.2) and we initialize the attention weights by $\mathcal{N}(0, \omega^2 \frac{1}{m})$ (see Algorithm 1).
One of the consequences of removing the $1/\sqrt{m}$ factor in the softmax attention is that now stage 2 of the training becomes shorter and only runs in $O(\log (1/\omega))$ time (see Theorem 4.1).
We also updated the experiment plots in Section 3.1 accordingly to reflect this change.

- We have included more experiment results on the multi-mixture case in Appendix A.4, and we changed the observation 1 accordingly in Section 6.

- Per request by Reviewer 7eXW, we have made a new construction of transformer for the general $K$ mixtures in Lemma 6.1.

- As suggested by Reviewer 7eXW, we rename \``class'' to \``group'' to avoid confusion.

---

### Meta-Review · Area_Chair_sdsp · 2024-12-22

**Metareview:**

This paper studies the learning dynamics of a two-headed transformer on a two-mixture linear classification task, analyzing how the model learns to utilize hidden correspondences between tokens. The authors propose and theoretically analyze a three-stage training algorithm that characterizes the gradient flow dynamics, showing how neurons first align with signals, followed by attention feature learning and final convergence.
Reviewers appreciated the well-chosen combination of data distribution and architecture that enabled rigorous theoretical analysis while maintaining interpretability. They also noted the clear presentation of complex technical proofs and their potential applicability to broader transformer research.
The main technical concerns raised during discussion centered on the theoretical treatment of finite vs infinite-width transformers, particularly regarding the mean-field approximation and its validity. Through detailed discussion, the authors provided additional analysis showing how the perturbation from finite-width discretization can be bounded, though some technical details required further clarification.
Given the solid theoretical development and the authors' constructive engagement in addressing key technical questions during discussion, I recommend accepting this paper. I encourage the authors to incorporate their clarifications from the discussion period into the final version.

**Additional Comments On Reviewer Discussion:**

See above

---

### Decision · Program_Chairs · 2025-01-22

Accept (Poster)